# The pRb/RBL2-E2F1/4-GCN5 axis regulates cancer stem cell formation and G0 phase entry/exit by paracrine mechanisms

Chao-Hui Chang[1,5], Feng Liu[1,5], Stefania Militi[1], Svenja Hester[2], Reshma Nibhani [1], Siwei Deng [1], James Dunford [1], Aniko Rendek[3], Zahir Soonawalla[4], Roman Fischer [2], Udo Oppermann[1] & Siim Pauklin [1] ✉

The lethality, chemoresistance and metastatic characteristics of cancers are associated with phenotypically plastic cancer stem cells (CSCs). How the non-cell autonomous signalling pathways and cell-autonomous transcriptional machinery orchestrate the stem cell-like characteristics of CSCs is still poorly understood. Here we use a quantitative proteomic approach for identifying secreted proteins of CSCs in pancreatic cancer. We uncover that the cell-autonomous E2F1/4-pRb/RBL2 axis balances non-cell-autonomous signalling in healthy ductal cells but becomes deregulated upon KRAS mutation. E2F1 and E2F4 induce whereas pRb/RBL2 reduce WNT ligand expression (e.g. WNT7A, WNT7B, WNT10A, WNT4) thereby regulating self-renewal, chemoresistance and invasiveness of CSCs in both PDAC and breast cancer, and fibroblast proliferation. Screening for epigenetic enzymes identifies GCN5 as a regulator of CSCs that deposits H3K9ac onto WNT promoters and enhancers. Collectively, paracrine signalling pathways are controlled by the E2F-GCN5-RB axis in diverse cancers and this could be a therapeutic target for eliminating CSCs.

The retinoblastoma protein family (RBs: pRb/RB1, RBL1/p107 and RBL2/p130) are considered bona fide cell-autonomous cell cycle regulators of mammalian cells. RBs control the G1 to S phase transition by reducing the transcriptional activity of E2F proteins (E2Fs), thereby leading to transcriptional repression of target genes necessary for proliferation. In turn, the phosphorylation of RBs by Cyclin D/CDK4-6 blocks interactions with E2Fs, permitting the induction of E2F-mediated transcription[1–4]. The E2F family includes activators (E2F1, E2F2, E2F3a and E2F3b) and members that are described primarily as repressors (E2F4, E2F5, E2F6, E2F7 and E2F8)[5].

RBs are central to early mammalian development[6–13] while E2F components in this cell cycle regulatory axis also impact developmental processes. For instance, E2F1/E2F2 mutant mice develop exocrine pancreatic insufficiency and diabetes[14,15]. Besides their developmental function in tissue formation, RBs have an important role as tumour suppressors while their functional inactivation promotes tumorigenesis. Mutations in RBs are found in various human cancers including breast, pancreas, lung, blood and brain malignancies[5,16–18]. Similarly, E2Fs tend to accumulate genetic alterations in human cancers[5]. Interestingly, the developmental and tumour suppressive effects are intertwined with each other as seen in stem cells. RBs impact embryonic stem cell differentiation as shown by the disruption of the three Rb-related genes in mESCs, while the absence of RB protein function in hESCs induces cell death[19]. Reminiscent of pathological self-renewal compared to the physiological self-renewal of embryonic or adult tissue-specific stem cells[20], Mouse Embryonic

[1]Nuffield Department of Orthopaedics, Rheumatology and Musculoskeletal Sciences, Botnar Research Centre, University of Oxford, Old Road, Oxford OX3 7LD, UK. [2]Target Discovery Institute, Nuffield Department of Medicine, Old Road, University of Oxford, Oxford OX3 7FZ, UK. [3]Department of Histopathology, Oxford University Hospitals NHS Foundation Trust, Oxford, UK. [4]Department of Hepatobiliary and Pancreatic Surgery, Oxford University Hospitals NHS, Oxford, UK. [5]These authors contributed equally: Chao-Hui Chang, Feng Liu. ✉e-mail: siim.pauklin@ndorms.ox.ac.uk

Fibroblasts with a knockout for the three RB genes display a loss of G1 control and cellular immortalisation[21,22]. Furthermore, pRb can restrict reprogramming and tumorigenesis by inhibiting pluripotent stem cell circuitries[23]. These studies hint at a possible role of RBs in cancer stem cells (CSCs)−a subpopulation of cancer cells in various tumours which acquire a stem cell-like state with developmental plasticity reminiscent of naturally occurring stem cells[24–26]. This allows CSCs to efficiently metastasise by an increased invasive capacity, give rise to a cellularly heterogeneous tumour and resist elimination by conventional cancer therapeutics. Hence, this developmentally plastic cancer subpopulation has crucial importance as a target of anti-cancer therapies. The existence of CSCs has been found in the breast, pancreas, brain, colon, oesophagus, liver, lung, ovarian, prostate, stomach and thyroid cancers, among others[24,25]. However, the involved mechanisms and the potential relevance of RB proteins in the stemness characteristics and dedifferentiation of cancer cells are largely unclear. The function of the RB-E2F axis is generally considered to be limited to cell-autonomous effects which impact only the cell expressing the functional or deregulated RB-E2F axis components.

Pancreatic ductal adenocarcinoma (PDAC) is among the most lethal malignancies in humans[27]. The mortality caused by pancreatic cancer is projected to reach second place in the next decade due to its late diagnosis, the incidence of risk factors including obesity and metabolic syndrome[28], and a limited response to treatment[29]. PDAC has a dismal response to chemotherapy and radiotherapy and disease re-emergence is found in 90% of the surgically treated patients[30]. In turn, breast cancer (BRCA) is the most frequent malignancy in women while advanced breast cancer with metastases is considered incurable with currently available therapies[31,32].

In this work, we use a multi-omic approach with quantitative proteomics via SILAC/mass-spectrometry on the secretome of pancreatic CSCs. We uncover elevated WNT ligand secretion with an accompanying β-catenin signalling pathway activation in CSCs. The combination of functional studies and molecular analyses in PDAC and BRCA reveals a non-cell autonomous role for the pRB/RBL1/RBL2-E2F1/4 axis in tissue homoeostasis and tumorigenesis.

## Results
### Characterising CSCs for subsequent quantitative secretome analyse
The PDAC CSCs are a subpopulation of PDAC cells that have a stem cell-like state, allowing them to self-renew and give rise to tumours[24–26]. This phenotype allows CSCs to give rise to the whole tumour with its entire cellular heterogeneity and thereby supports metastases formation and development of resistance to current cancer therapeutics. The existence of developmentally plastic CSCs has been discovered in the brain, breast, colon, oesophagus, liver, lung, ovarian, prostate, stomach and thyroid cancers, among others. In the case of PDAC, the first reports of cancer stem cells date back to 2007[24,25]. Since then, pancreatic CSCs have been shown to be involved in PDAC resistance to chemotherapy, displaying increased prevalence within the tumour after treatment with gemcitabine[33,34]. We have performed extensive experimental validation of the cells used in our study.

In order to establish the composition of CSC states we performed a characterisation of marker expression in our PDAC cell lines by analysing markers that have been associated with the CSC cellular phenotype: EpCAM, CD44, CD24, PROM1/CD133, SSEA4, CXCR4, ABCG2. Cells were grown in 1) standard adherent conditions that do not enrich for CSCs, and 2) in 3D suspension as spheres from single cells that enrich for anoikis-resistant cells that have CSC characteristics. PDAC non-CSCs undergo anoikis as single cells in 3D non-adherent conditions but proliferate readily in adherent conditions. In contrast, CSCs are anoikis-resistant and will give rise to spheres that are enriched for CSCs. We have isolated and characterised the CSCs in detail from different cell lines.

A13A cells showed an increase in CD44, PROM1/CD133, SSEA4 and CXCR4 expression in CSC spheres, whereas L3.6pl cells showed an increase in CD44, PROM1, SSEA4 and ABCG2 expression in CSC spheres (Supplementary Fig. 1a; Supplementary Information), thus indicating that several CSC markers attributed to the CSC populations are enriched in 3D spheres in our cell lines. To investigate the self-renewal capacity we sorted cells into CD133+/SSEA4+ and CD133-/SSEA4- subpopulations, and performed tumour sphere assays on these by using unsorted cells as a control. Marker-positive cells showed a significantly higher number of spheres compared to both marker negative and unsorted cells (Supplementary Fig. 1b), indicating increased self-renewal capacity for CSC marker-positive cells.

To investigate the chemoresistance of the CSC population in our PDAC cells, we treated the cells with chemotherapy reagents Gemcitabine (GEM), FOLFIRINOX, and Nab-paclitaxel for 5 days. Gemcitabine, FOLFIRINOX and Nab-paclitaxel treatment of heterogeneous PDAC cells for 5 days enriched for cells expressing CD133+/SSEA4+, CD44+/CD133+, OCT4+/CD133+, OCT4+/SSEA4+ from ~1% double positive cells to 80%, 60–75%, 18–20% and 16–25%, respectively (Supplementary Fig. 1c). Triple positive cells increased from 0.5% to 33%, 21% and 14% upon Gemcitabin, FOLFIRINOX and Nab-paclitaxel treatment (Supplementary Fig. 1d). The increase of the CSC marker expressing cells occurs by inducing cell death of a large majority of PDAC non-CSCs that do not express these CSC markers because 95-99% of PDAC cells treated with Gemcitabine, FOLFIRINOX, and Nab-paclitaxel die through apoptosis (Supplementary Fig. 1e). This results in selective survival and enrichment of the rare CSCs expressing CD133, SSEA4, CD44 and OCT4. Collectively, CSCs survive and are increased upon treatment with anti-cancer agents whereas non-CSCs decrease due to apoptosis. Next, we also investigated the chemoresistance of cells grown in 3D sphere conditions that enrich of CSCs, and standard adherent 2D culture conditions. Cells grown as spheres have higher chemoresistance as shown by the higher survival of CSC spheres upon Gemcitabine and FOLFIRINOX treatment, whereas cells grown in standard adherent 2D culture conditions have higher chemosensitivity as indicated by the more drastic loss of viable cells (Supplementary Fig. 1f).

Since the TGFβ/Activin-SMAD2/3 signalling pathway regulates stem cell-like characteristics of CSCs[35,36], we also tested the impact of TGFβ/Activin signalling on CSC resistance to currently used chemotherapeutics by treating the cells with the TGFβ/Activin signalling inhibitor SB431542 in combination with Gemcitabine and FOLFIRINOX. Inhibition of TGFβ/Activin signalling strikingly reduced the chemoresistance of PDAC cells as indicated by reduced numbers of CSC marker-expressing cells and therefore the overall number of surviving PDAC cells (Supplementary Fig. 1f). These results emphasise the crucial importance of TGFβ/Activin signalling on CSC maintenance and their elevated chemoresistant characteristics.

We confirmed the relevancy of our CSC markers in primary human PDACs at single-cell level by analysing single-cell RNA-sequencing data[37]. This showed a subset of PDAC cells co-expressing OCT4+/CD133+/SSEA4+ markers, whereas cell trajectory modelling of these cells indicates their likely role in giving rise to the different cell populations in patient PDACs (Supplementary Fig. 1g–i). Hence these CSC markers have in vivo co-expression and relevance in human PDACs, which support previous studies characterising these markers.

### Multiomic analysis of secretomes and transcriptomes reveals elevated WNT ligand expression and secretion by pancreatic CSCs compared to non-CSCs
PDACs are aggressive and metastatic cancers, and their high mortality reflects inefficient therapeutics. Since pancreatic CSCs have a central role in metastatic dissemination and chemoresistance[38,39], we investigated the paracrine effects in this therapeutically important cancer cell population. We hypothesised that human pancreatic CSCs have a

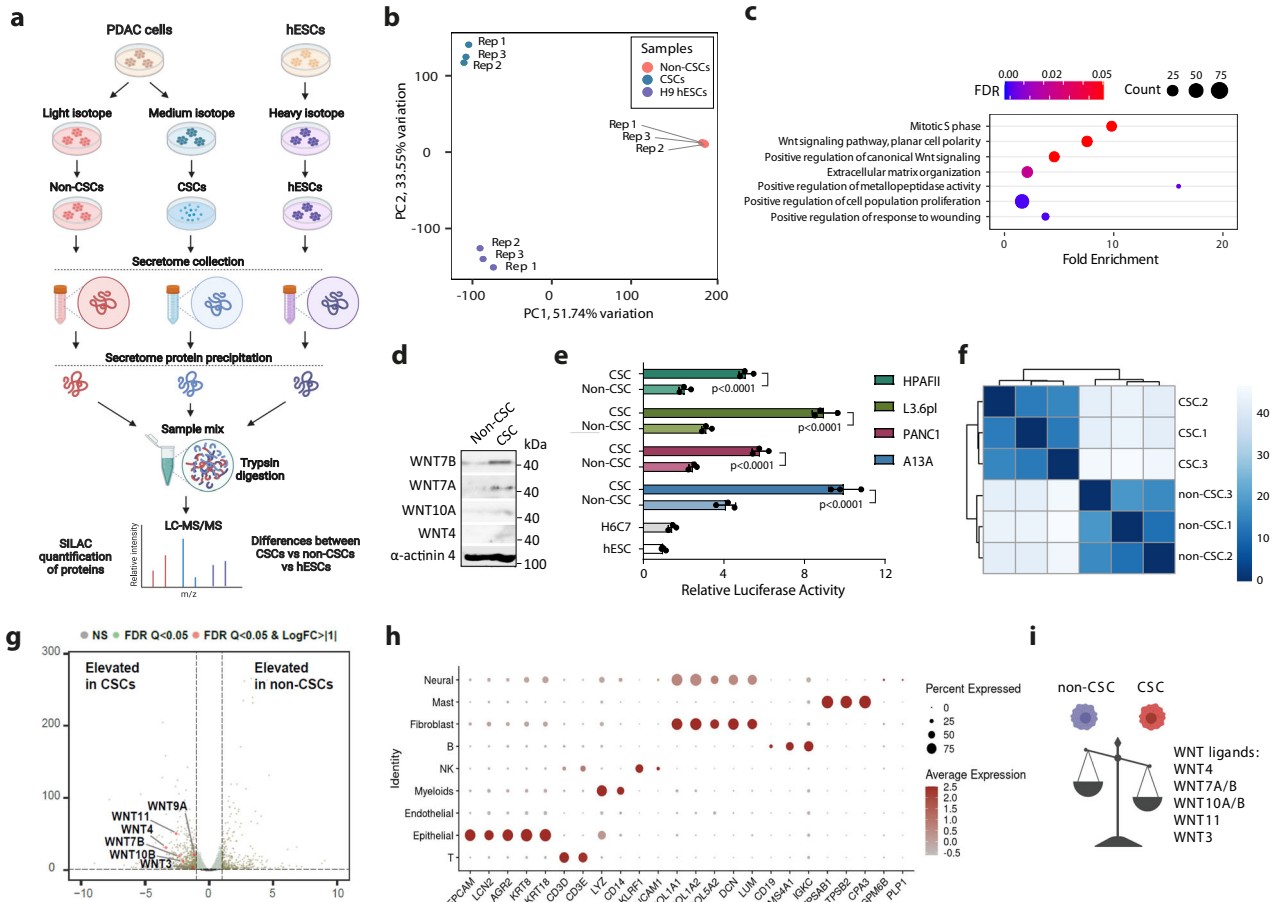

**Fig. 1 | Secretome analysis of pancreatic CSCs by quantitative SILAC/mass-spectrometry and transcriptomic analysis identifies WNT signalling pathway to be elevated in CSCs. a** Schematic depiction of the SILAC/mass-spectrometry experimental outline. **b** Principal Component Analysis (PCA) of secretome samples shows differences between the secretomes of cell types. **c** Pathway enrichment analysis indicates elevated WNT signalling in CSCs. **d** WNT ligand secretion is increased in CSCs compared to non-CSCs. **e** Promoter luciferase analysis with a construct containing TCF/LEF sites upstream of a luciferase reporter (M50 Super 8x TOPFlash) indicates elevated β-catenin transcriptional activity in pancreatic CSCs compared to non-CSCs and hESCs. Data are presented as mean values ± SD. *N* = 3 independent experiments. Statistical analysis was performed by 2-way ANOVA with multiple comparisons with Tukey correction. **f** Hierarchical clustering analysis of correlation coefficients for gene expression in RNA-seq samples. **g** Volcano blot of differential gene expression shows elevated expression of WNT ligands in pancreatic CSCs compared to non-CSCs. **h** Marker expression in different cell types in PDAC patient tumour sample RNA-sequencing data. **i** Schematics showing higher expression of WNT ligands in pancreatic CSCs compared to non-CSCs. Source data are provided as a Source Data file.

different secretome compared to non-cancer stem cells or embryonic stem cells. To identify the secretomes we decided to perform quantitative proteomics with SILAC-mass-spectrometry analysis to characterise the secretomes of CSCs compared to non-CSCs and hESCs. In order to quantitatively characterise the secreted factors, we first incorporated light (media supplemented with L-Arginine and L-Lysine), medium (supplemented with 13C6-L-Arginine-HCl and D4-L-Lysine) and heavy (supplemented with 13C615N4-L-Arginine-HCl and 13C615N4-L-Lysine) SILAC isotopes into either PDAC A13A cell line[40] CSCs, PDAC A13A non-CSCs or H9 hESCs respectively in each SILAC medium for at least five doublings. The CSCs from A13A PDAC cell line were firstly selected using CD133 magnetic beads and were anoikis-resistant and able to self-renew while being enriched for CSC marker expression when cultured in non-adherent tumour sphere-forming conditions as we have verified also previously[35]. We confirmed isotope incorporation by mass-spectrometry, which was >97% in secretome and >99% in total lysate, and then proceeded with analysing the secretomes of CSCs, non-CSCs and hESCs collected after 48 h followed by sample processing and quantitative SILAC/mass-spectrometry (Fig. 1a). Principal Component Analysis (PCA) of the samples revealed a clear difference between the three cell types, while all three biological replicates of the same cell types were reliably close to each other, thus

indicating good reproducibility (Fig. 1b). Pathway analysis of the identified proteins using Gene Ontology Biological Process (GO BP) revealed the enrichment of pathways in CSCs involved in extracellular matrix organisation, positive regulation of cell population proliferation, mitotic S phase regulation and positive regulation of metallopeptidase activity. Interestingly, the enriched pathways also included positive regulation of the canonical WNT signalling as well as WNT signalling pathway with cell polarity regulation (Fig. 1c). Among the detected WNT/β-catenin pathway components elevated in A13A CSCs were WNT7B, WNT7A, WNT10A and WNT4 ligands, which show elevated secretion in CSCs compared to non-CSCs in separate Western blotting experiments of conditioned media (Fig. 1d). β-catenin is essential in acinar-to-ductal metaplasia and Dkk1 secreted negative regulator of WNT pathway blocks the initiation of early PanIN lesions formation. However, so far, there is no evidence that WNT signalling would be essential for cancer maintenance or propagation where CSCs would be expected CSC play a role[41]. To investigate the activity of WNT pathway in CSCs versus non-CSCs we transfected cells from four distinct PDAC lines (HPAFII, L3.6pl, PANC1 and A13A) with a M50 Super 8x TOPFlash β-catenin reporter containing TCF/LEF sites upstream of a luciferase reporter and analysed the luciferase signal after 48 h (Fig. 1e). CSCs from all four cell lines showed elevated β-catenin activity

compared to their non-CSCs which are consistent with the enrichment of WNT pathway in A13A CSC as observed by secretome analyses.

WNT pathway activity is initiated by secreted WNT ligands which could be differentially regulated in CSCs vs non-CSCs at the protein secretion level or at earlier stages of gene expression such as transcriptional induction of the genes. To examine the expression of WNT ligands at the mRNA level we performed transcriptomic analyses of CSCs and non-CSCs to compare with the secretome data. RNA-sequencing was performed in triplicate and unsupervised clustering indicated reliable differences between CSCs and non-CSCs (Fig. 1f ). Differential gene expression analysis revealed that several WNT ligands have elevated mRNA levels in CSCs compared to non-CSCs (Fig. 1g), suggesting that WNT ligands are transcriptionally upregulated in CSCs over non-CSCs and this also results in increased secretion of these WNT ligands at the protein level.

In order to investigate the ligand-receptor signalling that might occur in patient-derived primary PDACs, we analysed the ligand-receptor pairs by predicting the pairs between ligands identified by our quantitative secretome identification and the mRNA expression at the single cell level of their corresponding receptors in cancer cell subpopulations in patient tumours[42]. For this, we used single-cell RNA-sequencing expression data in different cell types detected in primary PDACs from pancreatic cancer patients. We annotated these gene expression signatures in PDAC patient samples which revealed that the tumours from PDAC patients contained epithelial cancer cells but also T-cells, myeloid cells, NK cells, fibroblasts, B-cells, endothelial cells, mast cells and neural cells (Fig. 1h). We further clustered the subpopulations of cancer cells specifically (Supplementary Fig. 2a). One subpopulation among the cancer cells (PDAC_C1 cluster) had a more extensive expression of several well-known CSC markers[43], such as CD133, CD44, KLF4, and ALDH1A1 (Supplementary Fig. 2b), indicating that this subpopulation could more closely resemble PDAC CSCs in our cell lines grown as spheres. This cancer cell subpopulation also showed statistically significant elevated expression of WNT7B, WNT7A, WNT10A and WNT4 besides the CSC markers, compared to the rest of the PDAC cells (Supplementary Fig. 2c; Supplementary Table 6). Altogether, CSCs have increased WNT ligand (WNT7A/B, WNT10A, WNT4) expression (Fig. 1i).

Next, we investigated the expression of the factors identified by our secretome experiments, in the subpopulations of cancer cells in PDAC patient tumour samples. The annotation of ligand-receptor interactions indicated the presence of putative CSC-specific signalling with other non-CSC cancer cell subpopulations (Fig. 2a). To model the implied crosstalk between CSCs (PDAC_C1 cluster) and other cancer cell subpopulations in primary tumours, we paired the expression of the factors identified in our secretome studies, and their receptors on each of the cancer cell populations present in the PDAC patient tumours. The paracrine signalling between CSCs and subpopulations of cancer cells revealed extensive bidirectional crosstalk, indicating that CSCs can provide autocrine signalling but also can affect the other non-CSC cancer cell subpopulations, and vice versa (Fig. 2a, b). Among the ligand-receptor pairs mediating the crosstalk are FZD3-LRP5/6-WNT7B, FZD5/6-LRP5/6-WNT7A/7B, FZD5-LRP5/6-WNT10A (Fig. 2a, b). The WNT ligands and their receptors FZD5 and FZD6 are expressed at variable levels in PDAC tumour in the cancer cell subpopulations (collectively designated epithelial cells in Fig. 2c) but also in the surrounding stromal cell types, especially fibroblasts, and immune cells (Fig. 2c, d; Supplementary Fig. 2c). Among the WNT ligands, WNT7A, WNT7B and WNT10A have the highest expression in cancer cells (Fig. 2c, d), and the largest increase in expression in bulk PDACs compared to normal pancreatic tissue (Supplementary Fig. 3a). Based on TCGA/GTEx data accessed through GEPIA2[44], their elevated expression in top 25% expressing versus bottom 25% expressing cancer patients correlates very significantly with lower patient survival (Supplementary Fig. 3b, c) suggesting that WNT expression has an

important supportive function on disease progression of PDAC patients. The expression among WNT ligands is most abundant for WNT7B (Fig. 2d) which has been implicated in the past to have a prominent role in PDACs[45].

Collectively, these results indicate that the WNT/β-catenin signalling pathway is active in CSCs and this seems to be mediated by increased WNT ligand (particularly WNT7A/B, WNT10A, WNT4) expression and elevated WNT ligand secretion into the extracellular space in these cells. In the tumour this mediates autocrine signalling of CSCs but also paracrine signalling between cancer cell subpopulations.

## WNT ligand loci in pancreatic CSCs are regulated by E2F-RB binding and KRAS-CDK4/6/2 activity

Pancreatic cancer has an increasing prevalence in western countries and it is one of the most lethal cancers in humans with highly metastatic characteristics and poor responsiveness to currently used chemotherapies such as gemcitabine partly due to the high therapeutic resistance of CSCs[38]. To uncover potential transcriptional regulators of WNT ligand genes in pancreatic CSCs we decided to perform chromatin accessibility analysis by ATAC-sequencing in A13A CSCs and non-CSCs. Principal Component Analysis revealed clear separation of non-CSC and CSC samples, thus indicating differences in ATAC-sequencing peaks (Fig. 3a), and sample replicates clustered together showing good reproducibility (Fig. 3b). We identified 52049 peaks in non-CSCs and 64416 peaks in CSCs (Fig. 3c). The general distribution of ATAC-seq peaks was not fundamentally different in non-CSCs and CSCs since approximately a third of the peaks were located at gene promoters and a third was in the introns or exons in both non-CSCs and CSCs (Fig. 3d). Motif enrichment analyses of the total peaks indicated CTCF, BORIS, and the AP-1 family members (FRA1, FRA2, JUN, JUNB) as the top transcription factor motifs in non-CSCs and CSCs (Supplementary Tables 7–3), thus suggesting that there is not a major change among the most abundant transcription factor motifs. However, we uncovered 8522 differential ATAC-seq peaks between CSCs and non-CSCs, which were nearby 957 genes. ATAC-seq peaks nearby 352 genes were non-CSC specific, whereas peaks near 605 genes were CSC specific (Fig. 3e). WNT8A peak was non-CSC specific, whereas WNT10B was found in CSCs. The other ATAC-seq peaks at the TSS or in the close proximity were shared by non-CSCs and CSCs. ATAC-seq peaks at the TSS or in the close proximity in CSCs included WNT1, WNT2B, WNT3, WNT3A, WNT4, WNT5A, WNT5B, WNT6, WNT7A, WNT7B, WNT9A, WNT9B, WNT10A, WNT10B, WNT11 and WNT16 (Fig. 3e). Importantly, our transcription factor binding motif analyses uncovered the presence of E2F1 and E2F4 motifs in the ATAC-seq peaks at the proximal promoters of several WNT loci (Fig. 3f, g). These WNT ligands are expressed in PDAC cells as shown by analysis of scRNA-seq data (Fig. 3h). Therefore, we analysed ChIP-sequencing data of E2F1 and E2F4[46–49], which uncovered that the regions in the proximity of WNT loci are bound by E2F1 or E2F4 transcription factors (Fig. 3i; Supplementary Fig. 3d, e).

E2F transcription factors are known as central regulators of cell-autonomous effects on cell cycle progression in most mammalian cells by cooperating with the Retinoblastoma (RBs: pRb/RB1, RBL1/p107 and RBL2/p130) family proteins that upon binding switch the E2F transcription factor complex from transcriptional activators to transcriptional repressors. Thus, a central regulatory mechanism for E2Fs is their regulation via RB1/RBL1/2 phosphorylation through CDKs. Increased E2F transcriptional activity during tumour development is often achieved by increased expression or genomic amplification of the Cyclin D or Cyclin E genes that lead to increased CDK4/6/2 activity. Similarly, RB1/RBL1/RBL2 can be deleted in cancers, which leads to increased E2F activity. However, there have been reported other regulatory routes through increased E2F expression in various cancers[50–57]. Hence, the expression of E2Fs can impact its target gene expression in non-cancerous cells and possibly in PDAC cells. Given the

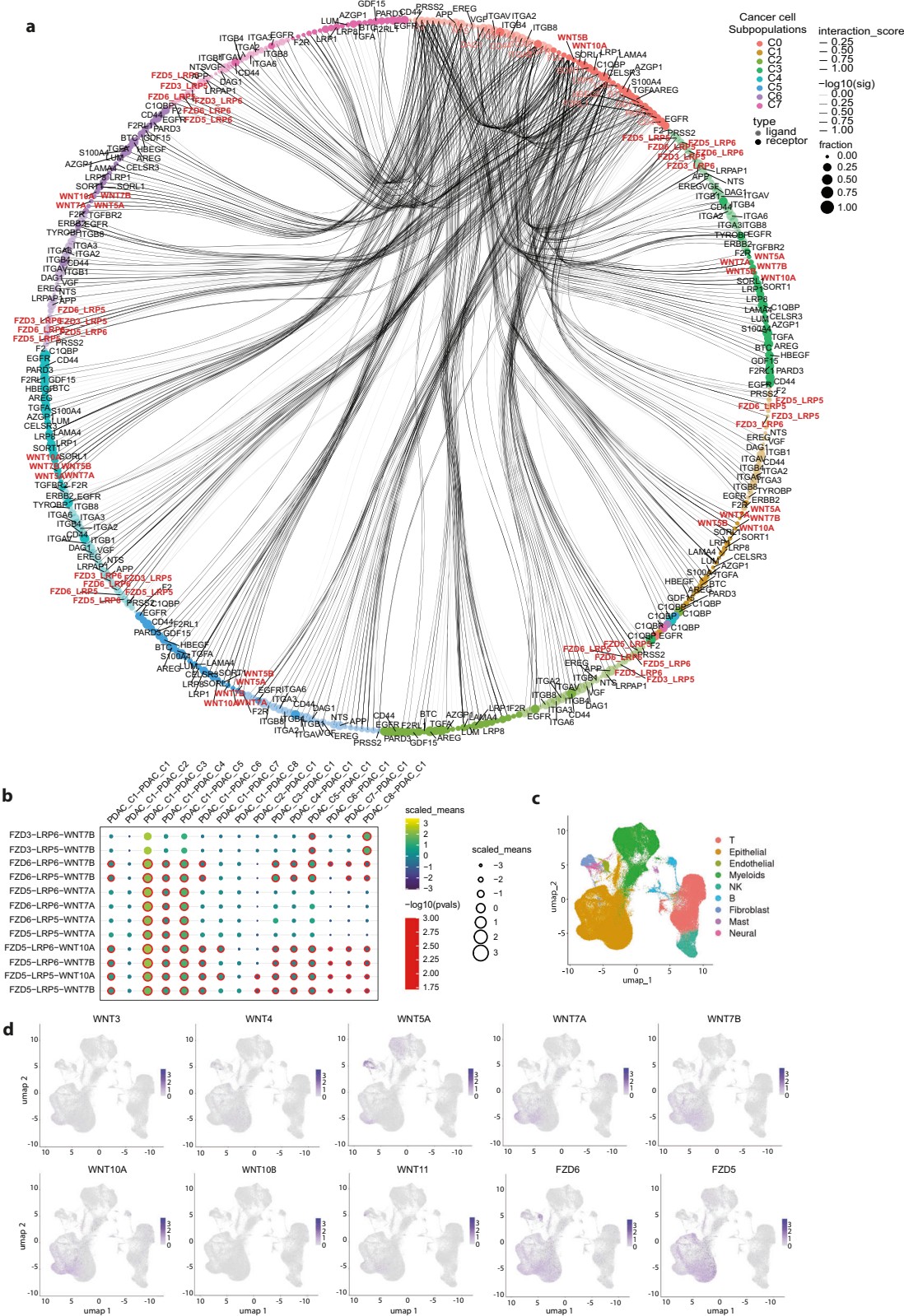

**Fig. 2 | Paracrine signalling between CSCs and cancer cell subpopulations.**
**a** Paracrine signalling between CSCs (PDAC Cluster 1) and cancer cell subpopulations in PDAC patient primary tumours. **b** Paracrine signalling between ligand-receptor pairs mediating the paracrine crosstalk between CSCs (PDAC Cluster 1)
and cancer cell subpopulations in PDAC patient primary tumours. **c** Cell type annotation in PDAC patient tumour sample RNA-sequencing data. Epithelial cells mark cancer cells. **d** WNT ligand and FZD5/6 expression in PDAC patient tumours based on single-cell RNA-sequencing.

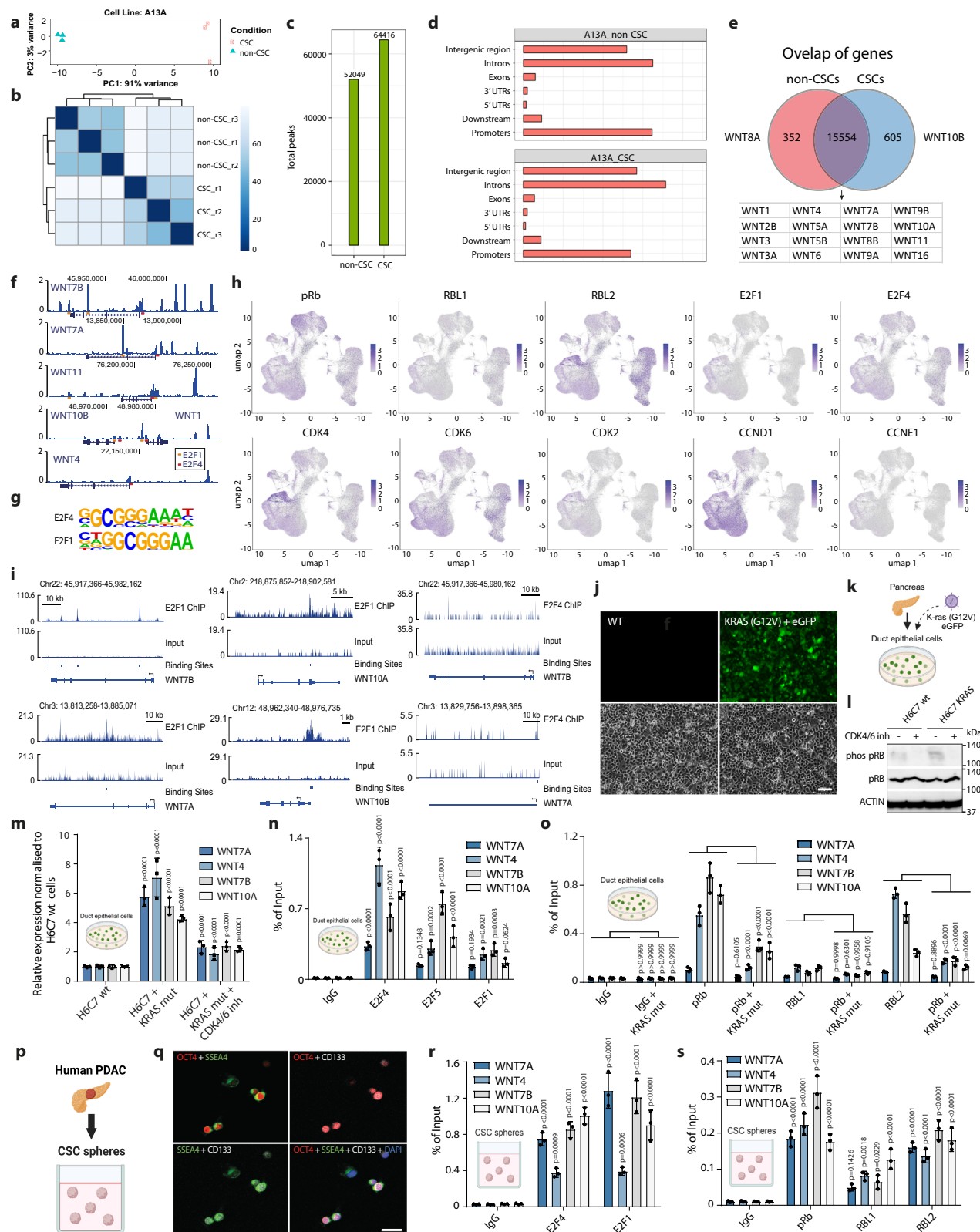

presence of E2F binding motifs on WNT ligand regulatory regions, we first investigated whether there could be a correlation of gene expression between the individual eight E2F family members (E2F1-E2F8) and the combined expression of WNT ligands (WNT3, 4, 7A, 7B, 10A, 10B, 11) in the normal non-cancerous pancreatic tissue. Analysis of GTEx data for 169 normal pancreatic tissue samples revealed that all E2Fs have a correlation with WNT ligand expression (Supplementary

Fig. 4a). One of the strongest positive correlation is with E2F4 ($R = 0.8$, $p$ value = 0) and E2F5 ($R = 0.73$, $p$ value = 0) while E2F4 has also the highest expression in pancreatic tissue. This indicates a possible mechanistic circuitry between E2Fs and WNT ligands during physiological tissue maintenance and homoeostasis. Since the activity of E2F proteins is tightly regulated by Cyclin D-CDK4/6 and Cyclin E-CDK2 complexes, we also looked at their correlation with WNT ligands in

**Fig. 3 | E2F1/4 and RB family proteins bind to regulatory regions near WNT ligand loci in CSCs. a** Principal Component Analysis of ATAC-seq data in A13A CSCs and non-CSCs. **b** Hierarchical clustering analysis of correlation coefficients for gene expression in RNA-seq samples. **c** ATAC-seq peak number in A13A non-CSCs and CSCs. **d** The distribution of ATAC-seq peaks in the genome. **e** The overlap of genes with peaks near their loci in A13A non-CSCs and CSCs. **f** E2F1 and E2F4 binding motifs are present around WNT loci based on ATAC-seq data analysis in pancreatic CSCs. **g** E2F4 and E2F1 consensus binding motifs at total ATAC-seq peaks. **h** Expression of RBs, E2F1/4, CDKs, Cyclin D1 and Cyclin D3 in the cell types in PDAC patient samples analysed by single-cell RNA-seq. and FZD5/6 expression in PDAC patient tumours based on single-cell RNA-sequencing. **i** E2F1 and E2F4 binding to WNT loci based on ChIP-seq data. **j** Representative images of H6C7 wt and H6C7 KRAS (G12V) cells. Live cells were visualised by fluorescence microscopy for detecting eGFP signal in addition to brightfield images. Scale bar 50 µm. **k** Schematic depiction of the H6C7 non-cancerous pancreatic ductal cells and the

introduction of a KRAS (G12V) oncogenic mutation which is an early hallmark mutation of pancreatic ductal adenocarcinoma. The KRAS (G12V) cells also express eGFP. **l** CDK4/6-mediated phoshorylation of pRb protein is increased in cells with constitutively activated KRAS. **m** KRAS (G12V) increases WNT ligand expression in H6C7 cells. **n** E2F4 and E2F5 bind to the regulatory regions near WNT ligand loci in H6C7 wt cells. **o** pRb and RBL2 binding to WNT ligand loci is attenuated by KRAS mutation. **p** Schematic depiction of the CSC model system used in the following experiments. **q** Immunofluorescence imaging of cancer stem cell markers OCT4, CD133 and SSEA4 in CSCs. **r** E2F4 and E2F1 bind to the regulatory regions near WNT ligand loci in pancreatic CSCs. **s** RB family proteins bind to WNT ligand loci in CSCs. Data are presented as mean values ± SD. All bar graphs depicted in this figure are $n = 3$ independent experiments. Statistical analysis was performed by 2-way ANOVA with multiple comparisons with Tukey correction. Source data are provided as a Source Data file.

pancreatic tissue. Cyclin D1, CDK4, CDK6, Cyclin E and CDK2 all showed a statistically significant positive correlation with WNT ligand expression (Supplementary Fig. 4a) whereas pRB showed a negative correlation ($R = -0.3$, $p$ value = 9.5e-05; Supplementary Fig. 4b), indicating that the E2F/Cyclin-CDK/RB axis could regulate the expression on WNT ligands in normal pancreatic tissue. This relationship could allow for tissue development during organogenesis or tissue homeostasis by gene expression regulation, thereby mediating cellular function, cell identity and differentiation, or proliferation upon tissue damage.

Since E2Fs, particularly E2F4 and E2F5, showed a strong positive correlation with WNT ligand expression in normal pancreatic tissue, we utilised H6C7 non-cancerous pancreatic ductal epithelium cells (Fig. 3j, k). Because KRAS oncogenic activation through mutations is the earliest event in PDAC formation, we also used H6C7 non-cancerous pancreatic ductal cells stably transfected with constitutively active KRAS (G12V) (Fig. 3j, k). Oncogenic KRAS activation leads to CDK4/6-mediated elevated phosphorylation of pRb compared to wt H6C7 (Fig. 3l). KRAS activation also increased WNT ligand expression compared to wild-type cells while inhibition of CDK4/6 with a small molecule inhibitor Palbociclib (PD-0332991) decreased WNT ligand expression in KRAS mutant cells (Fig. 3m). Using wild-type H6C7 non-cancerous pancreatic ductal cells we analysed E2F4 and E2F5 but also pRb, RBL1 and RBL2 binding to WNT loci by ChIP-qPCR on the regions identified by ATAC-seq and ChIP-seq (Fig. 3n, o). These analyses revealed robust binding of E2F4 and E2F5 (Fig. 3n) and RBs pRb and RBL2 (Fig. 3o) on WNT loci in wild-type cells. On the other hand, constitutive KRAS activation in these cells by G12V mutation reduces the binding of pRB and RBL2 on WNT loci compared to wildtype cells (Fig. 3o). Our results suggest that the E2F-RB axis regulates the intensity of WNT signalling by direct transcriptional regulation in normal pancreatic ductal cells while the introduction of the early hallmark mutation KRAS G12V in pancreatic ductal cells can cause an imbalance of WNT ligand regulation by changing the binding of E2Fs and RBs to the loci of these secreted factors.

Next, we analysed similar gene expression correlations in PDACs. The gene expression correlations in 179 TCGA PDAC samples between E2F family members and the combined expression of WNT ligands (WNT3, 4, 7A, 7B, 10A, 10B, 11) indicated a statistically significant positive correlation for E2F1, E2F3, E2F4, E2F6 and E2F7 with WNTs in PDACs (Supplementary Fig. 4c). Since E2F1 and E2F3 are activating transcription factors, while E2F4 can exert both activating and inhibitory effects on gene transcription, the positive correlation provides support for the hypothesis that these E2F transcription factors can induce WNT ligand expression during tumorigenesis in PDACs. Additionally, Cyclin D1, CDK4, CDK6, Cyclin E and CDK2 all showed a statistically significant positive correlation with WNT ligand expression in PDACs (Supplementary Fig. 4d). Given this information, we decided to

further explore the functional and mechanistic links between the E2F-RB axis and WNT ligands in PDAC cells with further experiments.

This line of experiments emphasised a possible function of E2F4 and E2F1 in PDACs. To investigate if E2Fs and RBs are also able to co-bind to the predicted regions at the binding motifs we used A13A PDAC CSCs and PDAC organoids as bulk PDAC/non-CSCs. Patient-derived organoids represent the bulk of the PDAC cancer cells, non-CSCs, which have been shown to retain inter- and intrapatient variations of the disease while maintaining multiple aspects of cancer traits and being suitable for modelling therapeutic drug response in patients with high accuracy[58,59]. Among the E2F family members, E2F4 and E2F1 have the highest combined expression in three different human PDAC patient-derived organoids (HCM-BROD-0124-C25; HCM-BROD-0009-C25; HCM-CSHL-0073-C25) based on RNA-seq data available via the Human Cancer Models Initiative (Supplementary Fig. 4e) while in CSCs of PDAC cell lines E2F4 is similarly the most abundant (Supplementary Fig. 4f). ChIP-qPCR analyses revealed that the transcription factors E2F1 and E2F4 both bind to the identified binding sequences in the proximity of WNT7A, WNT7B, WNT10A and WNT4 loci in CSCs (Fig. 3p–r). Similarly, RBs bind to the same regions as E2Fs nearby WNT loci in CSCs (Fig. 3s). E2F1/4 and RBs bound to WNT loci also PDAC organoids (Supplementary Fig. 4g, h). The inhibition of CDK4/6 enzymatic activity with the small molecule inhibitor PD-0332991 significantly increased the enrichment of pRb and RBL2 on WNT loci (Supplementary Fig. 4h, i), suggesting that CDK4/6 enzymatic activity reduced pRb/RBL2 function on WNT loci. It should also be noted that these same regions in the proximity of WNT loci are bound by E2F1 or E2F4 in other cancer types including breast cancer as shown by E2F1 and E2F4 ChIP-sequencing data analyses[46–49] (Fig. 3i and Supplementary Fig. 3f, g), suggesting that some E2F binding regions are shared across diverse cancer types including BRCA and PDAC.

Next we studied the mechanisms of WNT signalling in CSCs by hypothesising that WNT/β-catenin signalling impacts the maintenance of CSCs. Therefore, we investigated the expression of transcription factors that are involved in the maintenance of CSCs[51]. We found a number of CSC regulatory factors induced by WNT signalling and repressed by WNT inhibition by using qPCR for gene expression analyses. These include CSC maintenance factors *HOXA4*, *HOXA5*, *HOX3A*, *YAP1*, *MSI2*, *HIF1A*, *NOTCH2*, *MEIS1*, *OCT4*, *NES*, which are known as self-renewal factors in CSCs in PDACs and various other cancers (Supplementary Fig. 4i). Furthermore, these self-renewal factor loci are bound by β-catenin as shown by ChIP-sequencing analyses (Supplementary Fig. 4j) of published data ([60]; ENCODE: GSM816437), and ChIP-qPCR of β-catenin in A13A CSCs treated with WNT7A/7B ligands (Supplementary Fig. 4k). Collectively, these data indicate that WNT/β-catenin signalling directly regulates the expression of several well-known CSC self-renewal factors in pancreatic CSCs.

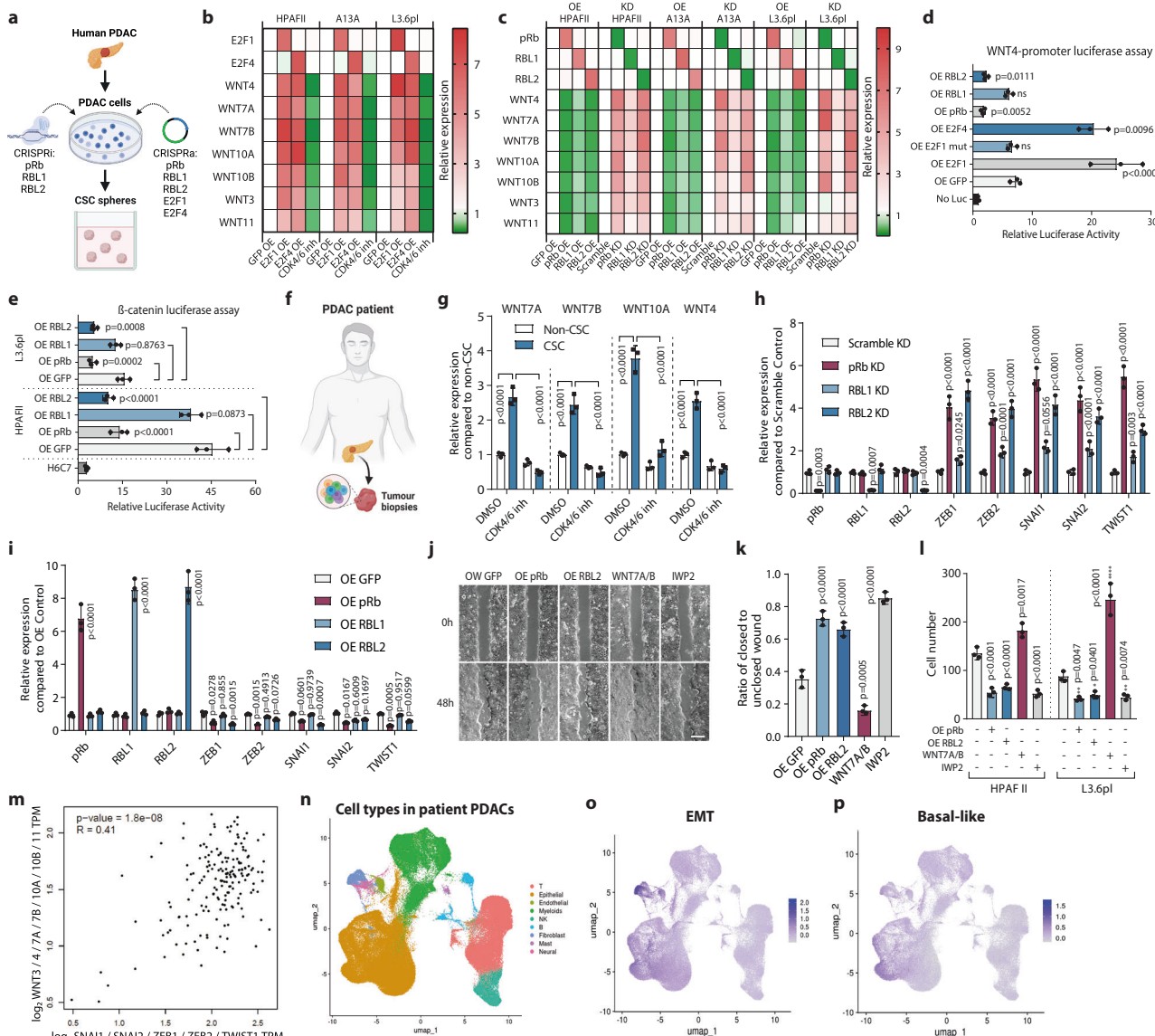

**Fig. 4 | E2Fs induce WNT ligands while RB family proteins reduce WNT ligands in CSCs. a** Schematic depiction of CRISPRi knockdown and CRISPRa induction of RB proteins in CSCs. **b** Overexpression of E2F1 and E2F4 induces WNT ligands whereas CDK4/6 inhibition reduces WNT ligands in CSCs. The heatmaps in (**b**, **c**) show the normalised mean mRNA expression of three biological replicates analysed by qPCR. **c** Overexpression of RB family proteins reduces but knockdown induces the expression of WNT ligands in CSCs. **d** E2F1/4 induce whereas RB family proteins reduce WNT4 transcription as revealed by WNT4 promoter-luciferase assays in CSCs. Data are presented as mean values ± SD. *N* = 3 independent experiments. Statistical analysis was performed by 1-way ANOVA with Dunnett's multiple comparisons test. **e** RB family proteins lead to the reduction of β-catenin dependent transcription in CSCs as measured by the M50 Super 8x TOPFlash construct. Data are presented as mean values ± SD. *N* = 3 independent experiments. Statistical analysis was performed by 1-way ANOVA with Dunnett's multiple comparisons test. **f** Schematic depiction of using primary tumour sample from PDAC patients. **g** WNT ligands have elevated expression in CSCs over non-CSCs that

depends on CDK4/6 activity in primary tumour sample from PDAC patients. **h** Knockdown of RBs leads to the upregulation of EMT-inducing transcription factors. **i** Overexpression of RBs leads to the downregulation of EMT-inducing transcription factors. **j**, **k** Conditioned media from OE RB cells have slower wound healing whereas WNT ligand promotes wound healing as shown by (**j**) brightfield images and **k** measurement of the unclosed wound area. Scale bar 50 μm. **l** Conditioned media from OE RB cells supports lower cell invasiveness compared to control OE GFP cell-conditioned media whereas WNT ligand promotes cell invasiveness. **m** WNT ligand expression is positively correlated with the expression of EMT-inducing transcription factors in PDACs. **n** Cell type annotation in PDAC patient tumour sample RNA-sequencing data. Epithelial cells mark cancer cells. **o**, **p** EMT-enriched and basal-like gene signatures in the PDAC patient tumours. Data are presented as mean values ± SD. Bar graphs depicted in (**g**, **h**, **i**, **k**, **l**) are *n* = 3 independent experiments. Statistical analysis in (**g**, **h**, **i**, **k**, **l**) was performed by 2-way ANOVA with multiple comparisons with Tukey correction. ns = not significant. Source data are provided as a Source Data file.

## E2Fs induce while RBs repress WNT ligand expression in pancreatic cancer stem cells

To investigate the effects of E2F1/4 and RBs on WNT gene expression we used CSC spheroids and patient-derived PDAC organoids for CRISPRi and CRISPRa-mediated mechanistic and functional studies (Fig. 4a). First, we established the relevant PDAC cells for CRISPR/dCAS9-VPR mediated overexpression or CRISPR/dCAS9-KRAB

mediated gene repression of E2Fs or RBs. To uncover the effects in pancreatic CSCs we analysed WNT gene expression (WNT4, 7A, 7B, 10A, 10B, 3, 11) upon E2F overexpression in CSCs from three PDAC cell lines. These results indicated elevated WNT ligand expression upon E2F1 and E2F4 overexpression (Fig. 3b), indicating that E2Fs can have some overlapping functions in WNT regulation. Of note, the function of E2F4 on WNT ligands was activating although E2F4 has in the past

been mostly associated with gene repressive effects. The inhibition of CDK4/6 by the small molecule compound PD-0332991 led to decreased WNT ligand expression (Fig. 4b) suggesting that CDK4/6 inhibition can no longer inactivate RBs through hyperphosphorylation and hence RBs can repress WNT ligand expression. Next, we studied the effect of all three RBs (pRb, RBL1, RBL2) on WNT ligands. Over-expression (OE) of pRB and RBL2 led to reduced expression of WNT ligands in CSCs while CRISPR-mediated knockdown of pRb and RBL2 increased WNT ligands (Fig. 4c) suggesting that RBs have partly overlapping effects on WNT repression in PDAC cells. RBL1 OE and KD had consistently weaker effects compared to pRb and RBL2 in PDAC cells. Next, we studied promoter activation by RBs and E2Fs. For this, we cloned the WNT4 promoter region containing the E2F motif into a luciferase construct and transfected it into PDAC cells together with E2F or RB constructs (Fig. 4d). OE E2F1 resulted in increased luciferase activity while mutated E2F1 version did not increase luciferase activity. Overexpressing E2F4 similarly increased luciferase signal while OE of pRB and of RBL2 decreased the luciferase signal, whereas OE RBL1 did not significantly change luciferase activity compared to the control OE GFP transfection. Furthermore, OE of pRb and RBL2 led to a reduction of β-catenin-dependent transcriptional activity with a promoter-luciferase assay in CSCs enriched from two separate PDAC lines (Fig. 4e) and also patient-derived PDAC organoids (Supplementary Fig. 5a) indicating that the effects on WNT ligand expression translate to effects on β-catenin-dependent transcription.

We used PDAC patient-derived cancer organoids to further char-acterise WNT regulation by E2F-RB axis (Supplementary Fig. 5b). CRISPRa-mediated upregulation of E2F1 and E2F4 led to increased expression of several WNT ligands including WNT7A/B, WNT4 and WNT10A whereas CDK4/6 inhibition with PD-0332991 decreased WNT ligand in all three organoid lines cultured in 3D conditions (Supple-mentary Fig. 5c, f, g). CRISPR-mediated downregulation of pRb and RBL2 increased the expression of WNT ligands while RBL1 knockdown has a weaker effect (Supplementary Fig. 5d, h, i). In contrast, over-expression of RBs had the opposite effect on WNT ligands as indicated by their reduced expression (Supplementary Fig. 5e, j, k).

Pancreatic cancers have rare mutations in pRb[16,61] while pRb is frequently phosphorylated by CDKs in ki67-positive PDAC cells which indicates that this tumour suppressor is functionally inactivated in proliferating pancreatic cancer cells[62]. RBL2 is mainly known to act as a tumour suppressor regulating the cell cycle[1], and its mutations or loss of expression cause uncontrolled proliferation and tumorigenesis in various tissues[63]. Since little is known about the status of RBL2 in pancreatic cancers, we focused on RBL2 and characterised the muta-tion and expression status of RBL2 in PDACs. We found that RBL2 expression has a weakly negative correlation with WNT ligand expression based on the data in 183 PDAC patients ($p$ value = 0.03) (Supplementary Fig. 5j). Similarly to the pRb locus, the RBL2 mutations are generally not found in PDAC patient samples[64]. We also analysed the expression of WNT ligands from surgically removed PDACs from three patients by isolating CD133+/SSEA4+/EPCAM+ CSCs and com-paring them to CD133-/SSEA4-/EPCAM+ non-CSCs (Fig. 4f, g). WNT7A, WNT7B, WNT10A and WNT4 indicated higher expression in CSCs compared to non-CSCs. Furthermore, CDK4/6 inhibition with Palbo-ciclib led to a reduction in these WNT ligands in CSCs (Fig. 4g). Col-lectively, these data suggest that PDACs limit the activity of pRb/RBL2 mostly through inactivating phosphorylation events via CDKs.

CSCs are able to metastasise due to their increased invasive capacity mediated by EMT[65]. Gene expression analysis of EMT-inducing transcription factors SNAI1, SNAI2, ZEB1, ZEB2 and TWIST1 in pRB KD, RBL1 KD and RBL2 KD indicated their increased expression (Fig. 4h), while in contrast, pRb OE and RBL2 OE had a repressive effect on these EMT inducers (Fig. 4i). Next, we performed a wound healing assay to measure the paracrine effects on PDAC cell migration (Fig. 4j, k). These data indicated that conditioned media from cultures

with OE of pRb and OE RBL2 had a weaker effect on wound closure compared to control OE GFP cells. Similarly, WNT pathway inhibitor IWP2 reduced wound closure whereas purified WNT7A and WNT7B had an inducing effect (Fig. 4j, k). Since the WNT pathway also impacts cell proliferation and it is difficult to fully separate these effects on wound closure assay, we performed a transwell assay to measure cell invasiveness (Fig. 4l). Importantly, the conditioned media from OE of RBL2 OE, pRB OE cells and WNT pathway inhibition decreased the invasiveness of PDAC cells while WNT ligands increased invasiveness of PDAC cells (Fig. 4l). In agreement with these results, we found that in PDAC patient tumours the combined expression of WNT ligands (WNT7A, WNT7B, WNT4, WNT10A, WNT3, WNT10B, WNT11) has a positive correlation ($R = 0.41$, $p$ value = 1.8e-08) with the combined expression of key EMT-inducing transcription factors SNAI1, SNAI2, ZEB1, ZEB2 and TWIST1 (Fig. 4i). We also compared the transcriptional features of different tumour subsets identified from other studies[17] and found that the population of cells expressing WNTs and EMT-inducing transcription factors overlap with gene expression signatures of the basal subtype of PDACs and an EMT signature (Fig. 4n–p), which has been found to correlate with higher mortality, chemotherapy resis-tance and higher metastatic capacity. Collectively, WNT expression regulation by E2F-Rb axis impacts the invasive capacity of PDAC cells which is known to be an aggressive cancer type with highly metastatic characteristics.

## Regulation of WNT ligands by the RB-E2F pathway mediates the characteristics of pancreatic CSCs

To uncover the effect of RBL2 on WNT ligands in PDAC CSCs, we stu-died the paracrine effects on CSC self-renewal by tumour sphere assays (Fig. 5a). Conditioned media from pancreatic CSCs with WNT pathway inhibition by IWR and IWP2 led to lower CSC sphere formation, whereas supplementation of media with purified WNT7A/B and WNT4 ligands significantly increased CSC sphere formation. Conditioned media col-lected from cells ovexpressing pRb or RBL2 led to lower sphere num-bers compared to media from OE GFP control CSCs, while media from pRb KD and RBL2 KD had the opposite effect (Fig. 5a). The positive effect of RBL2 KD on sphere numbers was partially lost when the conditioned media from RBL2 KD cells was antibody-depleted from WNT ligands WNT7B, WNT7A, and WNT4. Furthermore, CSCs grown in conditioned media containing WNTs had larger spheroids than those grown in WNT7B, WNT7A, and WNT4 depleted media (Fig. 5b, c). We observed similar effects on cell proliferation in organoids following 7 days of growth (Supplementary Fig. 6a, b).

Since pancreatic CSCs are more recalcitrant to conventional chemotherapy reagents such as gemcitabine (Gem), we investigated the effect of RB-WNT regulatory circuitry on CSC chemoresistance. The conditioned media from cells treated with IWP2 was less efficient in supporting CSC sphere formation upon Gemcitabine treatment (Fig. 5d), indicating that WNT ligand secretion supports pancreatic CSC recalcitrance to this cytotoxic molecule. Conditioned media from cells with elevated expression of pRb or RBL2 was less efficient in supporting CSC chemoresistance than media collected from GFP expressing cells, while adding purified WNT ligands to the conditioned media increased CSC sphere numbers (Fig. 5f). Next, we analysed the relative expression of CSC markers on spheroids by using engineered L3.6pl cells where we had integrated a GFP sequence c-terminally in-frame into the endogenous locus of OCT4 that produces an endo-genously expressed OCT4-GFP fusion protein (Fig. 5e, g). Conditioned media from IWR and IWP2 treated cells led to a relative reduction of PDAC cells that co-express CSC markers CD133+/OCT4+/SSEA4+ (Fig. 5e, g). WNT ligands increased these CSC-marker expressing cells and conditioned media from pRb and RBL2 overexpressing cells showed a lower proportion of CSC-marker expressing cells compared to cells grown in GFP-conditioned media (Fig. 5g). At the same time, the WNT pathway inhibition with IWR and IWP2 reduced cell survival

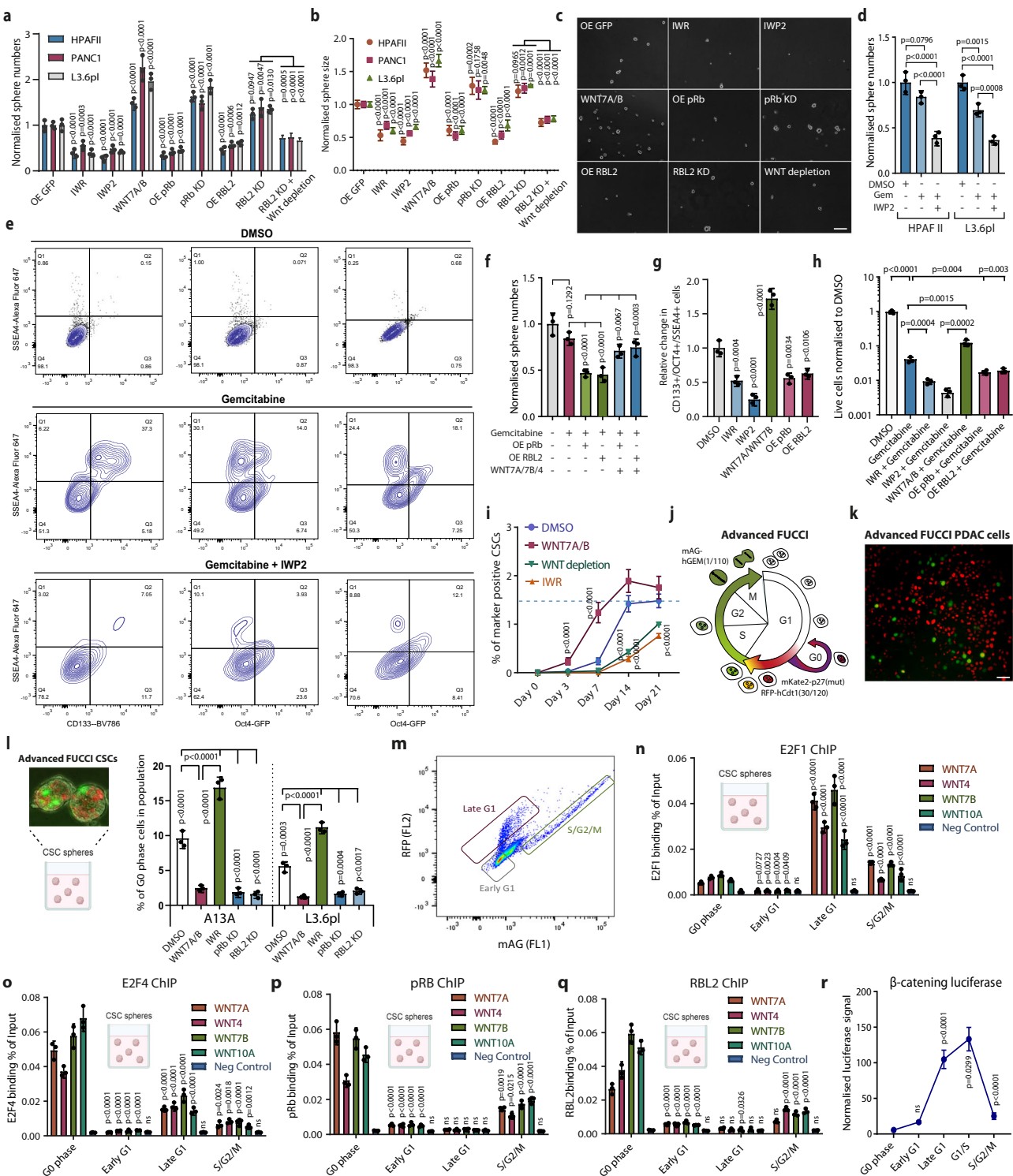

upon Gemcitabine treatment as did conditioned media from OE RBL2 cells compared to GFP cells, while purified WNT7A/B increased cell survival (Fig. 5h).

The differentiation and dedifferentiation of CSC form a dynamic balance that is mediated by signalling pathways and epigenetic plasticity of cells that regulate their cellular identity. Since RB-mediated repression of WNT ligands or WNT signalling inhibition by small molecule compounds decreased CSC self-renewal, we studied whether WNT could impact the re-emergence of CSC marker-expressing cells. To investigate this aspect, we sorted cells into CSC marker OCT4-GFP/CD133/SSEA4 negative population and analysed the re-emergence of

these CSC triple marker-expressing cells upon WNT signalling inhibition or WNT ligand treatment (Fig. 5i). In the steady-state cell culture, the population contains ~1.5% OCT4-GFP+/CD133+/SSEA4+ CSCs. Control cells treated with DMSO showed the beginning of the re-emergence of OCT4-GFP+/CD133+/SSEA4+ CSCs after 3 days and reached the steady-state level similar to the untreated and sorted cells after 14 days. These data suggest that WNT signalling promotes CSC dedifferentiation or increases cellular plasticity by possibly decreasing the epigenetic barriers that are necessary for the cells to dedifferentiate from non-stem cancer cells to CSCs and thus help maintain CSC characteristics.

**Fig. 5 | pRB and RBL2 regulate pancreatic CSC characteristics through non-cell-autonomous mechanisms involving WNT ligands.** RB family proteins control non-cell-autonomously the (**a**) self-renewal and proliferation (**b**) of CSCs at least partly through WNT ligands. **c** Bright-field images of tumour sphere assay showing the non-cell-autonomous effects of pRB and RBL2 overexpression (OE) or knockdown (KD) and WNT ligand depletion from the conditioned media. Scale bar 50 μm. **d** Inhibiting WNT ligand processing with IWP2 sensitises CSCs to Gemcitabine (Gem). **e** Flow cytometry analyses of CSC markers CD133, OCT4-GFP and SSEA4 on cells treated with conditioned media show sensitisation of PDAC CSCs to Gemcitabine by IWP2 treatment. **f** pRb and RBL2 overexpression decrease CSC resistance to Gemcitabine in a paracrine way whereas WNT7A/7B/4 ligands reverse this effect. **g** The relative change in triple positive CD133+/OCT4+/SSEA4+ cells treated with conditioned media from different treatments. **h** RBs and WNT signalling regulate the resistance of PDAC cells to gemcitabine. Data are presented as mean values ± SD. $N$ = 3 independent experiments. Statistical analysis in (**f**, **g**, **h**) was performed by 1-way ANOVA with Dunnett's multiple comparisons test. **i** WNT signalling in the conditioned media shifts the balance toward CSCs in the differentiation and dedifferentiation of CSCs and non-CSCs. **j** Graphic depiction of advanced FUCCI system. **k** Representative image of FUCCI-PDAC cells grown in 2D conditions. Scale bar 50 μm. **l** The percentage of CSCs in G0 phase is regulated by WNT signalling and conditioned media from RB-expressing cells. A representative image of FUCCI-PDAC cells grown in 3D sphere conditions is shown. **m** Dot blot of the RFP-hCdt1(30/120) and mAG-Geminin(1/110) FUCCI signals and gates marking the different cell cycle phases which were used for cell sorting. Cell cycle-dependent binding of E2F1 (**n**), E2F4 (**o**), (**p**) pRb, (**q**) RBL2 in CSCs to WNT ligand loci. Data are presented as mean values ± SD. Graphs depicted in (**a**, **b**, **d**, **i**, **l**, **n**, **o**, **p**, **q**) are $n$ = 3 independent experiments. Statistical analysis in (**a**, **b**, **d**, **i**, **l**, **n**, **o**, **p**, **q**) was performed by 2-way ANOVA with multiple comparisons with Tukey correction. **r** Cell cycle-dependent β-catenin dependent promoter-luciferase activity. Data are presented as mean values ± SD. $N$ = 3 independent experiments. Statistical analysis in r was performed by 1-way ANOVA with Tukey's multiple comparisons test. ns = not significant. Source data are provided as a Source Data file.

## E2Fs induce and RBs repress WNT ligand expression in a cell cycle-dependent manner in CSCs

We proceeded to investigate the cell cycle progression of PDAC cells in more detail with the fluorescent ubiquitination-based cell cycle indicator (FUCCI) system. We have previously used a dual-colour FUCCI system in hESCs to detect cells in G1, S and M-phases[66]. Recently, the presence of the cell cycle inhibitor p27 with distinct site-specific mutations has been shown to characterise G0 cells[67]. To study the effects of RBs and WNT signalling on CSC proliferation we developed an advanced three-colour FUCCI system (RFP-hCdt1(30/120)_mAG-hGEM(1/110)_mKate2-p27(mut)) by combining the truncated hCdt1, truncated hGeminin and p27K(-). We established PDAC cell lines by TALEN-mediated targeting of the three-colour FUCCI construct into the AAV1 locus for stable genomic integration. We verified the FUCCI system by flow cytometry analyses of the advanced FUCCI (RFP-hCdt1(30/120)_mAG-hGEM(1/110)_mKate2-p27(K-)) integrated into the AAVS1 locus in A13A PDAC CSCs show the signals of late G1 (hCdt+), G1/S transition (hCdt+/Geminin+), S/G2/M (Geminin+), and early G1 (hCdt-/Geminin-) (Supplementary Fig. 6c). The visualisation of hCdt and p27 shows the separation of hCdt+/p27- and hCdt+/p27+ cells (Supplementary Fig. 6d). Cells residing in the G0 phase are hCdt+/p27+ and they should be Ki67 negative, which is a marker for proliferating cells. The staining of FUCCI-CSCs for Ki67 shows a population of cells with very low Ki67, which marks G0 phase cells (Supplementary Fig. 6e). Gating of p27+ cells (highest signal for p27) shows that these cells are indeed negative for Ki67 signal (Supplementary Fig. 6f ), thus confirming that the advanced FUCCI is able to distinguish G0 cells in PDAC. Collectively, this FUCCI system enables distinguishing between the cells in early G1, late G1, early S, S/G2/M and G0 phases (Fig. 5j, k), which has not been possible earlier in live pancreatic cancer cells.

We first aimed to determine the cell cycle kinetics in FUCCI cells after RB knockdown and WNT inhibition. We treated the CSC-FUCCI spheres for 5 days with WNT7A/B or conditioned media collected from IWP2, pRb KD and RBL2 KD, and performed flow cytometry analysis with the FUCCI (Fig. 5l and Supplementary Fig. 6g–i). Gemcitabine treatment led to an increased fraction positive for both Cdt1-mRFP and p27k(-)-mKATE2 signals which marks G0-phase cells (Fig. 5l and Supplementary Fig. 6h). Since exposure to many chemotherapeutic agents typically results in G2/M checkpoint arrest or the induction of apoptosis upon extensive cytotoxic effects, these results indicate that Gemcitabine at the used concentrations leads mostly to cell death. Interestingly, media from IWP2-treated cells also increased G0-phase cells while WNT7A/B ligands and media from pRB KD or RBL2 KD cells led to a reduction in G0-phase cells (Fig. 5l and Supplementary Fig. 6h). The stimulation of proliferation by the WNT7A/B ligand also increased the susceptibility of cells to Gemcitabine-induced apoptosis (Supplementary Fig. 6i). This suggested that WNT activation promotes proliferation while reducing the capacity of cells to enter the G0 phase which is protective for the cells upon genotoxic insults such as Gemcitabine. Hence, the RB-E2F axis through WNT ligand regulation is involved in non-cell autonomous effects on CSC proliferation and escape from genotoxic insults by controlling the temporary dormancy or quiescent state of cancer cells by entering the G0 phase.

Since the E2F-RB cell cycle regulatory axis controls cell cycle progression we wanted to gain insight into the regulation of the G0 phase of CSCs. We sorted the FUCCI-CSC cells into different cell cycle phases (Fig. 5m), according to the expression of the FUCCI marker expression and performed E2F1/4 and pRb/RBL2 ChIP-qPCR of the WNT ligand loci (Fig. 5n–q). Importantly, we observed a cell cycle-dependent binding of E2F1/4 (Fig. 5n, o) and pRb/RBL2 (Fig. 5p, q) on WNT loci. E2F1 was most enriched in the late G1 phase (Fig. 4n) while E2F4 was particularly enriched in G0 phase cells (Fig. 5o). pRb and RBL2 were most enriched in G0 phase and more modestly in S/G2/M. Of note, pRb and RBL2 were not binding to WNT in the late G1 phase whereas E2F1 and E2F4 showed binding to WNT loci in late G1 phase (Fig. 5n–q). The activity of β-catenin dependent promoter-luciferase activity showed the lowest activity in G0 phase and gradually increased with the highest activity in the late G1 and G1/S transition (Fig. 5r). These results indicate that the E2F-RB axis regulates the expression of WNT ligands in a cell cycle-dependent manner in pancreatic CSCs.

Altogether, our data indicate that pRb and RBL2 expression in PDAC cells has a negative effect on CSC characteristics via WNT ligand-mediated non-cell autonomous effects emphasising the importance of RB tumour suppressors as regulators of autocrine and paracrine signalling that shapes the stem cell-like characteristics of cancer cells and cross-talks with the surrounding cells through the tumour microenvironment.

## Paracrine WNT signalling mediates CSC crosstalk with several stromal fibroblast subtypes

The annotation of ligand-receptor interactions indicated the presence of CSC-specific signalling with other cell types found in patient PDACs (Fig. 6a). To model the crosstalk between CSCs and other cancer cell subpopulations in primary tumours, we paired the expression of the factors identified in our secretome studies, and their receptors on each of the cell types present in the PDAC patient tumours. We found that the ligands identified in the CSC secretome also mediate putative paracrine signalling between CSCs and other cell types with the corresponding receptors such as fibroblasts, T-cells, macrophages and neural/nerve cells as suggested by their expression in PDAC tumours isolated from patients (Fig. 6a). Among the implied ligand-receptor pairs mediating the crosstalk are particularly many between CSCs and fibroblasts that include WNT ligands and receptors. Among the ligand-receptor pairs mediating the crosstalk are FZD6-LRP5/6-WNT7B, FZD5/6-LRP5/6-WNT2, FZD5-LRP5/6-WNT5A (Fig. 6b). We further aimed to determine what subtype of CAF is most responsible for WNT crosstalk

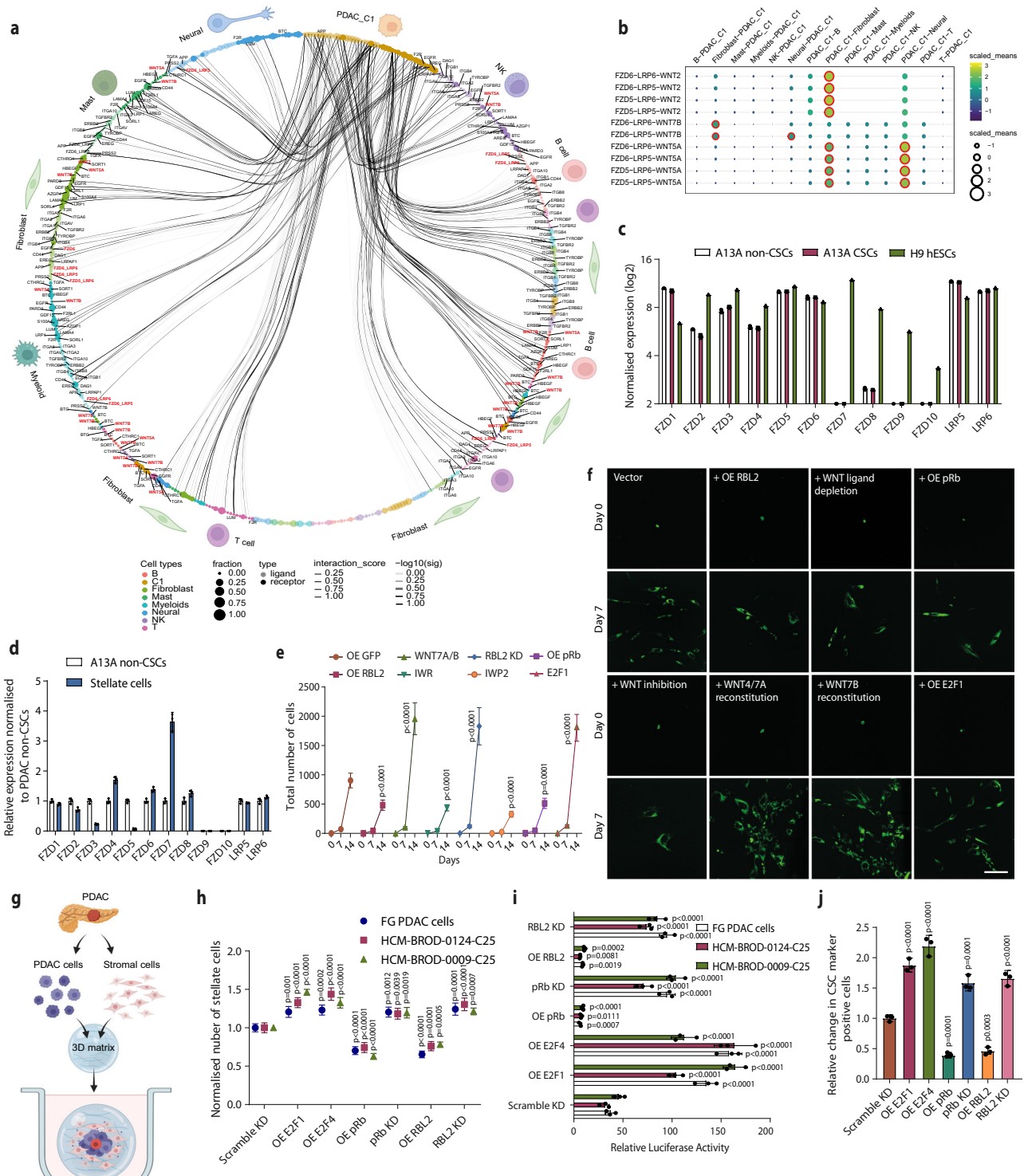

**Fig. 6 | RBs decrease EMT and invasiveness of CSCs. a** Ligand–receptor pair mediating signalling between CSCs and non-CSCs. **b** Paracrine signalling between CSCs (C1) and other cell types in PDAC patient primary tumours. **c** Expression of FZD receptors in A13A non-CSCs, A13A CSCs and H9 hESCs based on RNA-seq data. **d** Relative expression of FZD receptors in A13A non-CSCs compared to stellate cells based on qPCR. **e, f** Conditioned media from OE RB cells leads to slower CAF proliferation while conditioned media from RB KD cells increases CAF proliferation. Scale bar 50 μm. **g** Schematic depiction of 3D co-cultures of PDAC cells and stromal fibroblasts in a matrix. **h** Effects of OE E2Fs, OE RBs and RB KDs in PDAC cells on stellate cell numbers in co-culture conditions in patient-derived organoids. **i** Effects

of OE E2Fs, OE RBs and RB KDs in PDAC cells on β-catenin dependent promoter-luciferase activity in co-culture in patient-derived organoids. Data are presented as mean values ± SD. Graphs depicted in (**c, d, e, h, i**) are *n* = 3 independent experiments. Statistical analysis in (**c, d, e, h, i**) was performed by 2-way ANOVA with multiple comparisons with Tukey correction. **j** Effects of OE E2Fs, OE RBs, and RB KDs on the relative number of CSCs in co-culture conditions in patient-derived organoids. Data are presented as mean values ± SD. *N* = 3 independent experiments. Statistical analysis in (**j**) was performed by 1-way ANOVA with Dunnett's multiple comparisons test. Source data are provided as a Source Data file.

between CSCs and fibroblasts. Detailed scRNA-seq analysis of PDAC patients indicated the presence of different types of CAFs such as myCAFs, apCAFs, iCAFs, pericytes, epithelial-like, chondrocyte-like, peri-islet Schwann cells, and proliferating CAFs (Supplementary Fig. 6j, k), which have been identified previously in PDACs[68]. The ligand-receptor pairs mediating the putative crosstalk between CSCs and the different types of CAFs revealed for instance, the possible crosstalk between CSCs and proliferating CAFs mediated by WNT7B-FZD5/6-LRP5/6, and WNT7B-FZD2/8-LRP5/6 between CSCs and iCAFs as well as myCAFs (Supplementary Fig. 6l). Therefore, CSCs are likely to be in functional crosstalk with several cell types in PDACs, including several fibroblast subtypes, which contribute to the characteristics and the CSC niche in the tumour.

Fibroblasts are abundant in PDAC tumours due to the highly desmoplastic and fibrosis-promoting conditions in the transforming pancreatic tissue[69]. Due to high fibrosis, the dense stroma provides a physical barrier for therapeutics to reach cancer cells[70]. Stromal cells also form an environment that can support cancer cell proliferation and CSC characteristics[71]. Hence, the formation of the dense desmoplastic stroma is important not only for understanding the cell signalling forming the CSC niche but also for therapeutic accessibility[69–71]. There are 10 (FZD1-10) Frizzled receptors and Lrp5/6 co-receptors that function as WNT receptors. We have confirmed the expression of WNT receptors on A13A PDAC non-CSCs and CSCs by RNA-sequencing (Fig. 6c). These data show that A13A cells express FZD5, FZD6 and FZD1 most highly among FZD receptors, and they also express LRP5/6 co-receptors. We also investigated FZD and Lrp5/6 expression in stellate cells by comparing the expression of FZDs in PDAC non-CSCs and stellate cells by qPCR (Fig. 6d). Stellate cells express FZD1-8 at varying levels. Considering the levels in PDAC non-CSCs, the expression in stellate cells is highest for FZD1, FZD4 and FZD6. There is also moderate expression of FZD2 and FZD3 whereas the expression of FZD5 and FZD7 is low, and there is no expression of FZD8 and FZD9. The co-repressors LRP5 and LRP6 are highly expressed in stellate cells similarly to PDAC cells.

Next, we examined the effects of pRb and RBL2 status and the possible paracrine effects on the proliferation of cancer-associated fibroblasts labelled with GFP (Fig. 6e, f). The conditioned media from RBL2 overexpressing cells had a lower capacity to promote fibroblast proliferation, similarly to WNT ligand depletion from conditioned media, while purified WNT4/WNT7A or WNT7B reconstitution increased fibroblast proliferation. We also tested pRb effects and found that it showed a similar inhibitory effect on paracrine signalling as RBL2, whereas E2F1 OE conditioned media promoted fibroblast proliferation (Fig. 6e, f). We decided to further investigate the mechanisms of WNT ligands stimulating proliferation of fibroblasts. Based on data that has attributed cellular functions to fibroblast proliferation in other tissues, we investigated a panel of ten central regulators of cell proliferation (c-Myc, Cyclin D1, Cyclin E1, FGF2, FGFR2, EGFR, EGF, FGF1, FGFR1 and VEGFA) in pancreatic cancer-associated fibroblasts (CAFs). We found that stimulation of pancreatic CAFs with WNT7A/7B for 24 h increased the expression of c-Myc, Cyclin D1, Cyclin E1, FGF2, FGFR2, EGFR, EGF, FGF1, FGFR1 and VEGFA, whereas IWR treatment reduced the expression of these genes (Supplementary Fig. 7a). These results suggested that β-catenin might bind directly to the regulatory regions of these loci. Since these genes represent central regulators of cell proliferation, we analysed previously published β-catenin ChIP-seq, and ChIP-seq of its transcriptional coregulators TCF7L2 and LEF1. These revealed binding peaks of β-catenin ChIP-seq in the proximity of all ten genes, and the binding regions overlapped with either TCF7L2 and/or LEF1 (Supplementary Fig. 7b). Next, we performed β-catenin ChIP-qPCR on the identified regions in pancreatic CAFs and uncovered enrichment of β-catenin protein binding at these regions near the ten genes (Supplementary Fig. 7c). To investigate, if these these β-catenin target genes mediate WNT/β-catenin effects in

pancreatic CAFs, we performed cell proliferation assays. Treating CAFs with WNT7A/7B in combination with FGFR inhibitor Pemigatinib and EGFR inhibitor Erlotinib, slowed the proliferation of CAFs compared to the cells treated with only WNT7A/7B (Supplementary Fig. 7d). In contrast, addition of FGF2 and EGF increased CAF proliferation compared to DMSO treatment (Supplementary Fig. 7d). To directly link the paracrine effects of E2F-driven WNT expression/secretion on stromal cells, we performed a CRISPR-KRAB knockdown of WNT7A and WNT7B (Supplementary Fig. 7e) and overexpressed either E2F4 or E2F1 in FG PDAC cells. Thereafter, we cocultured these FG cells with stellate cells expressing the β-catenin-dependent promoter-luciferase construct to measure WNT/β-catenin signalling in stellate cells. Overexpression of E2F4 or E2F1 together with WNT7A/7B KD in PDAC cells reduced β-catenin-dependent promoter-luciferase activity compared to E2F OE cell cocanluring (Supplementary Fig. 7f), indicating that WNT7A/7B mediate the E2F-driven WNT effects on stromal stellate cells. This also impacts the proliferation of the stromal cells, since WNT7A/7B KD in E2F4 OE or E2F1 OE PDAC cells slowed stellate cell proliferation upon coculturing (Supplementary Fig. 7g, h). We also investigated the corresponding WNT ligand receptors FZD1 and FZD6, which we found to be particularly highly expressed in stellate cells over the other FZD receptors. We performed a knockdown of FZD1 and FZD6 in stellate cells and found that this leads to reduced β-catenin-dependent promoter-luciferase activating in these cells upon coculturing with FG PDAC cells (Supplementary Fig. 7i, j). Furthermore, the FZD1 KD and FZD6 KD cause slower proliferation of stellate cells upon coculturing with FG PDAC cells (Supplementary Fig. 7k). Collectively, these data indicate that WNT7A/7B ligands and WNT receptors FZD1 and FZD6 mediate the paracrine signalling between pancreatic cancer cells and stellate cells in the tumour.

To more closely mimic the in vivo tumour tissue with cell-cell communication, we established a 3D coculture system by combining PDAC cells and PDAC-derived stellate cells in a matrix that resembles the physiological condition in pancreatic tumours (Fig. 6g). In this coculture system, we found that OE of E2F1 and E2F4 as well as pRb KD and RBL2 KD in PDAC cells lead to increased proliferation of stellate cells, whereas OE of pRb and OE RBL2 in PDAC cells decrease the number of stellate cells/fibroblasts (Fig. 6h). These treatment conditions also match β-catenin-dependent transcription as shown by increased luciferase activity in stellate cells (Fig. 6i). E2F1/4 OE and pRb/RBL2 KD increase the proportion of OCT4+/CD133+/SSEA4+ CSCs in this coculture while pRb/RBL2 OE reduce the proportion of OCT4+/CD133+/SSEA4+ CSCs (Fig. 6j). We also checked whether WNT-dependent signalling increases when pancreatic cancer cells and fibroblasts are co-cultured. We transfected FG PDAC cells with M50 Super 8x TOPFlash and Renilla luciferase as an internal control, and placed the cells in the co-culture system together with fibroblasts. Co-culturing of cancer cells with fibroblasts for 5 days resulted in an increase in M50 Super 8x TOPFlash β-catenin dependent WNT signalling in cancer cells compared to culturing cancer cells without fibroblasts (Supplementary Fig. 8a). These results suggest that the RB-E2F-WNT regulatory circuitry has a paracrine effect on fibrosis in PDACs.

Next, we investigated whether chemotherapy increases WNT signalling in CSCs, and chemotherapy increase WNT signalling in CSC in particular, and whether WNT signalling remains at high levels if resistance to chemotherapy becomes constant over the long term. We transfected A13A PDAC cells with M50 Super 8x TOPFlash and Renilla luciferase as an internal control. Heterogeneous PDAC population was treated with Gemcitabine, FOLFIRINOX and Nab-paclitaxel for 1, 3, 5 and 10 days and M50 Super 8x TOPFlash and Renilla luciferase signals measured at each time point (Supplementary Fig. 8b). We observed an increase in the normalised WNT-dependent luciferase signalling at day 3 and day 5, whereas the prolonged treatment until day 10 resulted in a reduction of WNT-dependent luciferase signalling. These results allow us to make several conclusions by putting them in the context of our

other experimental results described in the manuscript: firstly, since we did not observe an increase in WNT-dependent signalling after day 1 it suggests that WNT signalling increase is not a general stress response of the cells. Rather, the increase in the WNT signalling is caused by the selective survival of CSCs over non-CSCs, and CSCs have a higher WNT signalling as we had found before (Fig. 1). We have shown that chemotherapy treatment selectively kills non-CSCs while CSCs are more resistant to chemotherapy reagents than non-CSCs. Secondly, when chemotherapy becomes constant over the longer term (day 10 in our case since all non-CSCs are killed by that time), the WNT signalling is reduced. The reason for the decrease of WNT signalling at day 10 is that CSCs were initially capable of self-renewal by progressing through the cell cycle but the constant chemotherapy treatment causes the accumulation of CSCs in the G0 phase (Supplementary Fig. 4d). We have shown that in G0 phase, WNT signalling activity is reduced via E2F-pRb/RBL2 mediated repression of WNT ligands (Fig. 5s), thus suggesting a mechanism for this observation.

We also checked whether CSCs decrease faster when enriched and WNT signalling is forcibly weakened. For this, we sorted cells to CD133+/OCT4+/SSEA4+ CSCs by FACS and placed them in CSC sphere culture conditions. We measured the number of cells positive for CD133+/OCT4+/SSEA4+CSC markers in A13A and L3.6pl lines upon WNT signalling inhibition with IWR and IWP2. Indeed, CSCs decreased when WNT signalling was inhibited by IWR and IWP2 for 3 days and further decreased at 7 days of treatment compared to enriched CSCs that maintained CSC marker expression better in both A13A and L3.6pl lines (Supplementary Fig. 8c).

Collectively, the ligand-receptor crosstalk analyses indicated extensive bidirectional paracrine signalling between CSCs and other cell types in PDAC tumours, particularly fibroblasts, which included WNT ligands with their receptors. Therefore, we investigated the function of WNT signalling in more detail between CSCs and fibroblasts.

## GCN5 inhibition represses E2F1/4-mediated induction of WNT ligands and abolishes pancreatic CSCs

Since WNT ligands induced CSC characteristics, we performed a small molecule compound screening in PDAC cells with the aim to identify epigenetic regulators that could eliminate CSCs with possible specificity over non-CSCs by measuring the expression of CSC markers OCT4, CD133/PROM1 and SSEA4 in a heterogeneous PDAC cell population (Fig. 7a). Using this strategy we have previously identified chromatin binding in KAT2A and the BAF complex that regulate CSC properties[35,72]. In the screening we used a library of 142 small molecule tool compounds with annotated specificity and biological activity to a broad range of epigenetic modulators (Supplementary Fig. 8d; Supplementary Data). Among these compounds were bromodomain inhibitors, HDAC inhibitors, histone methyltransferase (HMT) inhibitors or histone acetyl transferase (HAT) inhibitors, methyl lysine binding inhibitors, arginine methyltransferases and lysine demethylases among others (Supplementary Fig. 8d). For the screening, we plated the PDAC cells into 96-well plates and treated the cells for 5 days with each individual small molecule compound, and then measured expression of CSC marker OCT4, CD133/PROM1 and SSEA4, cell numbers, and cell death by DAPI signal via flow cytometry (Fig. 7a). Hence the screening allows for the detection of differential effects on the subpopulations of non-CSCs and CSCs.

From these analyses, we identified GCN5 inhibitors GSK4027[73] and L-Moses[74] as effective inhibitors that reduced the percentage of cells that express the CSC markers in the heterogeneous cell population, while the corresponding inactive negative control compounds GSK4028 and D-Moses did not have this effect on the percentages of CSC marker double-positive and OCT4+/CD133+/SSEA4+ positive cells (Fig. 7b, c and Supplementary Fig. 8e). Both compounds are potent and selective inhibitors of the bromodomain of GCN5/PCAF, domains that

are found in lysine acetyltransferases KAT2A and KAT2B, and that bind to acetylated lysine residues in chromatin. GCN5 is the shared catalytic subunit of the ATAC and SAGA complexes involved in gene transcription[75–77]. Therefore, the loss of CSC marker-expressing cells compared to non-CSCs upon GCN5/PCAF inhibition could indicate the possible cooperation of GCN5 with E2Fs in inducing WNT ligands in CSCs. To test this hypothesis, we performed co-immunoprecipitation of E2F1 and E2F4, which indicated binding to pRb and RBL2, as well as to GCN5 in CSCs (Fig. 7d, e). This indicated that E2F1 and E2F4 form a nuclear complex and cooperate with the GCN5 histone acetyl-transferase. Next, we analysed the effect of GCN5 bromodomain inhibition on the self-renewal capacity of CSCs by tumour sphere assays. GCN5 inhibition with GSK4027 and L-Moses reduced CSC sphere formation and sphere size while the negative control compounds GSK4028 and D-Moses did not reduce sphere formation compared to DMSO treatment (Fig. 7f–h). GCN5 inhibition with GSK4027 and L-Moses also reduced PDAC organoid formation whereas the control compounds GSK4028 and D-Moses did not have such effects (Supplementary Fig. 8f). Next, we performed genetic depletion of GCN5 to confirm the specificity of the pharmacological inhibitors of GCN5 by CRISPR-CAS9-KRAB mediated knockdown and investigated CSC sphere formation in three PDAC cell lines (Fig. 6i). GCN5 KD led to reduced CSC sphere numbers, indicating an effect of CSC self-renewal capacity as seen with GCN5 inhibitors.

As E2Fs induce WNT ligand expression as DNA-binding transcription factors and form a complex with GCN5, while RBs repress WNT expression, we hypothesised that E2F1/4 could recruit GCN5 to WNT loci and the binding of RBs to WNT loci could compete with GCN5 binding. To test this, we performed GCN5 ChIP-qPCR in E2F1 KD and E2F4 KD (Fig. 7j), which showed reduced binding of GCN5 to WNT loci in E2F1 KD and E2F4 KD compared to scramble KD in CSCs. These results indicate that E2F1 and E2F4 are mediating the binding of GCN5 to WNT loci in PDAC CSCs. To gain insight into how RBs could impact the function of GCN5, we performed GCN5 ChIP-qPCR upon pRb and RBL2 overexpression to identify the binding to WNT loci (Fig. 7k). GCN5 showed a reduced binding to WNT loci upon pRb and RBL2 overexpression. In order to find out if GCN5 inhibition can impact the WNT/β-catenin pathway activation, we performed the promoter luciferase assay with M50 Super 8x TOPFlash which showed reduced β-catenin-dependent promoter activation in response to GCN5 bromodomain inhibition (Supplementary Fig. 8g), indicating reduced WNT ligand production that leads to reduced β-catenin activation. Since GCN5/PCAF is the catalytic subunit of the histone acetyltransferase complex that can regulate H3K9 acetylation[78], we investigated the levels of H3K9ac upon GCN5 inhibition in CSCs by performing ChIP-qPCR on WNT loci (Fig. 7l). GCN5 bromodomain inhibition with GSK4027 and L-Moses led to a strong reduction of H3K9ac on WNT7A, WNT7B, WNT4 and WNT10A loci. Furthermore, overexpression of pRb and RBL2 reduced H3K9ac on WNT loci, supporting the hypothesis that RBs reduce GCN5 recruitment to these loci (Fig. 7l). GCN5 inhibition also moderately reduced H3K4me3 although these effects were weaker than the reduction of the H3K9ac signal on WNT loci (Supplementary Fig. 8h). Of note, the reduction of H3K4me3 was stronger upon overexpression of RBL2 and pRb (Supplementary Fig. 8h), indicating that the RBs are likely to mediate additional mechanisms such as the recruitment of repressive epigenetic regulatory complexes. The knockdown of GCN5 also decreased H3K9ac abundance on the promoters of these WNT loci (Supplementary Fig. 8i), and expression of WNT7A, WNT7B, WNT10A and WNT4 expression (Supplementary Fig. 8j).

Next, we studied the effects of GCN5 bromodomain inhibition in PDAC patient-derived freshly isolated primary tumours (Fig. 7m). We performed multiplexed flow cytometry analyses on PDAC tumours derived from three PDAC patients by obtaining biopsies from fresh surgically removed tumour tissue. Surgically removed PDAC patient

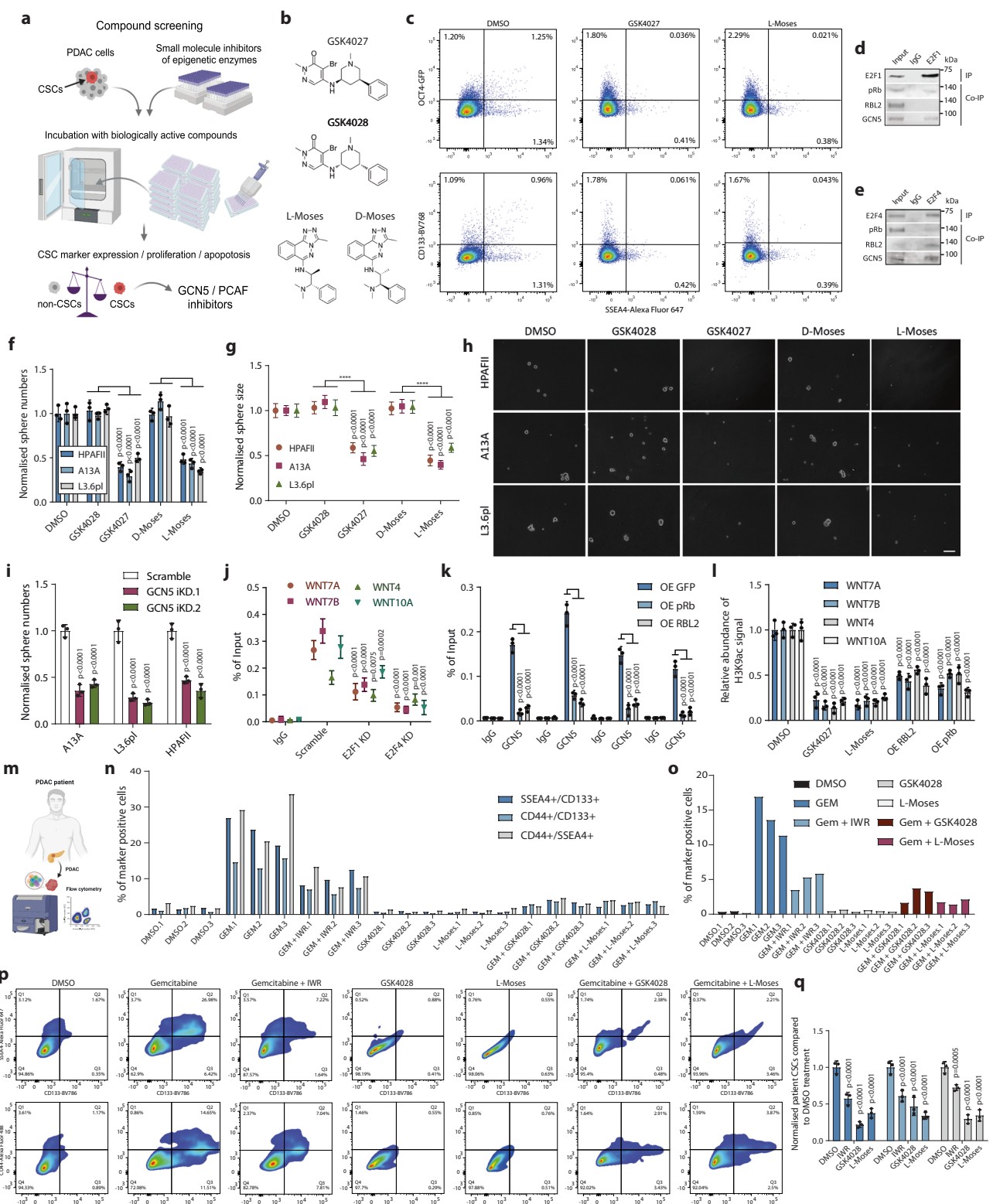

tumour samples from consenting patients were confirmed by histologists before further processing for 1) confirming the enrichment of CSC marker CD133, CD44 and SSEA4 expression upon Gemcitabine chemotherapy reagent treatment; 2) effects of WNT/β-catenin inhibition with IWR and its effect on CSCs; 3) the effects of GCN5 inhibitors GSK4028 and L-Moses on CSCs as single treatment and in combination with currently used chemotherapy reagent Gemcitabine. The tumour samples were dissociated into single cells, and CD45 capture MicroBeads were used for removing hematopoietic cells, which

allowed us to focus our analyses on cancer cells that were CD45 negative. This also reduces false positive signals that could come from non-cancer cells expressing the markers. Firstly, we investigated the enrichment of CSC markers CD133, CD44 and SSEA4 upon Gemcitabine chemotherapy reagent treatment in tumours from three PDAC patients. The treatment of patient-derived PDAC cells with 0.5 μM Gemcitabine for 3 days resulted in extensive cell death, whereas cells that survived the treatment were enriched for the expression of single CD133, CD44 and SSEA4 markers (Supplementary Fig. 8k–m), double

**Fig. 7 | Compound screening identifies GCN5/PCAF as a therapeutic target for eliminating CSCs. a** Schematic depiction of the screening process that identified GCN5/PCAF inhibitors as effective reducers of CSCs. **b** Chemical structures of GCN5 inhibitor GSK4027 and its negative control GSK4028, and L-Moses with its corresponding negative inhibitor D-Moses. **c** Dot plots of CSC markers in DMSO, GSK4027 and L-Moses treated samples. **d** Immunoprecipitation of E2F1 shows its binding to pRb and GCN5. **e** Immunoprecipitation of E2F4 shows its binding to pRb, RBL2 and GCN5. **f–h** GCN5 inhibition reduces the self-renewal of CSCs and sphere size increase assessed by tumour sphere assays. **h** Representative images of spheres in each condition. **i** GCN5 KD reduces the self-renewal capacity of CSCs in three PDAC cell lines. **j** E2F1 KD and E2F4 KD reduce GCN5 binding to WNT loci analysed by ChIP-qPCR in CSCs. **k** GCN5 binding to WNT loci and its binding is reduced by pRb and RBL2 as analysed by ChIP-qPCR in CSCs. **l** GCN5 mediates H3K9ac on WNT loci and this is reduced by pRb and RBL2. **m** Schematic depiction of using primary tumour sample from PDAC patients. **n–p** GCN5 inhibition reduces double and triple CSC marker positive CSCs from primary tumour sample from PDAC patients. **q** The abundance of CSCs is decreased by GCN5 inhibitors in primary tumours from patients. Data are presented as mean values ± SD. Graphs depicted in (**f, g, i, j, k, l, q**) are $n = 3$ independent experiments. Statistical analysis in (**f, g, i, j, k, l, q**) was performed by 2-way ANOVA with multiple comparisons with Tukey correction. Source data are provided as a Source Data file.

positive CD133+/CD44+, CD133/SSEA4+ and CD44+/SSEA4+ cells (Fig. 7n–p), and CD133+/CD44+/SSEA4+ cells (Fig. 7o, p). This indicates the clinically used chemotherapy reagent Gemcitabine is able to eliminate non-CSCs but does not reduce CSCs which are selectively surviving this treatment. Since WNT signalling inhibition reduced CSC self-renewal and chemoresistance of PDAC cell line-derived CSCs, we used co-treated patient cells with Gemcitabine and WNT/β-catenin signalling inhibitor IWR. The co-treatment with IWR reduced the total live cells and the number of cells expressing CSC markers, suggesting that IWR decreases the chemoresistance of CSCs to Gemcitabine (Fig. 7n–p). Lastly, we isolated the CSC subpopulation from freshly isolated patient primary PDAC tumours by using FACS-sorting of triple positive CD133+/CD44+/SSEA4+ cells and treated these with GCN5 inhibitors for 5 days followed by measuring live cell numbers. GCN5 inhibition decreased the number of live patient-derived CSCs while WNT inhibition with IWR also had moderate effects on live CSCs compared to DMSO-treated cells (Fig. 7q).

Collectively, we have shown that E2Fs form a complex with GCN5 that regulates H3K9ac on WNT loci and supports WNT ligand expression, whereas RBs counteract this transcriptionally activating mechanism. The utilisation of a small molecule inhibitor for GCN5 bromodomain inhibition could serve as a potential candidate for targeted therapeutics for selectively eliminating pancreatic CSCs (Supplementary Fig. 8n).

## Alterations in the RBL2-E2F axis and elevated expression of the WNT pathway are found in malignant subtype of breast cancers

Next, we hypothesised that our identified paracrine mechanisms mediated by E2F-RB axis might be deregulated also in other cancers besides PDAC due to RB loss, thus making our findings relevant beyond PDACs. Interestingly, WNT signalling is particularly important in the developing mammary gland and is among the self-renewal factors for mammary stem cells in basal as well as luminal lineages of the mammary epithelium[79,80]. Furthermore, since breast cancers have been genetically characterised more extensively than other human cancers, we decided to explore the potential interconnections of the RB-E2F cell cycle regulatory axis and WNT pathway in mammary tissue and breast tumours.

Analysis of 112 normal mammary tissue samples revealed that there is a positive correlation of WNT ligand expression with E2F4 ($R = 0.56$, $p$ value = 9.5e-11), E2F1 ($R = 0.38$, $p$ value = 4.2e-05), CDK4 ($R = 0.51$, $p$ value = 9.6e-09) and CDK2 ($R = 0.46$, $p$ value = 2.6e-07) (Supplementary Fig. 9a), including with WNT3A, WNT4, WNT7B and WNT10A (Supplementary Fig. 9b), indicating a possible mechanistic circuitry between E2Fs and WNT ligands during physiological mammary tissue maintenance and homoeostasis. A positive correlation was also present between E2F1/4 and WNT ligands in BRCA (Supplementary Fig. 9c). Among the WNT ligands, elevated expression of WNT7B, WNT9A, WNT3A in top 25% expressing versus bottom 25% expressing all BRCA patients combined correlate significantly with lower patient survival (Supplementary Fig. 9d, e). BRCA subtype stratification in addition to the compiled total BRCA Kaplan-Myer plots provides further insight. Interestingly, top 50% expressing versus the bottom 50% expressing WNT7A patients suggests a correlation trend with higher

mortality in HER2+ non-luminal subtype BRCAs ($p = 0.11$), whereas high WNT3A expression suggests a correlation trend with higher mortality in Luminal B subtype BRCAs ($p = 0.059$), and WNT9A in Luminal A subtype BRCAs ($p = 0.072$) (Supplementary Fig. 9d). Therefore, different WNT ligands may have more pronounced roles in specific BRCA subtypes, altogether suggesting that WNT ligand expression regulation by the RB-E2F axis has an important function on disease progression of BRCA patients.

Human breast cancer is commonly classified into five subgroups according to their genomic and transcriptomic architecture[81,82], while this clustering has been refined further into ten distinct breast cancer subgroups according to their molecular signatures[18]. To investigate the function of the RB-E2F axis in breast cancer, we took advantage of the METABRIC database via cBioPortal, which contains information on copy number aberrations (CNAs) and gene expression from 2000 primary breast tumours. Data mining revealed that RBL2 is more commonly gaining insertions and deletions in breast cancer than RBL1 and pRb, indicating that 275 out of 997 breast tumours had aberrations in the RBL2 locus whereas the corresponding numbers for RBL1 and pRb were 165 and 249, respectively (Supplementary Fig. 9f). Of this total of 275 aberrations in the *RBL2* locus, 231 were heterozygous deletions, three homozygous deletions, 35 nucleotide insertions and 6 with partial sequence amplifications. Interestingly, RBL2 CNAs are most frequent in luminal breast cancers (PAM50 gene-centric frequencies of somatic aberrations via gene loss for LumA = 0.592275 and LumB=0.492537) but are also found in the basal subgroup (Basal = 0.262712; Supplementary Fig. 9g), while RBL2 CNAs are frequent in breast cancer subgroups with a poor disease prognosis such as Cluster 2 (gene-centric frequencies of somatic aberrations via gene loss for RBL2 = 0.533333 vs pRb = 0.4 vs RBL1 = 0.022222; Supplementary Fig. 9g, h). Furthermore, the RBL2-regulatory axis seems to be a primary target for genomic aberrations (Supplementary Fig. 9i) among high-mortality subgroups[18]. Mutations were commonly found in genes known to directly control RBL2/pRb activity during G1/S phase transition, such as Cyclin D1, E2Fs, and CDK4 (average CNA frequency for G1/S regulators in Cluster 2 = 0.601; Supplementary Fig. 9j), all of which are known to be associated with aggressive breast tumours and among the most frequent driver genes[18,81]. Gene expression analysis obtained from the METABRIC database suggests that aggressive breast tumours also display an increase in expression not only for Cyclin D1 and E2F, but also for the WNT signalling pathway (relative enrichment of WNT pathway=1.75; Supplementary Fig. 9k).

Altogether, these results imply that breast cancers, particularly the aggressive breast cancer subgroup, display frequent disruption in the RBL2-tumour suppressor pathway (Cyclin D, CDK4/6, E2F, pRb, and RBL2), which regulates G1/S phase transition but also exhibit an increase in WNT signalling components.

## Tumour suppression by pRb and RBL2 in breast cancer involves paracrine inhibition of WNT ligands that repress stemness characteristics similar to pancreatic CSCs

The presence of RBL2 loss of function in aggressive breast cancer subgroups and pancreatic cancer with the associated increase in WNT

pathway components prompted us to test if RBL2 could regulate the activity of the WNT pathway in human breast and pancreatic tumours. We first identified human breast cancer lines with mutations in the RBL2 gene likely to decrease the protein expression using the Catalogue of Somatic Mutations in Cancer (COSMIC) database. Several basal B type breast cancers, such as HCC1500[82] and Cal-51[83], harbour RBL2 loss of heterozygosity (LOH) mutations induced by a frameshift in the gene product and have reduced expression of RBL2 protein (Supplementary Fig. 9l). We then analysed the expression of WNT ligands (WNT7B, WNT9A, WNT3A, WNT4) in RBL2-LOH cancer cell lines (HCC1500 and Cal-51) and in RBL2-wildtype cancers (MCF7 and T47D). These analyses revealed that RBL2-deficient cancer lines express a significantly higher level of these growth factors while overexpression of RBL2 in RBL2-deficient cancer lines reduced the expression of WNT ligands (Supplementary Fig. 10a). MCF-7 are more luminal than a basal subtype (such as HCC1500 and Cal-51), therefore there are multiple additional biological differences besides RBL2 expression that might explain the differences in WNT ligand expression. Nevertheless, this increase in gene expression also correlated with an increase in β-catenin-dependent transcriptional activity (Supplementary Fig. 10b), and with an augmented nuclear localisation of β-catenin (Supplementary Fig. 10c). pRB OE also led to the reduction of WNT ligand expression (Supplementary Fig. 10d) while pRb KD and RBL1 KD led to increased WNT ligand expression (Supplementary Fig. 10e). These results indicate a similar function of RBs in WNT expression regulation as observed in PDAC.

Next, we wished to learn whether RBL2 could control the transcription of WNT growth factors in breast cancer cells. Computational analysis of published E2F1 ChIP-seq data from breast cancer cells[46–48] and E2F4 ChIP-seq in lymphoblastoid cells[49] revealed that multiple WNT loci such as WNT7B, WNT3A, WNT9A and other WNT loci are bound by E2F1 and/or E2F4 (Fig. 8a and Supplementary Fig. 3f, g) in the proximal promoter in the genomic region within 5 kb upstream of TSS to 3 kb downstream of the gene, indicating the WNT loci are directly regulated by E2Fs not only in breast cancer but also in lymphoblastoid cells. The results prompted us to further investigate the functional consequences of pRb/RB1/RBL2 inactivation during tumorigenesis and to determine if RB status in breast cancer cells could alter their stem cell-like characteristics via the previously described paracrine mechanisms. CSCs have elevated therapeutic resistance, high metastatic capacity and the ability to give rise to new tumours in their entire heterogeneity, and hence they are particularly relevant for the aggressiveness and drug resistance of BRCA[84–86]. We focused on the E2F-regulated WNT ligands which had the most significant correlation with BRCA patient mortality (WNT7B, WNT9A, WNT3A) and investigated E2F and RB binding in BRCA CSCs of the frame-shift mutated RBL2 cells. For this, we performed RBL2 and E2F1 ChIP in RBL2-LOH cancer CSCs. E2F1 was found on the WNT7B, WNT9A, WNT3A and WNT4 loci, while RBL2 was absent (Fig. 8b). In turn, overexpression of RBL2 in these cells resulted in its binding to WNT loci (Fig. 8b). RBL2 shares a similar binding region on WNT loci with RBL1 and pRb, since these proteins also show enrichment on WNT promoter regions by ChIP while their binding is reduced upon RBL2 overexpression, probably due to competing to the same binding region (Fig. 8b).

To further explore the impact of RBL2 on breast cancer CSC proliferation, we collected media from RBL2-LOH cancer cells and from the same lines stably overexpressing RBL2. Thereafter, we placed the conditioned media onto fresh cells and measured cell growth after 48 h. Cell cycle analysis by EdU incorporation and DNA content showed that the cells grown in conditioned media from RBL2-LOH cancer cells displayed increased proliferation (Fig. 8c, d). Importantly, media collected from RBL2-OE cells showed a weaker effect on inducing cell proliferation, while depletion of WNT ligands (WNT4, WNT7B, WNT3A), from the conditioned media also attenuated the positive effect on cell proliferation. Addition of purified WNT ligands (WNT4,

WNT7B, WNT3A) to media had the opposite effect by promoting proliferation while WNT-inhibitor conditioned media slowed proliferation (Fig. 8c, d). The modest positive correlation of E2F4 expression with WNT ligand expression in bulk BRCA tumour (Fig. 8e) prompted to study the self-renewal of breast cancer CSC by tumour sphere assays (Fig. 8f). Conditioned media from cells with WNT pathway inhibition by IWR and IWP2 led to reduced CSC sphere numbers compared to conditioned media alone while the addition of purified WNT ligands significantly increased CSC sphere formation, indicating that the WNT pathway promotes the self-renewal capacity of CSCs. Furthermore, conditioned media from RBL2 OE cells showed reduced supportive effects on CSC sphere formation compared to conditioned media from control cells (Fig. 8f), indicating that RBL2 reduces the secretion of CSC-supportive paracrine factors such as WNT ligands into the media. Proliferation of non-stem breast cancer cells was also affected since depletion of WNT7B/WNT4/WNT3A reduced cell cycle progression while addition of purified WNT7B and WNT4/WNT3A to media had the opposite effect (Fig. 8g). These results indicate that RBL2 can regulate the expression of WNT ligands in breast cancer cells and particularly CSCs.

One important feature shared by PDAC and BRCA is chemoresistance that is primarily mediated by the relative insensitivity of CSCs to chemotherapeutics such as gemcitabine. We treated HCC1500, CAL-51 and MDA-MB-231 with typical chemotherapies for breast cancer, 5-FU, PTX, and DOX, for 5 days and measured CSC marker expressing cells CD133+/CD44+ by flow cytometry and also by measuring total cell survival upon these chemotherapy reagent treatments. The results indicated that the cells surviving 5-FU, PTX and DOX are strongly enriched for CSC markers CD133+/CD44+ whereas non-CSCs not expressing these markers are eliminated by apoptosis due to their lower resistance to 5-FU, PTX and DOX (Supplementary Fig. 10f, g). Next, we investigated whether RBL2 could impact the chemotherapy resistance of BRCA CSCs (Fig. 8h). Conditioned media collected from RBL2 KD cells increased the formation of CSC spheres in the presence of gemcitabine while the antibody-mediated depletion of WNT ligands from this conditioned media partially lost its resistance-promoting effect, suggesting that RBL2 mediated paracrine regulation of WNT signalling impacts CSC chemosensitivity to gemcitabine.

Cell motility and invasiveness represent other key characteristics of CSCs in aggressive cancers due to mediating the metastatic process. Combined expression of WNT ligands (WNT3A, WNT4, WNT7B, WNT9A, WNT10A, WNT11) in BRCA patient tumours has a positive correlation ($R = 0.27$, $p$ value = 0) with the combined expression of key EMT-inducing transcription factors SNAI1, SNAI2, ZEB1, ZEB2 and TWIST1 (Fig. 8i). As EMT contributes to metastases, we decided to analyse the importance of RBL2 on cell invasiveness. For that we used transwell assays with Engelbreth-Holm-Swarm (EHS) matrix extract as basement membrane to measure the invasive capacity of cancer cells exposed to conditioned media from RBL2-LOH cancers cells or media from cells overexpressing RBL2. Interestingly, cancer cell lines deficient for RBL2 were significantly more invasive than RBL2-wt cell lines (Fig. 8j), which is in agreement with their original classification. Conditioned media from cancer cells overexpressing RBL2 decreased invasiveness when compared to cells grown in the conditioned media from RBL2-LOH cancer cells. Importantly, the addition of WNT ligands promoted and WNT inhibitor IWR blocked this effect (Fig. 8j). Thus, WNT signalling seems to be at least partially mediating the invasive capacities of breast cancer cells similar to the observations in PDAC cells. Lastly, we had identified GCN5 as a regulator of PDAC CSCs through an unbiased screening (Fig. 7a–d), and showed its cooperation with E2Fs in WNT expression (Fig. 7i, j). Hence, we investigated the effect of GCN5 inhibition also in BRCA CSCs. Tumour sphere assays on BRCA CSCs indicated that GCN5 inhibition with GS4027 and L-Moses reduces CSC self-renewal (Supplementary Fig. 10h) and β-catenin-dependent promoter activation (Supplementary Fig. 10i), similar to

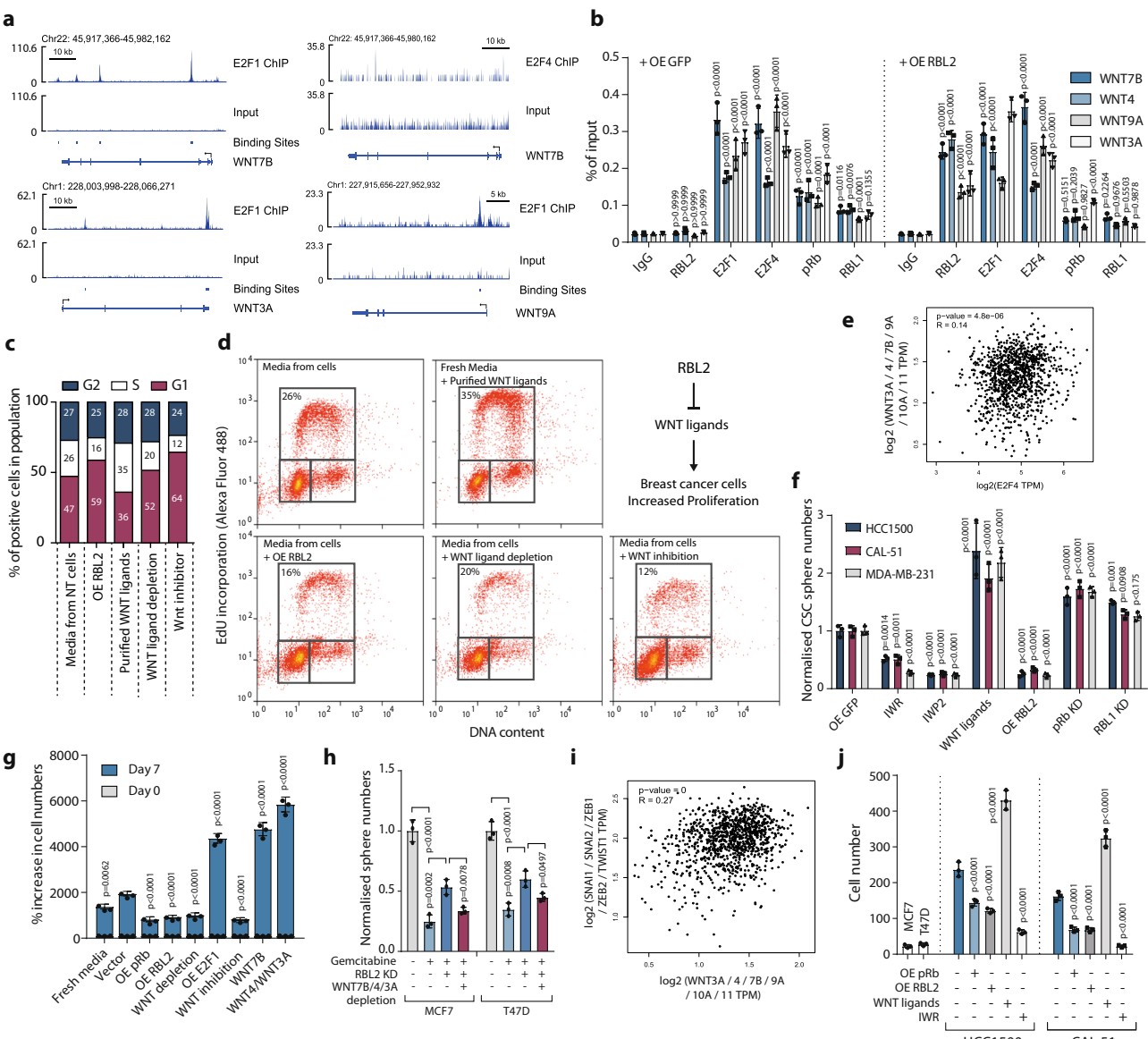

**Fig. 8 | E2Fs and RBs regulate WNT ligand expression in BRCA CSCs. a** E2F1 and E2F4 binding to WNT loci based on ChIP-seq data. **b** E2F1/4 and RBs bind to WNT loci in BRCA CSCs. Cell cycle analysis in HCC1500 CSCs treated with the collected medium. **c** EdU incorporation and DNA content analysis in cancer cells incubated with media for 3 days. **d** Cell cycle distribution in response to media incubation. **e** E2F4 expression has a modest positive correlation with WNT ligand expression in BRCA patients. **f** Conditioned media from pRb KD and RBL1 KD have a supportive effect on CSC sphere formation whereas conditioned media from RBL2 OE cells is less efficient in supporting CSC sphere formation. WNT pathway inhibition reduces CSC sphere formation. **g** The effects of WNT signaing and RB OE conditioned media on BRCA cell proliferation. Data are presented as mean values ± SD. *N* = 3 independent experiments. Statistical analysis was performed by 1-way ANOVA with Dunnett's multiple comparisons test. **h** Conditioned from RBL2 KD cells provide gemcitabine resistance signals to CSCs whereas this conditioned media depleted

from WNT7B/4/3A ligands has lower resistance to gemcitabine compared to undepleted cells. **i** WNT ligand expression has a positive correlation with EMT-inducing transcription factor expression in BRCA patients. **j** RBL2 pathway regulates breast cancer invasiveness via an autocrine mechanism involving WNT ligands. Breast cancer cell invasiveness through basement membrane was analysed by a modified Boyden chamber assay. HCC1500 were placed in media collected from cells with OE RBL2 cells. Purified WNT7B, WNT3A, WNT4 ligands and canonical WNT inhibitor was used to verify the involvement of WNT pathway in cell migration. Significant differences compared to media from untreated cells calculated by two-way ANOVA are marked. Data are presented as mean values ± SD. Graphs depicted in (**b**, **f**, **h**, **j**) are *n* = 3 independent experiments. Statistical analysis in (**b**, **f**, **h**, **j**) was performed by 2-way ANOVA with multiple comparisons with Tukey correction. Source data are provided as a Source Data file.

PDAC CSCs. These data collectively demonstrate that RBL2 and pRb loss leads to increased CSC self-renewal, chemoresistance and invasiveness of breast cancer cells, via a paracrine mechanism involving the overproduction of WNT signalling (Supplementary Fig. 10j). These mechanisms could be targeted by inhibitors of the epigenetic regulator GCN5 that cooperates with E2Fs to induce WNT ligands in CSCs in both PDACs and BRCAs.

## Discussion

By using a proteomic approach of quantitative secretome analysis by SILAC/mass-spectrometry, we uncovered a specifically elevated secretion of WNT ligands by CSCs that mediates autocrine and paracrine signalling in this clinically important cancer cell subpopulation (Supplementary Fig. 11). Upon investigating the mechanisms that regulate WNT ligand expression, we identified a non-cell autonomous

function for the E2F-RB cell cycle regulatory axis and the involved Cyclin D-CDK4/6 and Cyclin E-CDK2 complexes. It is functionally mediated by direct transcriptional induction of WNT ligand expression by E2F1 and E2F4 mostly in late G1 phase and G1/S transition, while pRB and RBL2 bind to E2Fs in G0 phase and S/G2/M phase to repress WNT ligand expression. This mechanism also seems to be functional in normal tissue by balancing the expression of cell-cell signalling molecules of the WNT family. However, the regulatory circuitry seems to be deregulated during tumorigenesis by RB mutations or downregulation or hyperactivation of oncogenic signals such as KRAS activating mutations thus causing elevated WNT ligand secretion that can have autocrine as well as paracrine effects on tumorigenesis. As KRAS activation by hallmark mutations is an early event that occurs in over 90% of all PDACs but is also frequently mutated in other cancers, it has fundamental importance for understanding the changes that drive tumour formation through secretory factors. We further showed that an autocrine/paracrine mechanism involving WNT-β-catenin signalling regulation by E2F-RB axis regulates cell growth, invasiveness and chemoresistance of stem cell-like cells in tumours, especially in pancreatic ductal adenocarcinoma and aggressive forms of breast cancer. Despite the different tissue and cancers contexts (pancreas vs mammary, pancreatic cancer vs breast cancer), the same molecular circuitry between E2F-RB and WNT ligand expression is conserved.

This function, paracrine regulation of cancer stem cells by WNT induction through E2F1/4 and tumour suppression through repression by RBs, extends the current view of the role of these cell autonomous cell cycle regulators. Our study demonstrates that the E2F-RB complex is not simply controlling the cell cycle in a cell-autonomous manner but could also have an essential function in regulating the extracellular niche of cancer stem cells. This has broader implications for understanding the tumorigenic processes and cell to cell signalling. We observed that oncogenes intersecting with the RB pathway and known as cell-autonomous inducers of G1/S transition (e.g. E2F1, E2F4), supported proliferation, chemoresistance and cell migration of pancreatic cancer cells via inducing the secretion of canonical WNT signalling. Of note, the paracrine function is not limited to E2F1/4-RB but extends to other cell cycle regulators including Cyclins D/E and CDK4/6/2 as indicated by our experimental data. In the past, tthe tumour suppressor p53 has been shown to exert paracrine effects in mice[87]. Hence, it is possible that the paracrine function we uncovered here is a broader mechanism by which tumour suppressors control extracellular milieu to limit proliferation and migration potential.

We used a co-culture method that aims to more closely mimic in vivo tissue complexity as well as organoids. Therefore, our results are likely to be directly relevant for in vivo situations because WNT signalling is known to play a role in tumorigenesis of a diversity of tissues including skin, gut, and HSCs[88]. Concerning breast cancer, aside from the role of WNT in normal mammary gland development[89–92], constitutive activation of WNT/β-catenin signalling by ligand overexpression or by increasing β-catenin stability has been shown to promote tumorigenesis in the mammary gland in vivo[93–98]. In addition, forced WNT/β-catenin activity in luminal cells induces adenocarcinomas[94–97]. In basal cells, it causes hyperplasia of basal-type cells that lack lineage markers in nulliparous mice, while in multiparous mice it leads to invasive basal-type carcinomas[99]. Through crosstalk between cancer stem cells and stromal cells, the extracellular signalling by WNT also promotes the formation of a niche for breast cancer stem cells and thereby supports the proliferation of cancer stem cells and their metastatic colonisation[100]. Therefore, the regulation of WNT ligands by RB-E2F pathway as identified by our study uncovers key regulatory mechanisms, which are disrupted upon the formation of aggressive breast cancers.

We found that the RBL2/E2F-WNT axis regulates the stem cell-like characteristics of CSCs in pancreatic cancer and breast cancer including self-renewal and expression of CSC markers,

chemoresistance and invasive capacity. Cancer stem cells are one of the main causes of the aggressiveness of pancreatic cancers and also other cancer types[38,39]. Besides pancreatic cancer and breast cancer, developmentally plastic cancer stem cells have been found in brain, colon, oesophagus, liver, lung, ovarian, prostate, stomach and thyroid cancers, among others[38]. Since our results were confirmed in cancer stem cells from two distinct cancer types, it suggests that similar paracrine mechanisms could also be involved CSCs from other cancers beyond pancreatic and breast cancer. This has therapeutic relevance since targeting the E2F-WNT pathway could result in a dual effect: the classical cell cycle promoting cell autonomous mechanism and the non-cell autonomous signalling that can support CSC dedifferentiation, tumour growth and relapse after chemotherapeutic treatment, and metastases formation in secondary sites due to the invasiveness of CSCs.

The advanced three-colour FUCCI system allowed us to investigate the regulation and mechanistic interlink with the cell cycle, in particular the G0 phase, which has not been done sufficiently in CSCs due to technical limitations studying cell cycle in live cells. We showed that E2F4 and E2F1 have different functions in CSCs since the binding of E2F4 and E2F1 to WNT loci during the different cell cycle phases was distinct. As E2F4 showed binding to WNT loci in G0 cells it is likely to regulate the G0 to G1 phase entry-exit by cooperating with RBs. CSCs are considered to be able to enter a dormant stage in the G0 phase which can protect cells from genotoxic insults such as gemcitabine treatment during chemotherapy. Hence the E2F-RB-WNT circuitry could regulate the entry and exit from G0/cell dormancy or the depth of the 'quiescent' state. CSCs could temporarily reside in a 'shallow quiescen' state, which has been termed 'GAlert' or 'primed'. This is different from a 'deeply quiescent' state from which CSC can exit but this take longer than exiting the shallow or primed state. For instance, the primed state in Hematopoietic Stem Cells (HSCs) has been shown to be regulated by the level of CDK6[101–103]. The strategy for leukaemia stem cells involves forcing the dormant cells to enter the cell cycle, which makes them more susceptible for chemotherapy-mediated elimination[104]. The phenomenon of inducing cells from G0/G1 to S/G2/M phase and killing them with anticancer treatments has also been demonstrated using FUCCI in gastric cancer cells for eliminating quiescent cancer stem-like cells by genetically-engineered telomerase-specific oncolytic adenovirus[105]. In vivo profiling of tumours has been performed[106], and this would provide valuable insight into drug treatment in various cancers including PDAC.

Interestingly, E2F4 is a factor that is traditionally described as a transcriptional repressor[107,108], but in our experiments it had an inductive effect of WNT ligands, while it switches to a repressor function upon the cooperation with RBs. E2F4 has recently been shown to be important for the proliferation and the survival of mouse embryonic stem cells where E2F4 acts in part as a transcriptional activator that promotes the expression of cell cycle genes and other loci by cooperating with histone acetyltransferases[109]. In our screening experiment for discovering epigenetic regulatory factors that would control CSCs, we identified GCN5 as important for maintaining CSC self-renewal and cooperating with E2Fs on WNT loci by regulating H3K9 acetylation while having a weaker impact on H3K4me3. GCN5 is known as a subunit of the ATAC complex that regulates histone acetylation[75–77], so the preferential loss of CSCs over non-CSCs upon GCN5/PCAF bromodomain inhibition using a small molecule inhibitor could serve as a potential candidate for targeted therapeutics if it represents sufficient specificity for certain cancer subpopulations and normal non-cancerous cells. The targeting of epigenetic regulatory factors in CSCs to resensitize or differentiate could be a relatively under-explored area in therapeutic strategies for solid tumours.

Finally, tissues in vivo contain a variety of stromal cell types including fibroblasts, endothelial cells in the blood and lymphatic circulatory systems, adipocytes and various bone marrow-derived cells

including macrophages, neutrophils and mesenchymal stem cells, between which there is likely to be crosstalk and therefore an effect on tumour cells via diverse secretory and intercellular factors[110]. Furthermore, the in vivo microenvironment and extracellular matrix consists of various other signalling factors including cell adhesion molecules, tight junction proteins, cytokines and growth factors[111]. These aspects are particularly relevant for PDACs that have extensive fibrosis that can provide a physical barrier for therapeutic compounds to reach cancer cells[70]. Our results indicated that the RBL2/E2F-WNT can signal through paracrine mechanisms to induce fibroblast proliferation, which could contribute to fibrosis in PDACs, breast cancers and other cancer types. Targeting the dense fibrotic microenvironment is considered to be among the promising strategies that could enhance the efficiency of therapeutics as observed for Sonic Hedgehog (Shh)[112,113]. It will be interesting to learn how the other components of the tissue microenvironment are affected by RBs during developmental processes, normal adult tissue homoeostasis and during tumorigenic processes. Targeting the CDK-RB-E2F axis for cancer therapy is becoming an increasingly attractive strategy due to recent advances in developing specific small molecule inhibitors targeting either CDK4/6 or E2Fs[114,115]. Deciphering the function of WNT signal in PDAC formation and it role particularly in PDAC CSCs, will benefit from in vivo tumorigenesis experiments to assess the relative importance of the WNT ligands in carcinogenesis, and the relative importance of RB1, RBL1 and RBL2 family members as regulators of WNT ligands in pancreatic tumorigenesis. In addition, the function of GCN5 as a possible therapeutic target in PDACs should be addressed in future in vivo studies. Thus, the discovery of paracrine functions for the tumour suppressor RBL2 paves the way for future investigations linking the regulation of early development and tumorigenesis, which has clear translational utility for devising more efficient cancer therapeutics in the future.

## Methods

### Ethical approval
Our research complies with all relevant ethical regulations. Patient samples were obtained from Oxford Biobank with written informed consent at the Oxford University Hospitals NHS and with approved ethics permissions (OCHRe ref: 21/A126, REC reference: 19/SC/0173; OCHRe ref: 19/A176, REC reference: 19/SC/0173).

### Cell lines and cell culture
The L3.6pl and FG cells were provided by Isaiah J. Fidler through the cell line bank at the University of Texas MD Anderson Cancer Center. L3.6pl and FG cell lines have a KRAS p.Gly12Asp (c.35G>A) genetic background. The A13A cells were provided by Christine Iacobuzio-Donahue at The Memorial Sloan Kettering Cancer Center. The genetic background of A13A cell line is copy number gain of GATA-6 and cTAGE1; KRAS G12V; Tp53 WT; SMAD4 WT. PANC1 and HPAFII cells were obtained from ATCC. PANC1 has a homozygous deletion of CDKN2A; Mutations: KRAS p.Gly12Asp (c.35G>A) heterozygous; TP53 p.Arg273His (c.818G>A) homozygous. HPAFII has mutations: CDKN2A p.Arg29_Ala34del (c.85_102del18) heterozygous; KRAS p.Gly12Asp (c.35G>A) heterozygous; TP53 p.Pro151Ser (c.451C>T) homozygous. MDA-MB-231 has a CDKN2A and CDKN2B Homozygous deletion; Mutations: KRAS p.Gly13Asp (c.38G>A) Heterozygous; BRAF p.Gly464Val (c.1391G>T) heterozygous; TERT c.1-124C>T (c.228C>T); TP53 p.Arg280Lys (c.839G>A) homozygous. CAL-51 has a PIK3CA p.E542K mutation. HCC1500 has PIK3CA c.3075C>T and c.*155C>T heterozygous mutations. MCF7 cell line has a homozygous deletion of CDKN2A and mutations: GATA3 p.Asp336Glyfs*17 (c.1006dupG) heterozygous; PIK3CA p.Glu545Lys (c.1633G>A) heterozygous; TP53 wt. T47D has mutations: PIK3CA p.His1047Arg (c.3140A>G) heterozygous; TP53 p.Leu194Phe (c.580C>T) homozygous. The HPDE6c7 cell line, with normal *KRAS, Tp53*, c-*myc*, and *p16*[INK4A] genotypes, was purchased

from Kerafast (USA) and maintained in keratinocyte serum-free medium supplemented with human recombinant epidermal growth factor and bovine pituitary extract (Thermo Fisher Scientific, 17005042). HPDE6c7+KRAS (H6C7) has a KRAS (G12V) that is expressed together with eGFP.

For standard cancer cell cultures, cells were grown at 37 °C humidified incubator containing 5% $CO_2$ in Dulbeccos modified Eagles medium (DMEM), high glucose, GlutaMAX™ Supplement, pyruvate (Thermo Fisher Scientific, Inc.), 100 U/ml penicillin, 100 mg/ml streptomycin (Thermo Fisher Scientific, Inc.), MEM Non-Essential Amino Acids Solution 1X (Life Technologies; Thermo Fisher Scientific), MEM Vitamin Solution 1X (Life Technologies; Thermo Fisher Scientific) and 10% inactivated foetal calf serum (FCS; Life Technologies). For 3D cultures, cells we have grown in ultra-low attachment plates (Thermo Fisher Scientific, Inc.) in Dulbecco's Modified Eagle Medium/F12 (Sigma-Aldrich) supplemented with 2.5 mM L-Glutamine (Thermo Fisher Scientific, Inc.), 1 % B27 supplement (Life Technologies), 100 U/ml penicillin, 100 mg/ml streptomycin (Thermo Fisher Scientific, Inc.), 20 ng/ml basic thermostable fibroblast growth factor (FGF-Basic TS, Proteintech) (5000 cells/ml). Cell lines were routinely authenticated and checked for mycoplasma infection.

Mutations in human cancer lines were identified based on data from Sanger Catalogue of Somatic Mutations In Cancer (COSMIC) v99[116]. Breast cancer lines were obtained from ATCC or from collaborators. Breast cancer cells were cultured according to ATCC guidelines.

### PDAC organoids
Human PDAC patient-derived organoids: HCM-BROD-0124-C25 (ATCC; Cat. No. PDM-137), HCM-BROD-0009-C25 (ATCC; Cat. No. PDM-107), HCM-CSHL-0073-C25 (ATCC; Cat. No. PDM-24) and cultured by using the items and conditions as suggested by ATCC guidelines.

### Generating retinoblastoma family protein knockdown cells
For RB single knockdown, previously validated shRNA expression vectors (Sigma-Aldrich, Cat no. SHCLNG NM_00032, SHCLNG NM_002895, SHCLNG NM_005611, directed against pRb, RBL1 or RBL2 respectively, were transfected into cells with lipofectamine 2000[66] and grown for 3 days. Cells were then cultured in the presence of puromycin until antibiotic-resistant colonies appeared. These were picked and characterised for knockdown efficiency. We characterised two knockdown clones generated from separate shRNA constructs in more detail.

### RB and E2F inducible knockdown (iKD)
Inducible knockdown of RBs and E2Fs was performed by using a dox-inducible CRISPR interference (CRISPRi) knock in construct that was first targeted to AAVS1 locus to create stable lines as described by Bruce Conklin. gRNA sequences for pRb, RBL1, RBL2, E2F1, E2F4 knockdown were obtained from the GenScript gRNA database and cloned into the gRNA construct according to the protocol from[1]. The 2 mM Doxycyclin (Dox) was used for dCas9-KRAB mediated inducible knockdown of genes. Dox was added together with fresh media. pAAVS1-NDi-CRISPRi (Gen2) was a gift from Bruce Conklin (Addgene plasmid # 73498; http://n2t.net/addgene:73498; RRID:Addgene_73498). pgRNA-CKB was a gift from Bruce Conklin (Addgene plasmid # 73501; http://n2t.net/addgene:73501; RRID:Addgene_73501).

### Generating RB and RBL2 overexpressing cells
For RB and RBL2 overexpression, sequence-validated Gateway attL-flanked entry clones (Source BioScience Lifesciences, Cat no. B0065, T8278, for RB and RBL2 overexpression respectively), were transferred into a Gateway-compatible pTP6 vector containing a CAG promoter.

The inserts were confirmed by sequencing. Vectors were transfected into cells by lipofection[66] and grown for 3 days. Thereafter, cells with a stable integration were selected by continuous presence of puromycin. Individual clones were picked, propagated and used for subsequent analyses.

### Inducible RB overexpression (iOE)

For overexpression of pRb/RBL1/RBL2 and E2F1/E2F4 we used SP-dCas9-VPR with a doxycycline-inducible expression system as described previously[117,118]. PB-TRE-dCas9-VPR was a gift from George Church (Addgene plasmid # 63800; http://n2t.net/addgene:63800; RRID:Addgene_63800). gRNA_Cloning Vector was a gift from George Church (Addgene plasmid #41824; http://n2t.net/addgene:41824; RRID:Addgene_41824). gRNA sequences were obtained from GenScript gRNA database. 2 mM Doxycyclin was used for inducible induction of RBs and E2Fs.

### Tumour sphere assays

Cells are seeded into ultra-low attachment plates in Dulbecco's Modified Eagle Medium/F12 (Sigma-Aldrich) supplemented with 2.5 mM L-Glutamine (Thermo Fisher Scientific, Inc.), 1% B27 supplement (Life Technologies), 100 U/ml penicillin, 100 mg/ml streptomycin (Thermo Fisher Scientific, Inc.), 20 ng/ml basic thermostable fibroblast growth factor (FGF-Basic TS, Proteintech) (5000 cells/ml). For tumour sphere formation assay, cells were passaged after 1-week incubation, and grown for another week after which the tumour sphere numbers were counted under a phase-contrast microscope using the 40× magnification lens or by Celigo Image Cytometer (Nexcelom). After 7 days of incubation the tumour spheres were harvested by using a 40 μm cell strainer and centrifuged for 5 min at $200 \times g$ at RT. Dissociate the pellet of tumour spheres to single cells using trypsin, and then expanded for another 4 days before performing the treatments of samples.

### Cocultures

Cells were cocultured by using the items and conditions as suggested by ATCC guidelines for organoids.

### Knockin cell lines for OCT4

pCCC construct and OCT4 TALEN constructs (pTALEN_V2-OCT4F, pTALEN_V2-OCT4R) constructs were a gift from Francis C. Lynn and have been published (PMID 25474420). OCT4-eGFP-PGK-Puro was a gift from Rudolf Jaenisch (Addgene plasmid #31937) and has been published[119]. Cells were transfected with Lipofectamine 3000 (ThermoFischer Scientific) and cultured for 4 days after transfection before selecting 0.25 μg/mL puromycin (Sigma). Colonies were individually picked, trypsinized and placed into 24-well plates with 500 μl of media. Once clones were close to confluent, cells were replica plated for genotyping, freezing and expanding the correctly targeted clones. Genomic DNA was extracted using Promega Wizard SV Genomic DNA Purification System (Promega) and genotyping was performed as described[120]. Positive clones were analysed by flow cytometry to estimate the frequency of eGFP positive cells in the cancer cell population. We additionally used primers GGTGCTCAGGTAGTGGTTGTCG and CTCTAATGTCCTCCTCTAACTGCTCTAGG for Oct4-GFP region verification, as well as CACAACCTCCCCTTCTACGAGC and GCATCATTGAACTTCACCTTCCCTC for Oct4-Puromycin region verification.

### RNA isolation and cDNA synthesis

Total RNA was isolated by RNeasy RNA Extraction Kit (Qiagen) according to manufacturer's guidelines. RNA was then eluted in 30 μl of water and the concentration was measured using Nanodrop. The master mix was prepared as follows: 8 μl 5× First-Strand Buffer (Invitrogen), 0.5 μl Random primers (0.5 μg/ml) (Promega Cat. C1181), 1 μl dNTP mix (10 mM each) (Promega Cat.U1515), 2 μl 0.1 M DTT, 0.5 μl RNase Out, 0.25 μl Superscript III Reverse Transcriptase (Life

Technologies). 500 ng of total RNA into a separate tube with 11.75 μl RNase-free water. RNA was heated to 65 °C for 5 min and allowed to chill on ice for 2 min. 8.25 μl of the master mix was added to RNA. The reaction was incubated at 25 °C for 10 min and then at 42 °C for 50 min. The reaction was then inactivated by heating at 70 for 15 min.

### Immunostaining

Methods for immunostaining have been described previously[66,121,122]. Cells were fixed for 20 min at 4 °C in PBS 4% PFA (electron microscopy grade), rinsed three times with PBS, and blocked and permeabilized at the same time for 30 min at room temperature using PBS with 10% Donkey Serum (Biorad) and 0.1% Triton X-100 (Sigma). Incubation with primary antibodies diluted in PBS 1% Donkey Serum 0.1% Triton X-100 was performed overnight at 4 °C. Samples were washed three times with PBS, and then incubated with AlexaFluor secondary antibodies for 1 h at room temperature protected from light. Cells were finally washed three times with PBS, and Hoechst (Sigma) was added to the first wash to stain nuclei. Images were acquired using an LSM 700 confocal microscope (Leica).

### Q-PCR

Methods for Q-PCR have been described previously[66,121,122]. Q-PCR data are presented as the mean of three independent experiments and error bars indicate standard deviations. Antibodies and primer sequences have been listed in Supplementary Tables 1 and 2.

### Chromatin immunoprecipitation

Cells were washed with PBS and detached from the plate by incubating them for 10 min at 37 °C in Cell Dissociation Buffer (GIBCO). ChIP was carried out as described before[66,121,122] with some modifications. The ChIP experiments were performed in triplicate. All steps were performed on ice or at 4 °C and ice-cold buffers and PBS were supplemented with 1 mg/ml Leupeptin, 0.2 mM PMSF, and 10 mM NaButyrate were used unless otherwise stated. Approximately $5 \times 10^6$ cells were used per sample and cross-linked with 1% formaldehyde for 15 min. Cross-linking was stopped by incubating samples with glycine at a final concentration of 0.125 M for 5 min at room temperature, and the cells were washed with PBS followed by pelleting at 250 g for 5 min. The cells were then crosslinked with 2 mM EGS in PBS solution and incubated on a rotator for 45 min at room temperature. Cells were pelleted, the crosslinking solution discarded and cells washed with 30 ml cold PBS twice. The pellet was re-suspended in 2 ml ChIP Cell Lysis Buffer (CLB: 10 mM Tris pH8, 10 mM NaCl, 0.2% NP-40) and incubated for 10 min to lyse the plasma membranes. Nuclei were pelleted at 600 g for 5 min, lysed in 1.25 ml of ChIP Nuclear Lysis Buffer (NLB: 50 mM Tris pH8, 10 mM EDTA, 1% SDS) for 10 min, and then 0.75 ml of ChIP Dilution Buffer (DB: 20 mM Tris pH8, 2 mM EDTA, 150 mM NaCl, 0.01% SDS, 1% Triton X-100) was added to the samples. Chromatin was sonicated in 15 ml Diagenode Bioruptor Pico water bath sonicator with an automated water cooling system, by performing 30 cycles of 30 s ON, 45 s OFF. This protocol resulted in the homogeneous generation of fragments of 100–400 bp. Samples were clarified by centrifugation at 16,000 g for 10 min, and diluted with 3.5 ml of DB. After pre-clearing with 10 μg of non-immune IgG for 1 h and 50 μl of Protein G-Agarose for 2 h, ChIP was performed overnight in rotation using specific antibodies (Supplementary Table 1) or non-immune IgG as a control. After incubation for 1 h with 30 μl of Protein G-Agarose, beads were washed twice with ChIP Washing Buffer 1 (WB1: 20 mM Tris pH8, 2 mM EDTA, 50 mM NaCl, 0.1% SDS, 1% Triton X-100), once with ChIP Washing Buffer 2 (WB2: 10 mM Tris pH8, 1 mM EDTA, 0.25 M LiCl, 1% NP-40, 1% Deoxycholic acid), and twice with Tris-EDTA (TE: 10 mM Tris pH8, 1 mM EDTA). Precipitated DNA was eluted with 150 μl of ChIP Elution Buffer (EB: 100 mM $NaHCO_3$) twice for 15 min at room temperature in rotation, and processed as follows in parallel with 300 μl of sonicated chromatin non-used for ChIP (Input). Cross-linking was reverted by

adding NaCl to a final concentration of 300 mM for protein-DNA de-crosslinking and incubated at 65 °C for 5 h and 1 µg RNase A (Sigma) to digest contaminating RNA. Finally, 60 µg of Proteinase K (Sigma) was added overnight at 45 °C. DNA was extracted by sequential phenol–chloroform and chloroform extractions and precipitated overnight at −80 °C in 100 mM NaAcetate, 66% ethanol and 50 µg of glycogen (Ambion) as a carrier. After centrifugation at 16,000 g for 1 h at 4 °C, DNA pellets were washed once with ice-cold 70% ethanol, and finally, air dried. ChIP samples were resuspended in 30 µl and 1:10 of the samples were used in Q-PCR for verifying the ChIP samples.

E2F/RB binding site identification was performed as follows. E2F/Rb proteins usually bind to their target loci within 2 kb of the transcription start site[46–49,123]. Hence we designed ChIP primers every 250 bp within 2 kb upstream to 500 bp downstream of transcription start site and tested them by Q-PCR after performing ChIP E2F1. These results identified primers that allowed the optimal detection of E2F binding, while primer pairs further away from These regions did not show enrichment for E2F and RBL2 binding on WNT loci. All ChIP experiments also included negative binding regions on other loci such as Smad7, which did not show any enrichment for E2F/RB binding.

## Recombinant proteins
The recombinant human WNT4 protein (RnD Systems; Cat. No. 6076-Wn; >60% purity), recombinant human WNT8A (Genemed PlexBio; Cat No. 90007-02; >95% purity), Recombinant Human WNT7A protein (Abcam; Cat No. ab116171), Recombinant Human WNT7B protein (Abcam; Cat No. ab289780) were used at 100 ng/ml for experiments. According to RnD Systems WNT4 product information, the typical $ED_{50}$ for this WNT4 protein is 25–100 ng/ml, so we chose to proceed with the upper limit of $ED_{50}$ in our experiments for WNT proteins.

## DNA constructs
The TOPFlash constructs have been described by ref. 124. M50 Super 8x TOPFlash was a gift from Randall Moon (Addgene plasmid #12456; http://n2t.net/addgene:12456; RRID:Addgene_12456), M51 Super 8x FOPFlash (TOPFlash mutant) was a gift from Randall Moon (Addgene plasmid #12457; http://n2t.net/addgene:12457; RRID:Addgene_12457).

## Depletion of WNT ligands from media
WNT4, WNT8A, WNT7A and WNT7B specific antibodies were bound to Protein G-Agarose beads (1ug of antibody per 2.5 µl of packed beads) for 2 h at 4 °C rotating and used at a ratio of 1ug of each antibody per 1 ml of collected media. Control depletion was carried out with an IgG antibody and confirmed by western blotting.

## Cell incubation with conditioned media
Media was incubated with 70–80% confluent cells for 24 h, before collection. The media collected from cells, aliquoted, stored at −80 °C, and thawed freshly just before use. Cells were cultured in the collected media not more than 24 h before substituting for a fresh aliquot. To avoid possible autocrine effects via WNT signalling due to inconsistent cell density (the more cells the higher levels of WNT in media), particular care should be taken for plating cells at the same density across the experimental samples.

## Luciferase assay
Cells were transfected with a SMAD2/3 reporter construct (SBE4-luciferase), SOX17 or GSC promoter constructs[125] and Renilla luciferase at a ratio of 10:1, using Lipofectamine 2000 (Invitrogen)[126]. Luciferase activity was measured with the dual luciferase assay kit following (Promega) manufacturer instructions. Firefly luciferase activity was normalised to Renilla luciferase activity for cell numbers and transfection efficiency. Samples were analysed on a Glomax Luminometer and software. We used a 500 bp actin promoter region driving luciferase expression as a negative control for WNT/β-catenin responsive gene.

## Transwell assays
Cancer cell invasiveness was analysed by using a modified Boyden chamber based assay, CultureCoat 96 Well High BME Cell Invasion Assay (Trevigen, Cat. No: 3483-096-K) or 24-well tanswell inserts with 8 µm pores (Sarstedt, Cat no. 83.3932.800) with EHS Matrix Extract as basal membrane (Merck) according to manufacturer's guidelines. 25,000 cells were used per well and incubated for 24 h before analysis. Cancer cells were placed in the media collected from their corresponding cell lines overexpressing RBL2, Cyclin D1 or treated with 1 µM CDK4/6 inhibitor PD0332991 or 1 µM IWR-1-endo for 24 h prior to migration assay.

## EdU incorporation assay
Cell cycle distribution was analysed by Click-It EdU incorporation Kit (Invitrogen) according to manufacturer's guidelines. Flow cytometry was carried out with a BD MoFlo flow cytometer and analysed by FloJo software. Cells were cultured in media collected from cells with different treatment conditions for 72 h, replacing the media every 24 h.

## Flow cytometry for cell cycle analysis
A13A and FG PDAC cells in which the FUCCI construct was incorporated were taken from adherent conditions and counted, then plated in spheroid conditions at a density of 5000 cells/1 mL medium for 10 days, before the cells were collected and analysed using Fortessa (BD Bioscience). Passaging was performed a day 5, after which cells were plated again in spheroid conditions, with the same initial density of 5000 cells/1 mL medium. Compounds were added on day 7, and the treatment lasted for 72 h. Experiment was performed in three replicates. The data was analysed in FlowJo.

## Cell sorting by FACS
FACS on FUCCI cells was performed as described before with modifications[66,67,127]. Cells were harvested with trypsin and washed twice with cold PBS. Approximately $5 \times 10^6$ cells were used per sample and cross-linked with 1% formaldehyde for 15 min. Cross-linking was stopped by incubating samples with glycine at a final concentration of 0.125 M for 5 min at room temperature, and the cells were washed with PBS followed by pelleting at 250 g for 5 min. The cells were then crosslinked with 2 mM EGS in PBS solution and incubated on a rotator for 45 min at room temperature. Cells were pelleted, the crosslinking solution discarded and cells washed with 30 ml cold PBS twice. Cells were resuspended gently in 2 ml sterile PBS 1% BSA, and subjected to cell sorting by gating according to the mAG/mKO2/mKate2 FUCCI signals. The cell sorting was performed with FACSAria II, II, Fusion cell sorters and the cells were sorted directly into collection tubes. After sorting, the cells were pelleted and subjected to ChIP-qPCR.

## Cell proliferation assay
Cells (GFP-expressing fibroblasts or cancer cells) were cultured in media collected from various treatments, by substituting the media every 24 h.

## Western blot analysis
Protein was isolated by lysing cells with RIPA Buffer (Sigma-Aldrich) supplemented by cOmplete EDTA-free protease inhibitor (Roche) and PhosSTOP™ (Sigma-Aldrich) and extracting the supernatant after high-speed centrifugation at 4 °C. Protein quantification was performed using the Pierce BCA Protein Assay kit following the manufacturer's protocol. Isolated proteins were prepared for SDS-PAGE separation by dilution with 4× NuPAGE Sample buffer (Invitrogen), addition of NuPAGE™ Sample Reducing Agent ((10X), Invitrogen), 95 °C for 5 min,

and cooling. Isolated proteins were then analysed by Western blotting. Protein separation via SDS-PAGE was performed on a NuPAGE 4–12% or 12% Bis-Tris gel (Life Technologies) with NuPAGE™ MOPS SDS Running Buffer (Life Technologies). Proteins were transferred to a PVDF membrane, blocked with 5% milk in PBS and 0.05% tween 20, probed with protein-specific antibodies (Supplementary Table 1), incubated with horseradish peroxidase-conjugated secondary antibodies, and visualised via enhanced chemiluminescence using the SuperSignal West Pico Chemiluminescent Substrate (Thermo Scientific). All antibodies were diluted in 5% milk in PBS and 0.05% tween 20. Quantification was performed using ImageJ gel analysis tool.

## Protein co-immunoprecipitation
Cells were harvested with trypsin and washed twice with cold PBS. For cytoplasmic lysis, cells were suspended in 5 times packed cell volume (1 μl PCV = $10^6$ cells) equivalent of Isotonic Lysis Buffer (10 mM Tris HCl, pH 7.5, 3 mM CaCl, 2 mM $MgCl_2$, 0.32 M Sucrose, Complete protease inhibitors and phosphatase inhibitors), and incubated for 12 min on ice. Triton X-100 was added to a final concentration of 0.3% and incubated for 3 min. The suspension was centrifuged for 5 min at 1500 rpm at 4 °C and the supernatant (cytoplasmic fraction) was transferred to a fresh chilled tube. For nuclear lysis, nuclear pellets were resuspended in 2 × PCV Nuclear Lysis Buffer+Triton X-100 (50 mM Tris HCl, pH 7.5, 100 mM NaCl, 50 mM KCl, 2 mM $MgCl_2$, 1 mM EDTA, 10% Glycerol, 0.3% Triton X-100, Complete protease inhibitors and phosphatase inhibitors) and dounce homogenised. The samples were incubated with gentle agitation for 30 min at 4 °C and then centrifuged with a Ti 70.1 rotor at 22,000 rpm for 30 min at 4 °C or with a Ti 45 rotor for 30 min at 20,000 rpm at 4 °C. The chromatin pellets were dounce homogenised in 2 x PCV Nuclear Lysis Buffer +Triton X-100 and Benzonase until the pellets gave much less resistance. The samples were incubated at RT for 30 min and centrifuged with either a Ti 70.1 rotor for 30 min at 22,000 rpm at 4 °C or with a Ti 45 rotor for 30 min at 20,000 rpm at 4 °C. Samples were incubated with 5 μg of cross-linked antibodies for 12 h at 4 °C. Beads were washed five times with ten bead volumes of Nuclear Lysis Buffer and eluted in SDS western blotting buffer (30 mM Tris pH 6.8, 10% Glycerol, 2% SDS, 0.36 M beta-mercaptoethanol (Sigma), 0.02% bromophenol blue) by heating at 90 °C for 5 min. Samples were analysed by standard western blotting techniques. As an alternative method, we also used the nuclear complex co-IP kit (Active Motif, cat. 54001) according to the guidelines.

## Human primary breast tumours
All human primary breast tumour data, including mutations, CNAs, expression data and clinical prognosis of ten breast cancer subgroups was obtained from databases collected by the METABRIC consortium as published in ref. 18 or by The Cancer Genome Atlas Network as published in ref. 128.

## Bulk RNA-sequencing analysis from TCGA-PAAD project
PDAC data were downloaded from the TCGA-PAAD project, and breast cancer data were downloaded from the TCGA-BRCA project. Scoring of WNT ligand gene expression was performed using ssGSEA in the corto R package (version 1.1.10)[129]. Pearson's correlation analysis was done between ssGSEA score and gene expression.

## Single-cell RNA-sequencing analysis
PDAC patient single-cell RNA-seq data were obtained from the study of ref. 42. All the PDAC samples were integrated using the harmony R package (version 0.1.0)[130]. Cells with more than 20% mitochondrial reads and less than 5% ribosomal reads were filtered. Potential doublets were removed using the DoubletFinder R package (version 2.0.3)[131]. Cell type was annotated using the SingleR R package (version 1.8.1)[132] and manually curated based on the marker genes. Malignant

cancer cells were found using the copykat R package (version 1.0.8)[133]. The other analyses were performed using the Seurat R package (version 4.1.0)[134]. All the analyses (including statistical analysis) were done in R (version 4.1.3). We used the following genes for cell type classification: epithelial/cancer cells (EPCAM, LCN2, AGR2, KRT8, KRT18), T cells (CD3D, CD3E), Myeloid cells (LYZ, CD14), NK cells (KLRF1m NCAM1), B cells (CD19, MS4A1, IGKC), fibroblasts (COL1A1, COL1A2, COL5A2, DCN, LUM), mast cells (TPSAB1, TPSB2, CPA3), neural cells (GPM6B, PLP1), endothelial cells (negative for listed markers).

## Constructing single-cell trajectories in PDAC
The Monocle2 package (v2.8.0)[135] was used to analyse single cell trajectories in order to discover the cell-state transitions.

## Ligand–receptor pairs determination
PDAC patient single-cell RNA-seq data were obtained from the study of ref. 42, where data from 20 treatment-naïve donors out of 27 total patients were included in this study. The paired reads were mapped to the hg38 reference genome to generate gene expression matrices using CellRanger v7.0.0 (https://www.10xgenomics.com/cn/support/software/cell-ranger). The raw matrices were then analysed using the Seurat R package (v4)[134]. Cells with low-quality profiles were excluded based on the number of detected genes (>400) and the total number of UMIs (>600). Mitochondria gene-enriched cells were further removed afterwards during clustering. The raw read counts were normalised using the NormalizeData function, and variable genes were identified for each sample. Data from different samples were then integrated using the FastMNN algorithm. To identify cell types, Uniform Manifold Approximation and Projection (UMAP) dimensionality reduction was performed on integrated data with the RunUMAP function. The FindNeighbors and FindCluster functions were used to cluster cells based on the global transcriptional profile.

For cell-cell interaction inference, normalised expression matrix and cluster annotations were exported from the Seurat object as input for cellphonedb. Cellphonedb v4.1.0[136] was used as a ligand-receptor interaction database. 'statistical analysis' method from Cellphonedb tools was used to assess the strength and significance of interactions. For the circus plots we used Circlize package[137], version 0.4.15 in R to draw the plots (https://www.R-project.org/).

## ATAC-sequencing
Cells were washed once with PBS, collected in Cell Dissociation Buffer (Gibco 13150-016) or TrypLE and centrifuged at 300 g for 3 min. The cell pellets were then resuspended in 2 ml of 4 °C PBS and counted by haemocytometer for using 100,000 cells in the subsequent step. Cells were centrifuged at 300 g for 3 min, the supernatant aspirated, the cell pellet resuspended in 150 μl of Isotonic Lysis Buffer (10 mM Tris-HCl pH 7.5, 3 mM CaCl, 2 mM $MgCl_2$, 0.32 M Sucrose and Protease Inhibitors, Roche), and incubated for 12 min on ice. Triton X-100 from a 10% stock was then added at a final concentration of 0.5%, the samples were vortexed briefly and incubated on ice for 6 min. The samples were centrifuged for 5 min at 400 g at 4 °C, and the cytoplasmic fraction removed from the nuclear pellet. The samples were resuspended gently in 625 μL of PBS and transferred to a fresh 1.5 ml eppendorf tube. The nuclei were centrifuged at 1500 g for 3 min at 4 °C and the supernatant aspirated thoroughly from the nuclear pellet. This step was immediately followed by tagmentation (Nextera DNA Sample Preparation Kit for 24 Samples, FC-121-1030) by resuspending each sample in 100 μL Nextera mastermix (52.5 μl TD buffer, 42.5 μl of water and 5 μl of TDE1 per reaction). The nuclear pellet was resuspended thoroughly by pipetting and incubated at 37 °C for 1 h shaking at 300 rpm. The reaction was stopped with 300 μL of buffer PB from the Qiagen PCR purification kit, followed by Qiagen PCR clean-up protocol using MinElute columns and eluting each sample in 18 μl buffer EB. For the control sample, the nuclear pellet was subjected to genomic DNA

isolation with GenElute Mammalian Genomic DNA Miniprep Kit (Sigma, G1N70) according to manufacturer's protocol, and the purified genomic DNA was thereafter immediately used for tagmentation as for other ATAC-seq samples.

Next, a PCR reaction (for all samples including control sample) was performed with the following constituents: 10 μl template from tagmentation, 2.5 μl I7 primer (Nextera® Index Kit with 24 Indices for 96 Samples, FC-121-1011), 2.5 μl I5 primer, 10 μL Nudease Free H₂O 25 μl NEBNext High-Fidelity 2x PCR Master Mix (New England Labs Cat #M054 and 10 μL Nuclease Free H₂O. The PCR settings were as follows: at 72 °C for 5 min, initial denaturation at 98 °C for 30 s, then 12 cycles of 98 °C for 10 s, primer annealing at 63 °C for 30 s and elongation at 72 °C for 1 min, and holding at 4 °C. After completing the PCR, the sample was stored at −20 °C. The PCR primers were removed with 1 ×0.9:1 SPRI beads (Beckman Coulter, Cat no. A63880) according to manufacturer's protocol and samples eluted in 20 μl. 2 μl of the samples were run on Agilent HS Bioanalyzer HS for confirming the size selection of the ATAC libraries. ATAC-sequencing was performed by Illumina HiSeq 2000 sequencing with 75 bp PE for obtaining more than 40 million mapped reads per library.

## ATAC-sequencing analysis

Sequencing reads from the ChIP-seq and ATAC seq experiment were aligned to the human genome (hg38) using bowtie with reporting mode,' −best −strata −v2'.Deeptools was used to generate covergae track(bigwig). Coverage track was visualised by using UCSC genome browser. Peak calling was performed by using macs2 peak caller with default parameters for ChIP seq, and with parameter '--nomodel --shift −100 --extsize 200' for ATAC seq. Peaks annotated with nearest gene information by using BEDTools. Peak distribution over different genomic features were summarised by using Bioconductor package ChiPpeakAnno. Motif enrichment analysis within peak regions was performed using HOMER. All plots were generated using R package 3.6.

## SILAC-labelling for mass spectrometry analysis

The A13A, A13A-CD133⁺, and H9 ESC cells were cultured in the same condition where the basal medium was replaced with SILAC DMEM: F12 basal medium (Thermo Fisher Scientific, Inc.) with separate supplements accordingly. Three SILAC-labels including light labelled with L-Arginine and L-Lysine, medium labelled with 13C6-L-Arginine-HCl and D4-L-Lysine, and heavy labelled with 13C615N4-L-Arginine-HCl and 13C615N4-L-Lysine, provided by the kit or purchased separately form the same company were added into the culture medium to generate light, medium and heavy labelled cells. For complete incorporation the cells were cultured in such SILAC condition for over five doubling times before proceeding experiments. The SILAC incorporation efficiency was checked and showed >97% in secretome and >99% in total lysate.

## Secretome collection and preparation for mass spectrometry analysis

The SILAC labelled cells were collected and washed with DPBS buffer (Life Technologies; Thermo Fisher Scientific, Inc.) for four times before seeding into fresh B27-depleted or FCS-depleted SILAC medium for 48 h. The condition medium was collected at 200 × g for 4 min at room temperature to pellet the cells, followed by adding cOmplete EDTA-free protease inhibitor cocktail (Roche Molecular Systems, Inc.) and PhosStop™ phosphoatase inhibitor (Sigma-Aldrich Ltd.) and a subsequent centrifugation at 2000 × g, 30 min at 4 °C to remove cell debris. Total secretome was precipitated in 6% trichloroacetic acid (Sigma-Aldrich Ltd.) at 4 °C overnight and centrifuged at 16,500 × g at 4 °C for 30 min. The secretome pellet was washed once with pre-cold 80% acetone (Fisher Scientific UK Ltd.) at 16,500 × g at 4 °C for 30 min and resolved in 200 μl SDS solubilisation buffer (5% SDS,

Sigma-Aldrich Ltd; 50 mM triethylammonium bicarbonate buffer pH 7.5, Sigma-Aldrich Ltd.). The protein concentration was determined using Pierce BCA protein assay (Thermo Fisher Scientific, Inc.) or Qubit protein assay (Thermo Fisher Scientific, Inc.). Equal amount secretome from three labelled cells was pooled together for a reduction in Bond-Breaker™ TCEP Solution (Thermo Fisher Scientific, Inc.) and alkylation in 20 mM iodoacetamide (Sigma-Aldrich Ltd.) at room temperature for 30 min. The pooled secretome was then acidified and applied to S-Trap™ micro spin column (Protifi™) for trypsin digestion (Promega UK) at 37 °C overnight according to the S-Trap™ digestion protocol. The digested secretome was eluded in 60 μl of 25% acetonitrile (Thermo Fisher Scientific, Inc.) and 0.1% trifluoroacetic acid (Thermo Fisher Scientific, Inc.) for MS proteome analysis.

## LC-MS/MS secretomic data search and analysis

Dried peptides were reconstituted in 5% DMSO and 5% formic acid and were analysed by liquid chromatography tandem mass spectrometry using Ultimate 3000 UHPLC (Thermo Fisher Scientific, Inc.) connected to an Orbitrap Fusion Lumos Tribrid (Thermo Fisher Scientific, Inc.). Briefly, peptides were loaded onto a trap column (PepMapC18; 300 μm × 5 mm, 5 μm particle size, Thermo Fisher) and separated on a 50 cm-long EasySpray column (ES803, Thermo Fisher) with a gradient of 2–35% acetonitrile in 5% dimethyl sulfoxide, 0.1% formic acid at 250 nl/min flow rate over 120 min. Eluted peptides were then analysed on an Orbitrap Fusion Lumos Tribrid platform (instrument control software v3.3). Data were acquired in data-dependent mode, with the advance peak detection enabled. Survey scans were acquired in the Orbitrap at 120 k resolution over a m/z range 400−1500, AGC target of 4e5 and S-lens RF of 30. Fragment ion spectra (MS/MS) were obtained in the Ion trap (rapid scan mode) with a Quad isolation window of 1.6, 40% AGC target and a maximum injection time of 35 ms, with HCD activation and 28% collision energy. The raw data files generated were processed using MaxQuant (Version 1.5.0.35), integrated with the Andromeda search engine. For protein groups identification, peak lists were searched against human database (UPR_Homo sapiens_9606_UP000005640_20200803.fasta) as well as list of common contaminants by Andromeda. Trypsin with a maximum number of missed cleavages of 2 was chosen. Oxidation (M) and Deamidiation (N, Q) were used as variable modifications while Carbamidomethylation (C) was set as a fixed modification. For quantitation SILAC labels K(4) R(6) and K(8)R(10) were selected. Protein and PSM false discovery rates were set at 0.01. Match between runs was applied. The Quantile normalisation was then applied, and the data was visualised in Perseus 1.6.15.0[138].

## Cytosol protein extraction

Cells were collected and washed with DPBS buffer (Life Technologies; Thermo Fisher Scientific, Inc.) once at 200 × g at room temperature for 4 min. The cells were lysed in RIPA buffer (Sigma-Aldrich Ltd.) with cOmplete EDTA-free protease inhibitor cocktail (Roche Molecular Systems, Inc.) and PhosStop™ phosphoatase inhibitor (Sigma-Aldrich Ltd.) on ice for 20 min then span down at 14,000 rpm at 4 °C for 20 min. The protein concentration was determined using Pierce BCA protein assay (Thermo Fisher Scientific, Inc.) or Qubit protein assay (Thermo Fisher Scientific, Inc.).

## RNA-sequencing analysis

Raw reads were cleaned using fastp v0.23.2 with default parameters[139]. Cleaned reads were confirmed high-quality using FastQC v0.11.9[140] and then mapped to the human genome hg38 using STAR v2.7.3a[141]. Mapped reads were quantified using featureCounts v2.0.0[142] and analysed using DESeq2 v1.34.0[143]. For differential gene expression analysis, the effect size was shrunk using the ashr method[144], p values were adjusted using the independent hypothesis weighting (IHW) method[145]. Significant differential gene expression was defined as |log2

Fold change| > 1 for upregulated genes and |log2 Fold change| < −1 for downregulated genes at an adjusted *p* value < 0.05.

## Functional enrichment analysis

Source annotations for gene ontology analysis and the approach for calculating differentially expressed genes and false discovery rates within computational suites were kept at 'default' unless specified. For RNA-seq data, GSEA was performed using fgsea v1.22.0[146]. The gene sets used in GSEA were obtained from the Molecular Signatures Database v7.5.1[147]. RNA-seq over-representation analyses were performed using g:GOSt[148,149].

## Statistical analysis

GraphPad Prism was used for statistical analysis by performing *t* test and two-way ANOVA tests followed by Bonferroni's corrected multiple comparisons between pairs of conditions. Unless otherwise indicated in the figure legends, we analysed three biological replicates for each data point in all graphs, and the level of significance was as follows: $P < 0.1$ (*), $P < 0.05$ (**), $P < 0.01$ (***), and $P < 0.001$ (****).

## Schematic illustrations

All schematic illustrations were created with Biorender.com

## The small molecule screening library

The screening library contained concentrated small molecule compounds with verified biochemical activity against their targets. Most of the compounds target epigenetic regulators with high specificity (Supplementary Table 5).

## Screening of the chemical compounds

The cells were grown in 96-well plates in standard growth medium with puromycin (1 μg/ml stock). Three technical replicates and three biological replicates were used for the screening. Cells were plated at a concentration of 10,000 cells in 100 μl of media per well in a 96-well plate. One day after plating the cells, the medium was changed to 90 μl standard growth medium supplemented with puromycin (0.5 μg/ml) and Activin A (10 ng/ml). On the same day, the compounds were added: first, 100× compound library dilutions were made, and 10 μL of 100× diluted chemical was added to each well to obtain 1000× final dilution of the compounds. Cells were then cultured with chemical compounds for five days with media change at day 0, day 2 and day 4 supplemented by fresh compounds. Each replicate was analysed using Celigo Image Cytometer (Nexcelom) and flow cytometry. Cells were lifted and dissociated into single cells with Trypsin. Details on the antibodies that were used for flow cytometry are listed in Supplementary Table 6. The cells were incubated with 0.5 μg/ml final concentration of conjugated antibodies in 1% BSA-PBS for 40 min on ice and washing was repeated as before. The cells were then suspended in 300 μL 1% BSA-PBS with DAPI (1:2000) for live/dead separation and kept on ice to be used for the flow cytometry analysis.

## Reporting summary

Further information on research design is available in the Nature Portfolio Reporting Summary linked to this article.

## Data availability

PDAC CSC ATAC-seq and RNA-seq data are available in the Gene Expression Ominbus (GEO) under accession code GSE244327 with no restrictions on data availability. The mass spectrometry proteomics data have been deposited to the ProteomeXchange Consortium via the PRIDE 150 partner repository with the dataset identifier PXD046026. ATAC-seq, RNA-seq and proteomic data from PDAC CSCs were generated for this study. The remaining data are available within the Article, Supplementary Information or Source Data file. Additional datasets used in this study have been published elsewhere and their references are provided throughout the Methods section. Source data are provided with this paper.

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

## Acknowledgements

We thank John Stingl and Ultan McDermott for discussions, and Ludovic Vallier for support with the initial stages of the project. We thank members of the Pauklin lab for assistance and help. This work was supported by the Cancer Research UK Career Development Fellowship Grant ID C59392/A25064 (S.P.), and Pancreatic Cancer UK Grant/Award Number: 2018RIF_03 (S.P.); The Kennedy Trust for Rheumatology Research Daphne Jackson Fellowship (C-H.C.), and the China Scholarship Council – University of Oxford Scholarship, SFF2122_CSCUO_1284663 (No. 202108330024) (S.D.). This work was supported by Cancer Research UK (CR-UK) grant number C5255/A18085 and the NIHR Oxford Biomedical Research Centre through the Oxford Centre for Early Cancer Detection and Cancer Research UK Oxford Centre (S.P.). The views expressed are those of the author(s) and not necessarily those of the NHS, the NIHR or the Department of Health.

## Author contributions

C.-H.-C., S.M., F.L., S.H., R.N., S.D., J.D., A.R. performed the experiments, data analyses and provided feedback on the manuscript. Z.S., R.F., U.O. provided research expertise, materials, facilities, and feedback on the manuscript. S.P. conceived the research, supervised the project, performed experiments and wrote the manuscript.

## Competing interests

The authors declare no competing interests.
