## [Peer Review File · Nature Communications]

Reviewers' Comments:

Reviewer #1:

Remarks to the Author:

The manuscript by Militi et al explores potential molecular mechanisms regulating pancreatic cancer (PDAC) cancer stem cells (CSC) maintenance. The Authors explore the role of secreted WNT ligands in CSC function and suggest a mechanism by which WNT ligands are regulated by RB-E2F via epigenetic regulation of the SAGA complex. Although the study is potentially interesting several critical issues with methods description, experimental design, and rigor were identified, specifically:

1. Fig 1a - How do the Authors define PDAC CSCs? and how were the CSCs isolated?
2. Fig 1c - the most enriched signatures in PDAC CSCs are associated with proliferation (S phase mitosis, positive regulation of proliferation) isn't that surprising? Do the Authors expect the CSCs to proliferate more than non-CSC? Please provide a list of factors in the identified WNT signature (Fig 1C), which of those factors were identified in secretome analysis?
3. The Authors cite (ref. 35) to stress that the WNT pathway is important for pancreas tumorigenesis. Indeed the cited papers show that b-catenin is essential in acinar-to-ductal metaplasia and that Dkk1 secreted negative regulator of WNT pathway blocks the initiation of early PanIN lesions formation. However, there is no evidence that WNT signaling would be essential for cancer maintenance or propagation where it would be expected CSC to play a role.
4. Fig 1g – Provide a source of data. I understand that the figure presents bulk tumor mRNA expression. What is the relation between secretome analysis in Fig 1a-e and mRNA expression in bulk cancer? Provide mRNA expression for CSCs and nonCSCs samples analyzed in Fig 1a-e. What was the rationale for analyzing only those 3 WNT ligands? Later in Figure 2 the Authors analyze WNT4 and WNT11 – please present the results for those ligands too. What is the rationale for comparing expression in tumor vs normal pancreas? Of course, a normal pancreas, of which ~95% of cells are digestive enzymes secreting acinar cells, is radically different but is that an argument for those ligands to play a role in CSC?
5. Fig 1h – Provide a source of data and details on how the analysis was performed. It is surprising to observe that in the data set analyzed here, the average 5-year survival for PDAC is over 20% and for WNT7A-low patients close to 50%. Include analysis for WNT4 and WNT11 ligands.
6. Fig 2A – Please provide ATAC-seq analysis for non-CSC and compare it with CSC results. Also, E2F-binding motifs are relatively abundant in promoter regions in fact, at promoter regions selected at random one could identify the E2F-binding motif. Therefore, full ATAC-seq analysis should be presented.
7. Fig 2C and Sup.Fig. 1a – E2F4 and E2F5 are abundantly expressed in all cells of the organisms. What is the rationale for investigating mRNA expression correlation between E2Fs with WNTs in a normal pancreas? Clarify which dataset from GTEx was utilized – Affymetrix or RNA-seq? E2Fs are regulated via RB1/RBL1/2 phosphorylation – what is the rationale for investigating mRNA expression levels?
8. Fig 2 j – Is there any evidence of CSC and non-CSC populations existing in H6C7 cells transformed with oncogenic Kras? If yes, perform the analysis on non-CSCs and compare it to CSC. H6C7 (HPDE6) cells are characterized by haploid loss of the Rb1 gene therefore, interpretation should be cautious.
9. Sup.Fig 1b – can the Authors define a significant correlation? $R=0.15$ to 0.22 is a rather poor correlation, isn't it? E2Fs are regulated via RB1/RBL1/2 phosphorylation – are there any changes in Rb phosphorylation levels.
10. Fig 2i and o – confirm Rb phosphorylation levels upon CDK4/6 inhibitor treatment.
11. Fig 3. E2Fs are regulated by RB1/RBL1/2, in turn regulated by phosphorylation – what is the

rationale for overexpressing those factors? Also, RB1 overexpression can cause senescence or cell death creating significant confounding factor in the interpretation of the results. Perform analysis into CSC and non-CSC.

12. Fig 3j. An expression correlation of $R=-0.15$ should not be considered indicative of any trend. The focus on RBL2 is not supported by data.

13. Fig 4. What is the mechanisms of WNT ligands stimulating proliferation of fibroblasts?

14. Fig. 6. The Authors suggest that GCN5 is potentially the most significant regulator of cancer stem cells in PDAC. That is a profound statement, given that GCN5 is not particularly essential for PDAC cell growth (see DepMap). Conform the results of pharmacological inhibition using the genetic depletion approach and confirm the specificity of the inhibitor.

Reviewer #2:

Remarks to the Author:

This is a novel and well performed study which demonstrates a novel role for RBL2/E2F/GCN5 in driving WNT ligand expression in pancreatic and breast cancer cells, and the implications of this upregulation for cancer stem cell maintenance, stromal cell proliferation and overall tumour proliferation, migration and chemoresistance. A role for GCN5 in co-operating with E2F to drive WNT expression is also revealed with potential therapeutic implications, although this was not investigated in vivo so potential toxicities remain unknown. Overall, the manuscript is well written, the data are clear and well controlled, and the Authors have used multiple in vitro model systems and orthogonal approaches from which to draw convincing conclusions. There are however some limitations.

1. The experiments are performed exclusively in in vitro model systems. Confirmation of at least one element of the findings in a genetically modified mouse model would have significantly increased the impact of the findings.

2. While the stated focus of the paper is on the biology of cancer stem cells, many of the analyses show correlations based on the whole tumour (e.g Fig 1e, 5a). Furthermore, almost all of the experimental manipulations (e.g overexpression and knockdown studies) are performed in the total cell population. Therefore, the findings of the study are more aligned with general implications for cancer biology, rather than effects on the cancer stem cell subset.

3. Authors should confirm that corresponding receptors for the WNT ligands investigated are present on target cells that are investigated (CSC's, stromal cells).

4. To directly link the paracrine effects of E2F-driven WNT expression/secretion on stromal cells, authors could delete/knockdown the relevant WNT ligands in E2F overexpressing cells or the corresponding receptors in stromal cells (e.g Fig 5k).

Specific comments:

Figure 1a, b. What is the cell line used in this analysis? How were CSCs separated from the non-CSC population and how were their CSC properties validated? What is the WNT pathway mutation status of these lines and do they express receptors for the WNT ligands identified?

Lines 149-151. Mechanisms of chemoresistance of PDACs is likely to be multi-modal and not simply attributable to CSCs. Statement should be tempered accordingly.

Figure 2a. Which cell line was used in the ATAC-seq analysis?

Lines 204-206: No references are provided in support of this statement.

Line 209. Fig 2j does not show E2F4 expression levels as stated in the text.

Line 213/214. Should just be Figure 2l (not l-m) and Figure 2n (not n-o)?

Supp Fig 2, Fig 7a. Source of these Chip-seq data should be provided the first time the data are cited.

Figure 3b, c. Authors do not state what the data are (mean mRNA expression?), how WNT expression was quantified (qPCR?) or how many time experiments were performed.

Line 249. Supp Fig 3e-f?

Line 265. Authors state there is stronger correlation of RBL2 expression with specific WNT ligands

but no data are shown.

Line 268. What is the data supporting low RBL2 expression in PDACs compared to other cell types?

Figure 3l. What are the RBL2+ cells?

Line 348: b-catenin misspelled.

Line 354: No reference provided to support statement.

Lines 370-374. No references provided to support statements.

Figure 5k. Which cells is b-catenin activity being measured in?

Fig 5k. Are WNT receptors expressed on stellate cells?

Line 475/Supp Fig 6D: Text refers to "mutations" in Rb family but data shown is for gains and deletions.

Line 518. Should "transcriptional activity" be just "transcription" of WNT growth factors?

Reviewer #3:

Remarks to the Author:

Stefania Mili et al.; pRb/RBL2-E2F1/4-GCN5 axis regulates cancer stem cell formation and G0 phase entry/exit by non-cell-autonomous mechanisms (MS No. NCOMMS-22-54010-T)

Summary of the key results

Stefania Mili et al. demonstrated the molecular biology underlying the regulation of cancer stemness by RB-E2F family and WNT signaling using pancreatic and breast cancer cell lines. They also claimed that depletion of cancer stem cells and recovery of sensitivity to anticancer drugs can be achieved by preventing their privileged entry into the G0 phase using GCN5 inhibitor.

While the authors provide detailed evidence for these findings, the characterization of cancer stem cells in pancreatic and breast cancer in the context of RB-E2F family and WNT signaling has already been extensively reported. Moreover, the use of cell cycle synchronization therapy by inhibiting G0 phase entry was demonstrated by Ito et al. in 2009 in *Nature* to induce cell cycle entry of quiescent leukemia stem cells using ascorbic acid, and Yano et al. in 2013 in *Clinical Cancer Research* demonstrated the recovery of sensitivity to cytotoxic agents by introducing FUCCI from the quiescent phase to the proliferative phase in gastric cancer stem cells, in vitro and in vivo.

In this paper, the authors focus on the G0 phase, which cannot be visualized by original FUCCI, using advanced FUCCI developed by them, and suggest that more imaging data should be presented to establish the link between RB-E2F family and WNT signaling.

One major concern in cancer stem cell research is that cancer stem cells are extremely rare compared to non-cancer stem cells. The authors proceed with their research without enriching cancer stem cells through cell sorting. Is there any meaningful comparison between 1% and 2%? In clinical practice, it falls within the margin of error. While the authors have performed TCGA analysis, there are concerns about the reproducibility of their results in their own cohort and the lack of evaluation as solid tumors such as in vivo or PDX.

Data & methodology

Suggested improvements: experiments, data for possible revision

The following are major and minor concerns that need to be addressed:

Major concern;

1. The description of the PDAC cells used in Figure 1a is missing. It is crucial to know whether cell lines, primary cultures from clinical samples, or both were used.
2. While the discussion primarily focuses on Panc1 cells, what was the reason for choosing Panc1 specifically? Also, why was MiaPaCa2, which has a higher CSC composition, not used?
3. Is the increase in WNT signaling in CSC sphere culture shown in Figure 2m-o really as low as 0.3%? Although the low value can be understood due to the mixture of CSC and non-CSC in the organoids, why is the value still extremely low in CSC sphere culture?
4. The development of advanced FUCCI in Figure 4 is excellent, especially depicting G0 phase, which is difficult. However, since mKate is ultimately a bright red color, it is difficult to distinguish it

from RFP. Please show evidence of mKate in FACS and its presence in G0 phase.

5. The CSC expressing FUCCI that formed spheres in Figure 4I should be in a quiescent state because CSC maintains cancer stemness during sphere formation. However, the picture shows more green colors for S/G2/M. Why is this?

6. It was mentioned that an increase in WNT signaling leads to entry into the proliferative phase. However, it is believed that CSC becomes non-CSC during the proliferative phase. Why can WNT signaling maintain CSC?

7. The study of CSC involves concentrating them from model cells using specific markers. Can CSC characteristics be concluded by only 3% of CSC present in bulk without cell sorting?

8. Did CSC decrease faster when enriched and WNT signaling was forcibly weakened than when they naturally decreased?

9. FOLFIRINOX and GnabP are representative chemotherapy for pancreatic cancer. Were treatments carried out using GEM, as well as 5-FU, OX, IRI, and PTX? When treated with various anticancer agents, does the CSC, which is resistant, increase, and the non-CSC, which is not resistant, decrease?

10. Does chemotherapy increase WNT signaling? Does chemotherapy increase WNT signaling in CSC in particular? Also, if resistance to chemotherapy becomes constant over the long term, does WNT signaling remain at high levels?

11. Merge brightfield and fluorescence images in Figure 5h. Fluorescence alone lacks objectivity.

12. The phenomenon of inducing cells from G0/G1 to S/G2/M phase and killing them with anticancer drugs demonstrated using FUCCI and spheres was first reported in Yano S. et al. CCR 2013. This should be cited.

13. Does WNT ligand increase when pancreatic cancer cells and fibroblasts are co-cultured?

14. In the study of breast cancer, why were the hormone receptor-positive and HER2-negative breast cancer cell lines HCC1500 and CAL-51 selected? Would the relationship between RB-E2F, WNT signaling, and CSC be similar in triple-negative breast cancer cell lines that are rich in CSCs, such as MDA-MB-231?

15. Were the typical chemotherapies for breast cancer, 5-FU, PTX, and DOX, used for treatment? Did treating with various chemotherapeutic agents result in an increase in CSCs with resistance and a decrease in non-CSCs without resistance?

16. The percentage of CSCs in Figure 6c is too low. It is not possible to discuss an increase in CSCs of less than 2%. Furthermore, it is difficult to assert that the positive cells are truly positive, given the clustering of the FACS population.

17. Only a reduction in the number of spheres is shown in Figure 6f, but what about their size? Photographs should also be included.

18. The results are based solely on in vitro cell line experiments for both pancreatic and breast cancer. Are the same results seen in vivo or in clinical samples? It is necessary to show similar results in a cohort other than TCGA.

Appropriate use of statistics, handling of uncertainty

No problems with statistical methods

Conclusions: robustness, validity, reliability

.

References: appropriate acknowledgements to previous studies?

The following references should be addressed;

Yano S, Tazawa H, Hashimoto Y, Shirakawa Y, Kuroda S, Nishizaki M, Kishimoto H, Uno F, Nagasaka T, Urata Y, Kagawa S, Hoffman RM, Fujiwara T. A genetically engineered oncolytic adenovirus decoys and lethally traps quiescent cancer stem-like cells in S/G2/M phases. *Clin Cancer Res*. 2013 Dec 1;19(23):6495-505. doi: 10.1158/1078-0432.

Chittajallu DR, Florian S, Kohler RH, Iwamoto Y, Orth JD, Weissleder R, Danuser G, Mitchison TJ. In vivo cell-cycle profiling in xenograft tumors by quantitative intravital microscopy. *Nat Methods*. 2015 Jun;12(6):577-85. doi: 10.1038/nmeth.3363.

Clarity, context: clarity of abstract and summary, adequacy of abstract, introduction, and

conclusions
No problem.

Point by point response

General Response

We would like to thank the Reviewers and the Editor for their helpful feedback and insightful suggestions. We provide a point-by-point response that addresses all the comments from the reviewers with extensive additional experimental data and textual changes to the revised manuscript. We have highlighted the changes within the revised manuscript in blue font colour, and added the additional research data into the manuscript figures and text as described in the point-by-point response.

Collectively, the feedback from the reviewers has further improved the manuscript and emphasises its timeliness for publication.

Reviewer #1 - E2F, CSCs (Remarks to the Author):

The manuscript by Militi et al explores potential molecular mechanisms regulating pancreatic cancer (PDAC) cancer stem cells (CSC) maintenance. The Authors explore the role of secreted WNT ligands in CSC function and suggest a mechanism by which WNT ligands are regulated by RB-E2F via epigenetic regulation of the SAGA complex. Although the study is potentially interesting several critical issues with methods description, experimental design, and rigor were identified, specifically:

1. Fig 1a - How do the Authors define PDAC CSCs? and how were the CSCs isolated?

Response 1.1. Thank you for this question.

The PDAC CSCs are a subpopulation of PDAC cells that have a stem cell-like state, allowing them to self-renew and give rise to tumours (Hermann et al., 2007; Li et al., 2007; Lonardo et al., 2011). This phenotype allows CSCs to give rise to the whole tumour with its entire cellular heterogeneity and thereby supports metastases formation and development of resistance to current cancer therapeutics. The existence of developmentally plastic CSCs has been discovered in the brain, breast, colon, oesophagus, liver, lung, ovarian, prostate, stomach and thyroid cancers, among others. In the case of PDAC, the first reports of cancer stem cells date back to 2007 (Hermann et al., 2007; Li et al., 2007). Since then, pancreatic CSCs have been shown to be involved in PDAC resistance to chemotherapy, displaying increased prevalence within the tumour after treatment with gemcitabine (Hermann and Sainz, 2018; Mueller et al., 2009). We have performed extensive experimental validation of the cells used in our study.

First, we performed a characterisation of marker expression in our PDAC cell lines by analysing markers that have been associated with the CSC cellular phenotype: EpCAM, CD44, CD24, PROM1/CD133, SSEA4, CXCR4, ABCG2. Cells were grown in 1) standard adherent conditions that does not enrich for CSCs, and 2) in 3D suspension as spheres from single cells that enriches for anoikis-resistant cells that have CSC characteristics. PDAC non-CSCs undergo anoikis as single cells in 3D non-adherent conditions but proliferate readily in adherent conditions. In contrast, CSCs are anoikis-

resistant and will give rise to spheres that are enriched for CSCs. We have isolated and characterised the CSCs in detail from different cell lines.

A13A cells showed an increase in CD44, PROM1/CD133, SSEA4 and CXCR4 expression in CSC spheres, whereas L3.6pl cells showed an increase in CD44, PROM1, SSEA4 and ABCG2 expression in CSC spheres (**Supplementary Fig. 1a**), thus indicating that several CSC markers attributed to the CSC populations are enriched in 3D spheres in our cell lines. To investigate the self-renewal capacity we sorted cells into CD133+/SSEA4+ and CD133-/SSEA4- subpopulations, and performed tumour sphere assays on these by using unsorted cells as a control. Marker positive cells showed a significantly higher number of spheres compared to both marker negative and unsorted cells (**Supplementary Fig. 1b**), indicating increased self-renewal capacity for CSC marker positive cells.

To investigate the chemoresistance of the CSC population in our PDAC cells, we treated the cells with chemotherapy reagent gemcitabine (GEM) and FOLFIRINOX (FOL) that are currently in clinical use as PDAC patient therapeutics. Gemcitabine and FOLFIRINOX treatment of PDAC cells for 5 days enriched for cells expressing CD133+/SSEA4+, CD44+/CD133+, OCT4+/CD133+, OCT4+/SSEA4+ from ~1% double positive cells to 80%, 60-75%, 18-20% and 22-25%, respectively (**Supplementary Figure 1c**). Triple positive cells increased from 0.5% to 45-60% upon Gemcitabine and FOLFIRINOX treatment (**Supplementary Figure 1d**). The increase of the CSC marker expressing cells occurs by eliminating most of the PDAC cells that do not express these CSC markers because 98-99.5% of cells treated with Gemcitabine and FOLFIRINOX die. This results in selective survival and enrichment of the rare OCT4-GFP+/CD133+/SSEA4+/EPCAM+ CSCs. Next, we also investigated the chemoresistance of cells grown in 3D sphere conditions that enriches of CSCs, and standard adherent 2D culture conditions. Cells grown as spheres have higher chemoresistance as shown by the higher survival of CSC spheres upon Gemcitabine and FOLFIRINOX treatment, whereas cells grown in standard adherent 2D culture conditions have higher chemosensitivity as indicated by the more drastic loss of viable cells (**Supplementary Figure 1e**).

Since the TGF β /Activin-SMAD2/3 signalling pathway regulates stem cell-like characteristics of CSCs (Feng et al., 2023; Lonardo et al., 2015), we also tested the impact of TGF β /Activin signalling on CSC resistance to currently used chemotherapeutics by treating the cells with the TGF β /Activin signalling inhibitor SB431542 in combination with Gemcitabine and FOLFIRINOX. Inhibition of TGF β /Activin signalling strikingly reduced the chemoresistance of PDAC cells as indicated by reduced numbers of CSC marker expressing cells and therefore the overall number of surviving PDAC cells (**Supplementary Figure 1f**). These results emphasize the crucial importance of TGF β /Activin signalling on CSC maintenance and their elevated chemoresistant characteristics.

We confirmed the relevancy of our CSC markers in primary human PDACs at single cell level by analysing single-cell RNA-sequencing data. This showed a subset of PDAC cells co-expressing OCT4+/CD133+/SSEA4+ markers, whereas cell trajectory modelling of these cells indicates their likely role in giving rise to the different cell populations in patient PDACs (**Supplementary Figure 1g-i**). Hence these CSC markers have *in vivo* co-expression and relevance in human PDACs, which support previous studies characterising these markers.

2. Fig 1c - the most enriched signatures in PDAC CSCs are associated with proliferation (S phase mitosis, positive regulation of proliferation) isn't that surprising? Do the Authors expect the CSCs to proliferate more than non-CSC? Please provide a list of factors in the identified WNT signature (Fig 1C), which of those factors were identified in secretome analysis?

Response 1.2.

Thank you for this comment. This point needs some clarification, and we also provide further experimental results on this aspect.

- Among the significantly enriched signatures in PDAC CSCs secretome analyses are terms associated with cell cycle regulation. This was indeed somewhat surprising, although it is known that various cells secrete growth factors that can impact cell proliferation either negatively or positively. It should be noted that the full list of statistically significantly enriched pathway terms also contains terms of cell cycle regulation and cell proliferation that do not hint to the possibility that CSCs could proliferate more (faster) than non-CSCs (see panel A below).
- Importantly, we have investigated the proliferation of CSCs vs non-CSCs, and these results do not indicate that CSCs are proliferating more rapidly. Rather, our extensive results show that

WNT signalling supports self-renewal capacity in CSCs and CSC factor expression (Fig. 4a-d, g), chemoresistance to gemcitabine (Fig. 4e-f,h), the transiency of G0 phase (Fig. l-m), and EMT/invasiveness (Fig. 5b-e,f). Hence we conclude that WNT ligands have more pronounced cellular effects on CSCs other than simply allowing them to proliferate faster, since E2Fs can impact various cellular processes (e.g. PMID: 11159908).

- In contrast, WNT signalling in pancreatic cancer associated fibroblasts it regulates various factors involved in cell proliferation (Supplementary Fig. 6j-k).

- It also needs to be emphasised that WNT ligand expression regulation is dynamic and can be increased or decreased depending on the external signalling. Our data provided evidence for the positive regulation of WNT ligand expression by E2F-GCN5, whereas E2F function can be switched to negative regulation of WNT ligand expression by E2F-RBL2 interaction. This indicates that WNT ligand expression in CSCs is also dynamic (see **Supplementary Figure 7n**).

3. The Authors cite (ref. 35) to stress that the WNT pathway is important for pancreas tumorigenesis. Indeed the cited papers show that b-catenin is essential in acinar-to-ductal metaplasia and that Dkk1 secreted negative regulator of WNT pathway blocks the initiation of early PanIN lesions formation. However, there is no evidence that WNT signaling would be essential for cancer maintenance or propagation where it would be expected CSC to play a role.

Response 1.3.

Thank you for highlighting this. Our current wording in the manuscript was the following: “PDAC is a cancer in which WNT signalling has an important pro-tumorigenic function ³⁵.”

The reference 35 is: Zhang, Y. et al. Canonical wnt signaling is required for pancreatic carcinogenesis. *Cancer Res* 73, 4909-4922 (2013). We agree with the reviewer that currently, there is no evidence that WNT signaling would be essential for cancer maintenance or propagation where it would be expected CSC to play a role. Hence, our research has significant novelty.

To avoid confusion in the text, we changed the description of the findings of ref. 35 as suggested by the reviewer: “β-catenin is essential in acinar-to-ductal metaplasia and Dkk1 secreted negative regulator of WNT pathway blocks the initiation of early PanIN lesions formation. However, so far, there is no evidence that WNT signaling would be essential for cancer maintenance or propagation where it would be expected CSC to play a role.”

We have now performed extensive experiments at the single-cell level in PDAC patient samples. Please see Response 1.4.

4. Fig 1g – Provide a source of data. I understand that the figure presents bulk tumor mRNA expression. What is the relation between secretome analysis in Fig 1a-e and mRNA expression in bulk cancer? Provide mRNA expression for CSCs and nonCSCs samples analyzed in Fig 1a-e. What was the rationale for analyzing only those 3 WNT ligands? Later in Figure 2 the Authors analyze WNT4 and WNT11 – please present the results for those ligands too. What is the rationale for comparing expression in tumor vs normal pancreas? Of course, a normal pancreas, of which ~95% of cells are digestive enzymes secreting acinar cells, is radically different but is that an argument for those ligands to play a role in CSC?

Response 1.4.

- The Fig 1g uses TCGA/GTEX data for pancreatic adenocarcinomas (TCGA: 179 tumour tissues, 4 normal control tissues; GTEX: 167 pancreas tissue). These are indeed bulk tumor mRNA expression.
- Fig 1b-c are secretome data that measures proteins secreted by CSCs or non-CSCs. The Fig 1e-f compare gene expression at the mRNA level in CSCs vs non-CSCs, thus complementing the secretome data. These show differences in CSCs vs non-CSCs in gene expression at the mRNA level, and indicate an elevated mRNA of WNT7A, WNT7B, WNT10B, WNT4, WNT11 and WNT3 in CSCs compared to non-CSCs.

The purpose of comparing in CSCs vs non-CSCs secretome analyses data (Fig 1a-c) to transcriptional changes in CSCs vs non-CSCs (Fig 1e-f), and bulk tumour samples (Fig 1g) was to gain insight if WNT ligands are regulated at protein secretion level (post-transcriptional to post-translational stages) or already at the transcriptional/gene expression stage. This gave us a first indication of the possible mechanism involved in WNT regulation since it suggested that it is related to transcriptional regulation of WNT ligands that then lead to changes also at the WNT ligand protein secretion.

- We have now included the expression data for all WNT ligands (new **Supplementary Figure 3a**). We showed only those 3 WNT ligands because there was no statistically significant change in gene expression the patient tumour samples and there was no correlation of these WNT ligand expression with higher patient mortality which suggested that WNT7A, WNT7B and WNT10B could play a particularly important role in PDAC whereas the other WNT ligands might have less functional importance for tumorigenesis.

- The bulk patient mRNA data was not meant to be an argument for those ligands to play a role in CSC, so we agree in this aspect with the reviewer. Rather, we hypothesised that if specific WNT ligands are involved in important pro-tumorigenic functions in PDAC, it would be hinted 1) by their higher expression in PDAC tumours compared to normal pancreatic tissue, and 2) the expression of these WNT ligands would show a positive correlation with higher patient mortality. We noticed from the expression data and correlation with patient mortality, only WNT7A, WNT7B, and WNT10A show both a higher expression in PDACs, and correlate with higher mortality. In contrast, the other WNT ligands do not show such patterns, implying that they have a less prominent or no supportive function in PDAC development. This data helped us to narrow down the candidate WNT ligands that might have a pro-tumorigenic effect in PDACs. We have included this in the manuscript for clarification.

- **Importantly, we have now included single-cell analyses in patient samples that allow us to investigate potential links between CSCs and WNT ligands.** We have included the following to our manuscript: “To investigate the ligand-receptor signalling that might occur in PDACs, we first performed the analysis of ligand-receptor pairs by predicting the pairs between ligands identified by our quantitative secretome identification and the expression of receptors in CSCs based on RNA-sequencing. We used single-cell RNA-sequencing expression data in different cell types detected in PDACs from patients. The tumours from PDAC patients contained epithelial cancer cells but also T-cells, myeloid cells, NK cells, fibroblasts, B-cells, endothelial cells, mast cells and neural cells (**Fig. 1h**).

The annotation of ligand-receptor interactions indicated the presence of CSC-specific signalling with other non-CSC cancer cell subpopulations (**Fig. 1i**). To model the crosstalk between CSCs and other cancer cell subpopulations in primary tumours, we paired the expression of the factors identified in our secretome studies, and their receptors on each of the cell types present in the PDAC patient tumours. The paracrine signalling between CSCs and subpopulations of cancer cells revealed extensive bidirectional crosstalk between the cancer cell populations, indicating that CSCs can provide autocrine signalling but also can affect the other non-CSC cancer cell subpopulations, and vice versa (**Fig. 1j**). Among the ligand-receptor pairs mediating the crosstalk are FZD3-LRP5/6-WNT7B, FZD5/6-LRP5/6-WNT7A/7B, FZD5-LRP5/6-WNT10A (**Fig. 1i-j**). The WNT ligands and their receptors FZD5 and FZD6 are expressed at variable levels in PDAC tumour in the cancer cell subpopulations but also in the cancer cell surrounding stromal cell populations and immune cells (**Fig. 1k-l**).”

5. Fig 1h – Provide a source of data and details on how the analysis was performed. It is surprising to observe that in the data set analyzed here, the average 5-year survival for PDAC is over 20% and for WNT7A-low patients close to 50%. Include analysis for WNT4 and WNT11 ligands.

Response 1.5.

The Fig 1h data uses TCGA/GTEX data as a source, the same data sets as Fig 1g. We would like to clarify the data that is visualised, to avoid confusion. The red solid line shows percent survival of patients

that are among the top 25% of patients with the highest expression of the WNT ligand. The blue solid line shows percent survival of patients that are among the bottom 25% of patients with the lowest expression of the WNT ligand. So the figure depicts the survival of patients with the 25% highest and 25% lowest expression of WNT ligand, it does not show the average survival of all patients (we moved this data to **Supplementary Figure 3c**).

We have now added a graph for the patient survival including all patients (n=177) (**Supplementary Figure 3b**). The solid black line shows percent survival considering all patients regardless of WNT7A expression. Based on this data the 5-year survival for PDAC patients (n=177) is below 20%.

6. Fig 2A – Please provide ATAC-seq analysis for non-CSC and compare it with CSC results. Also, E2F-binding motifs are relatively abundant in promoter regions in fact, at promoter regions selected at random one could identify the E2F-binding motif. Therefore, full ATAC-seq analysis should be presented.

Response 1.6.

Thank you for this feedback. We have now provided the ATAC-seq analysis for non-CSCs compared it to CSCs and added this to the manuscript: “Principal Component Analysis revealed clear separation of non-CSC and CSC samples, thus indicating differences in ATAC-sequencing peaks (**Fig. 2a**), and sample replicates clustered together showing good reproducibility (**Fig. 2b**). We identified 52049 peaks in non-CSCs and 64416 peaks in CSCs (**Fig. 2c**). The general distribution of ATAC-seq peaks was not different

in non-CSCs and CSCs since approximately a third of the peaks were located at gene promoters and a third were in the introns or exons in both non-CSCs and CSCs (Fig. 2d). Motif enrichment analyses of the total peaks indicated CTCF, BORIS, and the AP-1 family members (FRA1, FRA2, JUN, JUNB) as the top transcription factor motifs in non-CSCs and CSCs (Supplementary Fig. 3d-e), thus suggesting that there is not a major change among the most abundant transcription factor motifs. However, we uncovered 8522 differential ATAC-seq peaks between CSCs and non-CSCs, which were nearby 957 genes. ATAC-seq peaks nearby 352 genes were non-CSC specific, whereas peaks near 605 genes were CSC specific (Fig. 2e). WNT8A peak was non-CSC specific, whereas WNT10B was found in CSCs. The other ATAC-seq peaks at the TSS or in the close proximity were shared by non-CSCs and CSCs. ATAC-seq peaks at the TSS or in the close proximity in CSCs included WNT1, WNT2B, WNT3, WNT3A, WNT4, WNT5A, WNT5B, WNT6, WNT7A, WNT7B, WNT9A, WNT9B, WNT10A, WNT10B, WNT11, and WNT16 (Fig. 2e). Importantly, our transcription factor binding motif analyses uncovered the presence of E2F1 and E2F4 motifs in the ATAC-seq peaks at the proximal promoters of several WNT loci (Fig. 2f-g).

Rank	Motif	Name	P-value	log P-value	enrichment (Background)	# Target Sequences with Motif	% of Target Sequences with Motif	# Background Sequences with Motif	% of Background Sequences with Motif
1	ATAGTCACCACTCTGTGGCA	CTCF binding site - CTCF - C/EBP-beta/RelB/USF/ATF1/MyoD	1e-4208	-1.076e+04	0.0000	9687.0	18.64%	1430.3	3.89%
2	CCCCGGGCCCCCTCTGTGGC	BORIS binding site - CTCF - C/EBP-beta/RelB/USF/ATF1/MyoD	1e-2133	-7.236e+03	0.0000	1010.0	22.88%	323.6	6.59%
3	CATGACTCAATC	AP1 binding site - FRA1 - FRA2 - JUN - JUNB	1e-1258	-5.542e+03	0.0000	962.0	15.98%	255.8	5.16%
4	CATGACTCAATC	AP1 binding site - FRA1 - FRA2 - JUN - JUNB	1e-1047	-5.402e+03	0.0000	647.0	12.42%	187.9	3.79%
5	CATGACTCAATC	AP1 binding site - FRA1 - FRA2 - JUN - JUNB	1e-1042	-5.296e+03	0.0000	967.0	18.39%	370.9	7.64%
6	CATGACTCAATC	AP1 binding site - FRA1 - FRA2 - JUN - JUNB	1e-1000	-5.225e+03	0.0000	1067.0	24.46%	440.7	8.96%
7	CATGACTCAATC	AP1 binding site - FRA1 - FRA2 - JUN - JUNB	1e-1034	-5.124e+03	0.0000	1043.0	23.37%	370.2	10.87%
8	CATGACTCAATC	AP1 binding site - FRA1 - FRA2 - JUN - JUNB	1e-1029	-5.124e+03	0.0000	1043.0	23.37%	370.2	10.87%
9	CATGACTCAATC	AP1 binding site - FRA1 - FRA2 - JUN - JUNB	1e-1029	-5.088e+03	0.0000	1136.0	21.98%	522.0	16.94%
10	CATGACTCAATC	AP1 binding site - FRA1 - FRA2 - JUN - JUNB	1e-1187	-5.274e+03	0.0000	779.0	14.84%	292.1	5.98%
11	CATGACTCAATC	AP1 binding site - FRA1 - FRA2 - JUN - JUNB	1e-1143	-5.153e+03	0.0000	1233.0	23.30%	403.1	12.18%
12	CTGTTTACATA	AP1 binding site - FRA1 - FRA2 - JUN - JUNB	1e-2106	-5.070e+03	0.0000	1397.0	26.67%	320.4	14.67%
13	CCCCGGGCCCCCTCTGTGGC	BORIS binding site - CTCF - C/EBP-beta/RelB/USF/ATF1/MyoD	1e-1025	-5.263e+03	0.0000	1619.0	31.07%	974.6	19.72%
14	TTTTACTTA	FOXP1 binding site - FOXO1 - FOXO2 - FOXO3 - FOXO4 - FOXO5 - FOXO6 - FOXO7 - FOXO8 - FOXO9 - FOXO10 - FOXO11 - FOXO12 - FOXO13 - FOXO14 - FOXO15 - FOXO16 - FOXO17 - FOXO18 - FOXO19 - FOXO20 - FOXO21 - FOXO22 - FOXO23 - FOXO24 - FOXO25 - FOXO26 - FOXO27 - FOXO28 - FOXO29 - FOXO30 - FOXO31 - FOXO32 - FOXO33 - FOXO34 - FOXO35 - FOXO36 - FOXO37 - FOXO38 - FOXO39 - FOXO40 - FOXO41 - FOXO42 - FOXO43 - FOXO44 - FOXO45 - FOXO46 - FOXO47 - FOXO48 - FOXO49 - FOXO50 - FOXO51 - FOXO52 - FOXO53 - FOXO54 - FOXO55 - FOXO56 - FOXO57 - FOXO58 - FOXO59 - FOXO60 - FOXO61 - FOXO62 - FOXO63 - FOXO64 - FOXO65 - FOXO66 - FOXO67 - FOXO68 - FOXO69 - FOXO70 - FOXO71 - FOXO72 - FOXO73 - FOXO74 - FOXO75 - FOXO76 - FOXO77 - FOXO78 - FOXO79 - FOXO80 - FOXO81 - FOXO82 - FOXO83 - FOXO84 - FOXO85 - FOXO86 - FOXO87 - FOXO88 - FOXO89 - FOXO90 - FOXO91 - FOXO92 - FOXO93 - FOXO94 - FOXO95 - FOXO96 - FOXO97 - FOXO98 - FOXO99 - FOXO100	1e-1023	-5.213e+03	0.0000	1619.0	31.07%	973.1	19.87%
15	AAAGTAAACA	FOXP1 binding site - FOXO1 - FOXO2 - FOXO3 - FOXO4 - FOXO5 - FOXO6 - FOXO7 - FOXO8 - FOXO9 - FOXO10 - FOXO11 - FOXO12 - FOXO13 - FOXO14 - FOXO15 - FOXO16 - FOXO17 - FOXO18 - FOXO19 - FOXO20 - FOXO21 - FOXO22 - FOXO23 - FOXO24 - FOXO25 - FOXO26 - FOXO27 - FOXO28 - FOXO29 - FOXO30 - FOXO31 - FOXO32 - FOXO33 - FOXO34 - FOXO35 - FOXO36 - FOXO37 - FOXO38 - FOXO39 - FOXO40 - FOXO41 - FOXO42 - FOXO43 - FOXO44 - FOXO45 - FOXO46 - FOXO47 - FOXO48 - FOXO49 - FOXO50 - FOXO51 - FOXO52 - FOXO53 - FOXO54 - FOXO55 - FOXO56 - FOXO57 - FOXO58 - FOXO59 - FOXO60 - FOXO61 - FOXO62 - FOXO63 - FOXO64 - FOXO65 - FOXO66 - FOXO67 - FOXO68 - FOXO69 - FOXO70 - FOXO71 - FOXO72 - FOXO73 - FOXO74 - FOXO75 - FOXO76 - FOXO77 - FOXO78 - FOXO79 - FOXO80 - FOXO81 - FOXO82 - FOXO83 - FOXO84 - FOXO85 - FOXO86 - FOXO87 - FOXO88 - FOXO89 - FOXO90 - FOXO91 - FOXO92 - FOXO93 - FOXO94 - FOXO95 - FOXO96 - FOXO97 - FOXO98 - FOXO99 - FOXO100	1e-1021	-5.015e+03	0.0000	1782.0	36.52%	826.1	18.17%
16	TCTCCGGC	FOXP1 binding site - FOXO1 - FOXO2 - FOXO3 - FOXO4 - FOXO5 - FOXO6 - FOXO7 - FOXO8 - FOXO9 - FOXO10 - FOXO11 - FOXO12 - FOXO13 - FOXO14 - FOXO15 - FOXO16 - FOXO17 - FOXO18 - FOXO19 - FOXO20 - FOXO21 - FOXO22 - FOXO23 - FOXO24 - FOXO25 - FOXO26 - FOXO27 - FOXO28 - FOXO29 - FOXO30 - FOXO31 - FOXO32 - FOXO33 - FOXO34 - FOXO35 - FOXO36 - FOXO37 - FOXO38 - FOXO39 - FOXO40 - FOXO41 - FOXO42 - FOXO43 - FOXO44 - FOXO45 - FOXO46 - FOXO47 - FOXO48 - FOXO49 - FOXO50 - FOXO51 - FOXO52 - FOXO53 - FOXO54 - FOXO55 - FOXO56 - FOXO57 - FOXO58 - FOXO59 - FOXO60 - FOXO61 - FOXO62 - FOXO63 - FOXO64 - FOXO65 - FOXO66 - FOXO67 - FOXO68 - FOXO69 - FOXO70 - FOXO71 - FOXO72 - FOXO73 - FOXO74 - FOXO75 - FOXO76 - FOXO77 - FOXO78 - FOXO79 - FOXO80 - FOXO81 - FOXO82 - FOXO83 - FOXO84 - FOXO85 - FOXO86 - FOXO87 - FOXO88 - FOXO89 - FOXO90 - FOXO91 - FOXO92 - FOXO93 - FOXO94 - FOXO95 - FOXO96 - FOXO97 - FOXO98 - FOXO99 - FOXO100	1e-1021	-5.055e+03	0.0000	1874.0	36.07%	1178.2	25.49%
17	AAAGTAAACA	FOXP1 binding site - FOXO1 - FOXO2 - FOXO3 - FOXO4 - FOXO5 - FOXO6 - FOXO7 - FOXO8 - FOXO9 - FOXO10 - FOXO11 - FOXO12 - FOXO13 - FOXO14 - FOXO15 - FOXO16 - FOXO17 - FOXO18 - FOXO19 - FOXO20 - FOXO21 - FOXO22 - FOXO23 - FOXO24 - FOXO25 - FOXO26 - FOXO27 - FOXO28 - FOXO29 - FOXO30 - FOXO31 - FOXO32 - FOXO33 - FOXO34 - FOXO35 - FOXO36 - FOXO37 - FOXO38 - FOXO39 - FOXO40 - FOXO41 - FOXO42 - FOXO43 - FOXO44 - FOXO45 - FOXO46 - FOXO47 - FOXO48 - FOXO49 - FOXO50 - FOXO51 - FOXO52 - FOXO53 - FOXO54 - FOXO55 - FOXO56 - FOXO57 - FOXO58 - FOXO59 - FOXO60 - FOXO61 - FOXO62 - FOXO63 - FOXO64 - FOXO65 - FOXO66 - FOXO67 - FOXO68 - FOXO69 - FOXO70 - FOXO71 - FOXO72 - FOXO73 - FOXO74 - FOXO75 - FOXO76 - FOXO77 - FOXO78 - FOXO79 - FOXO80 - FOXO81 - FOXO82 - FOXO83 - FOXO84 - FOXO85 - FOXO86 - FOXO87 - FOXO88 - FOXO89 - FOXO90 - FOXO91 - FOXO92 - FOXO93 - FOXO94 - FOXO95 - FOXO96 - FOXO97 - FOXO98 - FOXO99 - FOXO100	1e-1021	-5.055e+03	0.0000	1774.0	34.47%	1093.6	22.20%
18	TCTCCGGC	FOXP1 binding site - FOXO1 - FOXO2 - FOXO3 - FOXO4 - FOXO5 - FOXO6 - FOXO7 - FOXO8 - FOXO9 - FOXO10 - FOXO11 - FOXO12 - FOXO13 - FOXO14 - FOXO15 - FOXO16 - FOXO17 - FOXO18 - FOXO19 - FOXO20 - FOXO21 - FOXO22 - FOXO23 - FOXO24 - FOXO25 - FOXO26 - FOXO27 - FOXO28 - FOXO29 - FOXO30 - FOXO31 - FOXO32 - FOXO33 - FOXO34 - FOXO35 - FOXO36 - FOXO37 - FOXO38 - FOXO39 - FOXO40 - FOXO41 - FOXO42 - FOXO43 - FOXO44 - FOXO45 - FOXO46 - FOXO47 - FOXO48 - FOXO49 - FOXO50 - FOXO51 - FOXO52 - FOXO53 - FOXO54 - FOXO55 - FOXO56 - FOXO57 - FOXO58 - FOXO59 - FOXO60 - FOXO61 - FOXO62 - FOXO63 - FOXO64 - FOXO65 - FOXO66 - FOXO67 - FOXO68 - FOXO69 - FOXO70 - FOXO71 - FOXO72 - FOXO73 - FOXO74 - FOXO75 - FOXO76 - FOXO77 - FOXO78 - FOXO79 - FOXO80 - FOXO81 - FOXO82 - FOXO83 - FOXO84 - FOXO85 - FOXO86 - FOXO87 - FOXO88 - FOXO89 - FOXO90 - FOXO91 - FOXO92 - FOXO93 - FOXO94 - FOXO95 - FOXO96 - FOXO97 - FOXO98 - FOXO99 - FOXO100	1e-1020	-5.046e+03	0.0000	1871.0	33.39%	1028.3	22.39%
19	TCTCCGGC	FOXP1 binding site - FOXO1 - FOXO2 - FOXO3 - FOXO4 - FOXO5 - FOXO6 - FOXO7 - FOXO8 - FOXO9 - FOXO10 - FOXO11 - FOXO12 - FOXO13 - FOXO14 - FOXO15 - FOXO16 - FOXO17 - FOXO18 - FOXO19 - FOXO20 - FOXO21 - FOXO22 - FOXO23 - FOXO24 - FOXO25 - FOXO26 - FOXO27 - FOXO28 - FOXO29 - FOXO30 - FOXO31 - FOXO32 - FOXO33 - FOXO34 - FOXO35 - FOXO36 - FOXO37 - FOXO38 - FOXO39 - FOXO40 - FOXO41 - FOXO42 - FOXO43 - FOXO44 - FOXO45 - FOXO46 - FOXO47 - FOXO48 - FOXO49 - FOXO50 - FOXO51 - FOXO52 - FOXO53 - FOXO54 - FOXO55 - FOXO56 - FOXO57 - FOXO58 - FOXO59 - FOXO60 - FOXO61 - FOXO62 - FOXO63 - FOXO64 - FOXO65 - FOXO66 - FOXO67 - FOXO68 - FOXO69 - FOXO70 - FOXO71 - FOXO72 - FOXO73 - FOXO74 - FOXO75 - FOXO76 - FOXO77 - FOXO78 - FOXO79 - FOXO80 - FOXO81 - FOXO82 - FOXO83 - FOXO84 - FOXO85 - FOXO86 - FOXO87 - FOXO88 - FOXO89 - FOXO90 - FOXO91 - FOXO92 - FOXO93 - FOXO94 - FOXO95 - FOXO96 - FOXO97 - FOXO98 - FOXO99 - FOXO100	1e-1020	-5.046e+03	0.0000	1806.0	32.47%	1481.5	13.94%
20	CCCCGGAGC	ELF1 binding site - E2F1 - E2F2 - E2F3 - E2F4 - E2F5 - E2F6 - E2F7 - E2F8 - E2F9 - E2F10 - E2F11 - E2F12 - E2F13 - E2F14 - E2F15 - E2F16 - E2F17 - E2F18 - E2F19 - E2F20 - E2F21 - E2F22 - E2F23 - E2F24 - E2F25 - E2F26 - E2F27 - E2F28 - E2F29 - E2F30 - E2F31 - E2F32 - E2F33 - E2F34 - E2F35 - E2F36 - E2F37 - E2F38 - E2F39 - E2F40 - E2F41 - E2F42 - E2F43 - E2F44 - E2F45 - E2F46 - E2F47 - E2F48 - E2F49 - E2F50 - E2F51 - E2F52 - E2F53 - E2F54 - E2F55 - E2F56 - E2F57 - E2F58 - E2F59 - E2F60 - E2F61 - E2F62 - E2F63 - E2F64 - E2F65 - E2F66 - E2F67 - E2F68 - E2F69 - E2F70 - E2F71 - E2F72 - E2F73 - E2F74 - E2F75 - E2F76 - E2F77 - E2F78 - E2F79 - E2F80 - E2F81 - E2F82 - E2F83 - E2F84 - E2F85 - E2F86 - E2F87 - E2F88 - E2F89 - E2F90 - E2F91 - E2F92 - E2F93 - E2F94 - E2F95 - E2F96 - E2F97 - E2F98 - E2F99 - E2F100	1e-1019	-5.098e+03	0.0000	1080.0	21.31%	495.6	12.28%

7. Fig 2C and Sup.Fig. 1a – E2F4 and E2F5 are abundantly expressed in all cells of the organisms. What is the rationale for investigating mRNA expression correlation between E2Fs with WNTs in a normal pancreas? Clarify which dataset from GTEx was utilized – Affymetrix or RNA-seq? E2Fs are regulated via RB1/RBL1/2 phosphorylation – what is the rationale for investigating mRNA expression levels?

Response 1.7. Thank you for this comment.

- We agree that a central regulatory mechanism for E2Fs is their regulation via RB1/RBL1/2 phosphorylation. This is often achieved by increased expression or genomic amplification of the Cyclin D or Cyclin E genes that lead to increased CDK4/6/2 activity. Similarly, RB1/RBL1/RBL2 can be deleted in cancers, which leads to increased E2F activity. However, there have been reported other regulatory routes through increased E2F expression in various cancers (e.g. PMID: 24023875, PMID: 14726656, PMID: 20026813, PMID: 34335934, PMID: 25440089, PMID: 33115417, PMID: 29072692, PMID: 30967995). Hence, the expression of E2Fs can impact its target gene expression, which was the rationale for looking at the expression levels of these factors and their correlation. We added this information in the manuscript in page 8.
- We have now provided the expression of WNTs, E2F1, E2F4, RB1/RBL1/2 in patient PDACs at the single cell level.

- The expression data and all correlation analyses are only supportive data in the manuscript, which we have moved to supplementary figures due to their lower importance. Nevertheless, the rationale for investigating mRNA expression correlation between E2F and WNT in normal pancreas was to see if E2Fs as positive regulators of gene expression will show a positive correlation with their expected target genes (WNT ligands). If E2Fs induce WNT expression, we hypothesised that we might see a positive correlation. This process can be involved in tissue homeostasis in the normal tissue by gene expression regulation, mediating cellular function, cell identity and differentiation, or proliferation upon tissue damage. We hypothesised that this can be counteracted by the Retinoblastoma family tumour suppressors that switch this positive signal off by transcriptional repression. Upon tissue transformation, E2F-RB circuitry might be deregulated to shift the balance of WNT expression to a more activated state (e.g. via Cyclin-CDK mediated phosphorylation of RBs, or reduced expression of RBs or elevated E2Fs, which all would have a similar effect on WNT ligand expression). We added this information in the manuscript in page 9.
- The datasets for gene expression were RNA-seq.

8. Fig 2 j – Is there any evidence of CSC and non-CSC populations existing in H6C7 cells transformed with oncogenic Kras? If yes, perform the analysis on non-CSCs and compare it to CSC. H6C7 (HPDE6) cells are characterized by haploid loss of the Rb1 gene therefore, interpretation should be cautious.

Response 1.8. Thank you for this interesting question.

To our knowledge, there are currently no published studies describing CSCs and non-CSC populations in H6C7 cells transformed with oncogenic Kras. Although it is of clear scientific interest in determining if Kras can be sufficient to drive the formation of cells with CSC characteristics, or are the additional hallmark mutations needed for the formation of CSCs in H6C7, we believe that this question would require a separate detailed study, since currently, such information is not available from other research publications.

9. Sup.Fig 1b – can the Authors define a significant correlation? $R=0.15$ to 0.22 is a rather poor correlation, isn't it? E2Fs are regulated via RB1/RBL1/2 phosphorylation – are there any changes in Rb phosphorylation levels.

Response 1.9.

Thank you for question. Generally, the correlations are considered as follows: $R < 0.25$ can be considered no correlation, $0.25 < R < 0.5$ is weak positive correlation, $0.5 < R < 0.75$ is moderate positive correlation and $R > 0.75$ is a strong positive correlation. The same can be considered for negative correlations. Hence, we agree that $R=0.15$ to 0.22 can be interpreted to have a very weak (poor) positive correlation around $R \sim 0.2$ even though p-values are statistically significant. This indicates that post-translational effects (phosphorylation) have a more important role in regulating RB-E2Fs.

For phosphorylation of Rb, please see the Response 1.10 below.

10. Fig 2i and o – confirm Rb phosphorylation levels upon CDK4/6 inhibitor treatment.

Response 1.10. We have now performed the western blot analyses (see Fig. 2i and Supplementary Fig 4i). These indicate that CDK4/6 inhibitor treatment results in the loss of pRb phosphorylation.

11. Fig 3. E2Fs are regulated by RB1/RBL1/2, in turn regulated by phosphorylation – what is the rationale for overexpressing those factors? Also, RB1 overexpression can cause senescence or cell death creating significant confounding factor in the interpretation of the results. Perform analysis into CSC and non-CSC.

Response 1.11. Indeed, E2Fs are regulated by RB1/RBL1/2 and these are in turn, regulated by phosphorylation. Our rationale was as follows: pancreatic cancer uses the phosphorylation mechanism to reduce the activity of RB1/RBL1/2, but also the reduction of RBL2 expression.

Similarly to the pRb locus, the RBL2 mutations are generally not found in PDAC patient samples (PMID: 34534465). In agreement with this, analysis of RBL2 expression in single-cell RNA-sequencing data indicated that it has a low expression in PDACs based on 16 primary tumours of PDAC patients compared to other cell types in the tissue (Fig. 3k-l). This indicates that RBL2 downregulation could be one means to reduce the tumour-suppressive effects of RBL2 in this cancer type besides CDK-mediated hyperphosphorylation. A similar observation has been made for myeloid cells in cancer, which exhibit a change in RB1 expression as a means to regulate its functional activity in the cells. In this case, the epigenetic silencing of RB1 regulates pathologic differentiation of myeloid cells in cancer (PMID: 23354483).

In our experimental rationale, the increase in RBL2 via overexpression can be used to investigate the function of RBL2 in cells since the even though some of the overexpressed RBL2 will be inactivated by phosphorylation, some of the overexpressed protein remains hypophosphorylated due to the significantly increased RBL2 protein pool in the cell. This hypophosphorylated RBL2 due to overexpression in the cells will be able to exert its regulatory function on E2Fs. The same reasoning was for RB1 overexpression. We understand the point of the reviewer that it is not an optimal experiment since at least some of the overexpressed RB1/RBL1/2 is inactivated by phosphorylation, but due to a high enough expression, we observed changes on gene expression, which were supported by the reverse experiments via gene knockdown.

12. Fig 3j. An expression correlation of $R=-0.15$ should not be considered indicative of any trend. The focus on RBL2 is not supported by data.

Response 1.12. We have now removed Fig 3k. Regardless of this correlation plot that looked at transcriptional levels in Fig. 3k, we have provided very extensive wet-lab experimental data throughout the manuscript that show a function of RBL2 and pRb/RB1 in PDAC CSCs. Hence, the focus on RBL2 is well justified by our other experimental results.

13. Fig 4. What is the mechanisms of WNT ligands stimulating proliferation of fibroblasts?

Response 1.12. We have now investigated the mechanisms further and added the following results to our manuscript: "We decided to further investigate the mechanisms of WNT ligands stimulating proliferation of fibroblasts. Based on data that has attributed cellular functions to fibroblast proliferation in other tissues, we investigated a panel of ten central regulators of cell proliferation (c-Myc, Cyclin D1, Cyclin E1, FGF2, FGFR2, EGFR, EGF, FGF1, FGFR1 and VEGFA) in pancreatic cancer associated fibroblasts. We found that stimulation of pancreatic CAFs with WNT7A/7B for 24h increased the expression of c-Myc, Cyclin D1, Cyclin E1, FGF2, FGFR2, EGFR, EGF, FGF1, FGFR1 and VEGFA, whereas IWR treatment reduced the expression of these genes (**Supplementary Figure 6j**). These results suggested that β -catenin might bind directly to the regulatory regions of these loci. Since these genes represent central regulators of cell proliferation, we analysed previously published β -catenin ChIP-seq, and ChIP-seq of its transcriptional coregulators TCF7L2 and LEF1. These revealed binding peaks of β -catenin ChIP-seq in the proximity of all ten genes, and the binding regions overlapped with either TCF7L2 and/or LEF1 (**Supplementary Figure 6k**). Next, we performed β -catenin ChIP-qPCR on the identified regions in pancreatic CAFs and uncovered enrichment of β -catenin protein

binding at these regions near the ten genes (**Supplementary Figure 6l**). To investigate, if these these β -catenin target genes mediate WNT/ β -catenin effects in pancreatic CAFs, we performed cell proliferation assays. Treating CAFs with WNT7A/7B in combination FGFR inhibitor Pemigatinib and EGFR inhibitor Erlotinib, slowed the proliferation of CAFs compared to the cells treated with only WNT7A/7B (**Supplementary Figure 6m**). In contrast, addition of FGF2 and EGF increased CAF proliferation compared to DMSO treatment (**Supplementary Figure 6m**)."

We have now also added the following to our manuscript: "To directly link the paracrine effects of E2F-driven WNT expression/secretion on stromal cells, we performed a CRISPR-KRAB knockdown of WNT7A and WNT7B (**Supplementary Fig. 6o**) and overexpressed either E2F4 or E2F1 in FG PDAC cells. Thereafter, we cocultured these FG cells with stellate cells expressing the β -catenin-dependent promoter-luciferase construct to measure WNT/ β -catenin signalling in stellate cells. Overexpression of E2F4 or E2F1 together with WNT7A/7B KD in PDAC cells reduced β -catenin-dependent promoter-luciferase activity compared to E2F OE cell coculturing (**Supplementary Fig. 6p**), indicating that WNT7A/7B mediate the E2F-driven WNT effects on stromal stellate cells. This also impacts the proliferation of the stromal cells, since WNT7A/7B KD in E2F4 OE or E2F1 OE PDAC cells slowed stellate cell proliferation upon coculturing (**Supplementary Fig. 6r-s**). We also investigated the corresponding WNT ligand receptors FZD1 and FZD6, which we found to be particularly highly expressed in stellate cells over the other FZD receptors. We performed a knockdown of FZD1 and FZD6 in stellate cells and found that this leads to reduced β -catenin-dependent promoter-luciferase activating in these cells upon coculturing with FG PDAC cells (**Supplementary Fig. 6r-s**). Furthermore, the FZD1 KD and FZD6 KD causes slower proliferation of stellate cells upon coculturing with FG PDAC cells (**Supplementary Fig. 6t**). Collectively, these data indicate that WNT7A/7B ligands and WNT receptors FZD1 and FZD6 mediate the paracrine signalling between pancreatic cancer cells and stellate cells in the tumour.

Specific comments:

14. Fig. 6. The Authors suggest that GCN5 is potentially the most significant regulator of cancer stem cells in PDAC. That is a profound statement, given that GCN5 is not particularly essential for PDAC cell growth (see DepMap). Conform the results of pharmacological inhibition using the genetic depletion approach and confirm the specificity of the inhibitor.

Response 1.10. Based on the screening experiments GCN5 was one of the factors that showed an effect on cancer stem cells, and hence, GCN5 is among the factors that contributes to regulating cancer stem cells in PDAC. We prefer to avoid statements that GCN5 is the most significant regulator of cancer stem cells in PDAC.

We have now performed the suggested genetic depletion of GCN5 by CRISPR-CAS9-KRAB mediated knockdown and added the results in the manuscript as follows: “Next, we performed genetic depletion of GCN5 to confirm the specificity of the pharmacological inhibitors of GCN5 by CRISPR-CAS9-KRAB mediated knockdown and investigated CSC sphere formation in three PDAC cell lines (Fig. 6i). GCN5 KD led to reduced CSC sphere numbers, indicating an effect of CSC self-renewal capacity as seen with GCN5 inhibitors. --- The knockdown of GCN5 also decreased H3K9ac abundance on the promoters of these Wnt loci (Supplementary Fig. 7i), and expression of WNT7A, WNT7B, WNT10A and WNT4 expression (Supplementary Fig. 7j).

Reviewer #2 - Pancreatic CSCs, epigenetics, mass-spec - (Remarks to the Author):

This is a novel and well performed study which demonstrates a novel role for RBL2/E2F/GCN5 in driving WNT ligand expression in pancreatic and breast cancer cells, and the implications of this upregulation for cancer stem cell maintenance, stromal cell proliferation and overall tumour proliferation, migration and chemoresistance. A role for GCN5 in co-operating with E2F to drive WNT expression is also revealed with potential therapeutic implications, although this was not investigated in vivo so potential toxicities remain unknown. Overall, the manuscript is well written, the data are clear and well controlled, and the Authors have used multiple in vitro model systems and orthogonal approaches from which to draw convincing conclusions. There are however some limitations.

1. The experiments are performed exclusively in in vitro model systems. Confirmation of at least one element of the findings in a genetically modified mouse model would have significantly increased the impact of the findings.

Response 2.1. Thank you for the comment. We have now performed extensive additional analyses by using surgically obtained tumour samples to compare the main findings in freshly isolated human tumours.

We have also used CSCs from primary tumours isolated from surgically removed PDAC tumours. We have now performed extensive additional experiments by using surgically isolated PDAC patient tumours, and added new experimental results to the manuscript: "Next, we studied the effects of GCN5 inhibition in PDAC patient-derived freshly isolated primary tumours (**Fig. 6m**). We performed multiplexed flow cytometry analyses on PDAC tumours derived from three PDAC patients by obtaining biopsies from fresh surgically removed tumour tissue. Surgically removed PDAC patient tumour samples from consenting patients were confirmed by histologists before further processing for 1) confirming the enrichment of CSC marker CD133, CD44 and SSEA4 expression upon Gemcitabine chemotherapy reagent treatment; 2) effects of WNT/ β -catenin inhibition with IWR and its effect on CSCs; 3) the effects of GCN5 inhibitors GSK4028 and L-Moses on CSCs as single treatment and in combination with currently used chemotherapy reagent Gemcitabine. The tumour samples were dissociated into single cells with a commercial verified kit, and CD45 capture MicroBeads were used for removing hematopoietic cells, which allowed us to focus our analyses on cancer cells that are CD45 negative. This also reduces false positive signals that could come from non-cancer cells expressing the markers. Firstly, we investigated the enrichment of CSC markers CD133, CD44 and SSEA4 upon Gemcitabine chemotherapy reagent treatment in tumours from three PDAC patients. The treatment of patient-derived PDAC cells with 0.5 μ M Gemcitabine for 3 days resulted in extensive cell death, whereas cells that survived the treatment were enriched for the expression of single CD133, CD44 and SSEA4 markers (**Supplementary Fig. 7k-m**), double positive CD133+/CD44+, CD133/SSEA4+ and CD44+/SSEA4+ cells (**Fig. 6n,p**), and CD133+/CD44+/SSEA4+ cells (**Fig. 6o,p**). This indicates the clinically used chemotherapy reagent Gemcitabine is able to eliminate non-CSCs but does not kill CSCs which are selectively surviving this treatment. Since WNT signalling inhibition reduced CSC self-renewal and chemoresistance of PDAC cell line-derived CSCs, we used co-treated patient cells with Gemcitabine and WNT/ β -catenin signalling inhibitor IWR. The co-treatment with IWR reduced the total live cells and the number of cells expressing CSC markers, suggesting that IWR decreases the

chemoresistance of CSCs to Gemcitabine (**Fig. 6n,p**). Lastly, we isolated the CSC subpopulation from freshly isolated patient primary PDAC tumours by using FACS-sorting of triple positive CD133+/CD44+/SSEA4+ cells and treated these with GCN5 inhibitors for 5 days followed by measuring live cell numbers. GCN5 inhibition decreased the number of live patient-derived CSCs while WNT inhibition with IWR also had a moderate effects on live CSCs compared to DMSO treated cells (**Fig. 6r**).

We also analysed the expression of WNT ligands from surgically removed PDACs from three patients by isolating CD133+/SSEA4+/EPCAM+ CSCs and comparing them to CD133-/SSEA4-/EPCAM+ non-CSCs (**Fig. 3f-g**). WNT7A, WNT7B, WNT10A and WNT4 indicated higher expression in CSCs compared to non-CSCs. Furthermore, CDK4/6 inhibition with Palbociclib led to a reduction in these WNT ligands in CSCs (**Fig. 3g**). Collectively, these data suggest that PDACs limit the activity of pRb/RBL2 mostly through inactivating phosphorylation events via CDKs.

Lastly, CSC sphere assays with freshly isolated patient cells indicated reduced CSC sphere formation upon WNT inhibitor IWR treatment compared to DMSO controls (**Supplementary Fig. 7n-o**). These results provide validation of our findings on WNT signalling between CSC subpopulation in PDAC cell lines and CSCs in freshly isolated patient tumours.

2. While the stated focus of the paper is on the biology of cancer stem cells, many of the analyses show correlations based on the whole tumour (e.g Fig 1e, 5a). Furthermore, almost all of the experimental manipulations (e.g overexpression and knockdown studies) are performed in the total cell population. Therefore, the findings of the study are more aligned with general implications for cancer biology, rather than effects on the cancer stem cell subset.

Response 2.2. Thank you for this question.

1. We performed extensive overexpression and knockdown studies specifically in CSCs in cell lines. Please see the full section with the title "E2Fs induce while RBs repress WNT ligand expression in pancreatic cancer stem cells" and the corresponding results in **Fig 3b-e** described in line 342-365.
2. We have now performed extensive additional analyses by using surgically obtained tumour samples to compare the main findings in freshly isolated human tumours as described in Response 2.1.
3. Importantly, we have now included single-cell analyses in patient samples that allow us to investigate potential links between CSCs and WNT ligands. We have included the following to our manuscript: "To investigate the ligand-receptor signalling that might occur in PDACs, we first performed the analysis of ligand-receptor pairs by predicting the pairs between ligands identified by our quantitative secretome identification and the expression of receptors in CSCs

based on RNA-sequencing. We used single-cell RNA-sequencing expression data in different cell types detected in PDACs from patients. The tumours from PDAC patients contained epithelial cancer cells but also T-cells, myeloid cells, NK cells, fibroblasts, B-cells, endothelial cells, mast cells and neural cells (**Fig. 1h**). The annotation of ligand-receptor interactions indicated the presence of CSC-specific signalling with other non-CSC cancer cell subpopulations (**Fig. 1i**). To model the crosstalk between CSCs and other cancer cell subpopulations in primary tumours, we paired the expression of the factors identified in our secretome studies, and their receptors on each of the cell types present in the PDAC patient tumours. The paracrine signalling between CSCs and subpopulations of cancer cells revealed extensive bidirectional crosstalk between the cancer cell populations, indicating that CSCs can provide autocrine signalling but also can affect the other non-CSC cancer cell subpopulations, and vice versa (**Fig. 1j**). Among the ligand-receptor pairs mediating the crosstalk are FZD3-LRP5/6-WNT7B, FZD5/6-LRP5/6-WNT7A/7B, FZD5-LRP5/6-WNT10A (**Fig. 1i-j**). The WNT ligands and their receptors FZD5 and FZD6 are expressed at variable levels in PDAC tumour in the cancer cell subpopulations but also in the cancer cell surrounding stromal cell populations and immune cells (**Fig. 1k-l**)."

4. We have now included single-cell analyses in patient samples that allow us to investigate the links between CSCs and fibroblasts mediated by WNT ligands: “The annotation of ligand-receptor interactions indicated the presence of CSC-specific signalling with other cell types found in patient PDACs (Fig. 5a). To model the crosstalk between CSCs and other cancer cell subpopulations in primary tumours, we paired the expression of the factors identified in our secretome studies, and their receptors on each of the cell types present in the PDAC patient tumours. We found that the ligands identified in CSC secretome also mediate paracrine signalling between CSCs and other cell types with the corresponding receptors such as fibroblasts, T-cells, macrophages and neural/nerve cells as suggested by their expression in

PDAC tumours isolated from patients (Fig. 5a). Among the ligand-receptor pairs mediating the crosstalk are particularly many between CSCs and fibroblasts that include WNT ligands and receptors. Among the ligand-receptor pairs mediating the crosstalk are FZD6-LRP5/6-WNT7B, FZD5/6-LRP5/6-WNT2, FZD5-LRP5/6-WNT5A (Fig. 5b). Therefore, CSCs are likely to be in functional crosstalk with several cell types in PDACs, including fibroblasts, which contribute to the characteristics and the CSC niche in the tumour. Collectively, the ligand-receptor crosstalk analyses indicated extensive bidirectional paracrine signalling between CSCs and other cell types in PDAC tumours, particularly fibroblasts, which included WNT ligands with their receptors. Therefore, we investigated the function of WNT signalling in more detail between CSCs and fibroblasts.”

5. We have done major changes to the manuscript. We have transferred all analyses that focus on bulk tumours into supplementary figures while keeping CSC data in main figures.

3. Authors should confirm that corresponding receptors for the WNT ligands investigated are present on target cells that are investigated (CSC's, stromal cells).

Response 2.3. Thank you for this comment.

There are 10 (FZD1-10) Frizzled receptors and Lrp5/6 co-receptors that function as WNT receptors. We have confirmed the expression of WNT receptors on A13A PDAC non-CSCs and CSCs by RNA-sequencing (Fig. 5c). These data show that A13A cells express FZD5, FZD6 and FZD1 most highly among FZD receptors, and they also express LRP5/6 co-receptors. We also investigated FZD and Lrp5/6 expression in stellate cells by comparing the expression of FZDs in PDAC non-CSCs and stellate cells by qPCR (Fig. 5d). Stellate cells express FZD1-8 at varying levels. Considering the levels in PDAC non-CSCs, the expression in stellate cells is highest for FZD1, FZD4 and FZD6. There is also moderate expression of FZD2 and FZD3 whereas the expression of FZD5 and FZD7 is low, and there is no expression of FZD8

and FZD9. The co-repressors LRP5 and LRP6 are highly expressed in stellate cells similarly to PDAC cells.

We also visualised the expression of FZD6 and FZD5 as dot plots by using single-cell RNA-seq data in patient PDAC samples (Fig. 1l). The data show different cell types in the patient sample, including cancer cells (dark yellow coloured subpopulation named “epithelial”) and stellate/fibroblast cells (blue coloured subpopulation). As in A13A cells, FZD5 has a lower expression in fibroblasts/stellate cells compared to cancer cells, but FZD6 is higher in fibroblasts/stellate cells in patient samples.

WNTs and FZD5/6 are expressed in all subpopulations of cancer cells in primary PDAC patient samples as shown by scRNA-seq analyses (Supplementary Fig. 2c).

4. To directly link the paracrine effects of E2F-driven WNT expression/secretion on stromal cells, authors could delete/knockdown the relevant WNT ligands in E2F overexpressing cells or the corresponding receptors in stromal cells (e.g Fig 5k).

Response 2.4. Thank you for this question. We have now performed the suggested experiments and added the following to our manuscript: "To directly link the paracrine effects of E2F-driven WNT expression/secretion on stromal cells, we performed a CRISPR-KRAB knockdown of WNT7A and WNT7B (**Supplementary Fig. 6o**) and overexpressed either E2F4 or E2F1 in FG PDAC cells. Thereafter, we cocultured these FG cells with stellate cells expressing the β -catenin-dependent promoter-luciferase construct to measure WNT/ β -catenin signalling in stellate cells. Overexpression of E2F4 or E2F1 together with WNT7A/7B KD in PDAC cells reduced β -catenin-dependent promoter-luciferase activity compared to E2F OE cell coculturing (**Supplementary Fig. 6p**), indicating that WNT7A/7B mediate the E2F-driven WNT effects on stromal stellate cells. This also impacts the proliferation of the stromal cells, since WNT7A/7B KD in E2F4 OE or E2F1 OE PDAC cells slowed stellate cell proliferation upon coculturing (**Supplementary Fig. 6r-s**). We also investigated the corresponding WNT ligand receptors FZD1 and FZD6, which we found to be particularly highly expressed in stellate cells over the other FZD receptors. We performed a knockdown of FZD1 and FZD6 in stellate cells and found that this leads to reduced β -catenin-dependent promoter-luciferase activating in these cells upon coculturing with FG PDAC cells (**Supplementary Fig. 6r-s**). Furthermore, the FZD1 KD and FZD6 KD

causes slower proliferation of stellate cells upon coculturing with FG PDAC cells (**Supplementary Fig. 6t**). Collectively, these data indicate that WNT7A/7B ligands and WNT receptors FZD1 and FZD6 mediate the paracrine signalling between pancreatic cancer cells and stellate cells in the tumour.

Specific comments:

Figure 1a, b. What is the cell line used in this analysis? How were CSCs separated from the non-CSC population and how were their CSC properties validated? What is the WNT pathway mutation status of these lines and do they express receptors for the WNT ligands identified?

Response 2.5. Thank you for this question.

- The cell line used in Figure 1a, b analysis was A13A cells there were obtained from the MSKCC from Prof Iacobuzio-Donahue (PMID: 15846069).
- We have added a new section to the manuscript that describes this part. CSC validation was conducted as follows.

Characterising CSCs for subsequent quantitative secretome analyses.

The PDAC CSCs are a subpopulation of PDAC cells that have a stem cell-like state, allowing them to self-renew and give rise to tumours (Hermann et al., 2007; Li et al., 2007; Lonardo et al., 2011). This phenotype allows CSCs to give rise to the whole tumour with its entire cellular heterogeneity and thereby supports metastases formation and development of resistance to current cancer therapeutics. The existence of developmentally plastic CSCs has been discovered in the brain, breast, colon, oesophagus, liver, lung, ovarian, prostate, stomach and thyroid cancers, among others. In the

case of PDAC, the first reports of cancer stem cells date back to 2007 (Hermann et al., 2007; Li et al., 2007). Since then, pancreatic CSCs have been shown to be involved in PDAC resistance to chemotherapy, displaying increased prevalence within the tumour after treatment with gemcitabine (Hermann and Sainz, 2018; Mueller et al., 2009). We have performed extensive experimental validation of the cells used in our study.

First, we performed a characterisation of marker expression in our PDAC cell lines by analysing markers that have been associated with the CSC cellular phenotype: EpCAM, CD44, CD24, PROM1/CD133, SSEA4, CXCR4, ABCG2. Cells were grown in 1) standard adherent conditions that does not enrich for CSCs, and 2) in 3D suspension as spheres from single cells that enriches for anoikis-resistant cells that have CSC characteristics. PDAC non-CSCs undergo anoikis as single cells in 3D non-adherent conditions but proliferate readily in adherent conditions. In contrast, CSCs are anoikis-resistant and will give rise to spheres that are enriched for CSCs. We have isolated and characterised the CSCs in detail from different cells lines.

A13A cells showed an increase in CD44, PROM1/CD133, SSEA4 and CXCR4 expression in CSC spheres, whereas L3.6pl cells showed an increase in CD44, PROM1, SSEA4 and ABCG2 expression in CSC spheres (**Supplementary Fig. 1a**), thus indicating that several CSC markers attributed to the CSC populations are enriched in 3D spheres in our cell lines. To investigate the self-renewal capacity we sorted cells into CD133+/SSEA4+ and CD133-/SSEA4- subpopulations, and performed tumour sphere assays on these by using unsorted cells as a control. Marker positive cells showed a significantly higher number of spheres compared to both marker negative and unsorted cells (**Supplementary Fig. 1b**), indicating increased self-renewal capacity for CSC marker positive cells.

To investigate the chemoresistance of the CSC population in our PDAC cells, we treated the cells with chemotherapy reagents Gemcitabine (GEM), FOLFIRINOX, and Nab-paclitaxel for 5 days. Gemcitabine, FOLFIRINOX and Nab-paclitaxel treatment of heterogeneous PDAC cells for 5 days enriched for cells expressing CD133+/SSEA4+, CD44+/CD133+, OCT4+/CD133+, OCT4+/SSEA4+ from ~1% double positive cells to 80%, 60-75%, 18-20% and 16-25%, respectively (**Supplementary Fig. 1c**). Triple positive cells increased from 0.5% to 33%, 21% and 14% upon Gemcitabine, FOLFIRINOX and Nab-paclitaxel treatment (**Supplementary Figure 1d**). The increase of the CSC marker expressing cells occurs by inducing cell death of a large majority of PDAC non-CSCs that do not express these CSC markers because 95-99% of PDAC cells treated with Gemcitabine, FOLFIRINOX, and Nab-paclitaxel die through apoptosis (**Supplementary Figure 1e**). This results in selective survival and enrichment of the rare CSCs expressing CD133, SSEA4, CD44 and OCT4. Collectively, CSCs survive and are increased upon treatment with anti-cancer agents whereas non-CSCs decrease due to apoptosis. Next, we also investigated the chemoresistance of cells grown in 3D sphere conditions that enriches of CSCs, and standard adherent 2D culture conditions. Cells grown as spheres have higher chemoresistance as shown by the higher survival of CSC spheres upon Gemcitabine and FOLFIRINOX treatment, whereas cells grown in standard adherent 2D culture conditions have higher chemosensitivity as indicated by the more drastic loss of viable cells (**Supplementary Fig. 1f**).

Since the TGF β /Activin-SMAD2/3 signalling pathway regulates stem cell-like characteristics of CSCs (Feng et al., 2023; Lonardo et al., 2015), we also tested the impact of TGF β /Activin signalling on CSC resistance to currently used chemotherapeutics by treating the cells with the TGF β /Activin signalling inhibitor SB431542 in combination with Gemcitabine and FOLFIRINOX. Inhibition of TGF β /Activin signalling strikingly reduced the chemoresistance of PDAC cells as indicated by reduced numbers of

CSC marker expressing cells and therefore the overall number of surviving PDAC cells (**Supplementary Fig. 1f**). These results emphasize the crucial importance of TGF β /Activin signalling on CSC maintenance and their elevated chemoresistant characteristics.

We confirmed the relevancy of our CSC markers in primary human PDACs at single cell level by analysing single-cell RNA-sequencing data. This showed a subset of PDAC cells co-expressing OCT4+/CD133+/SSEA4+ markers, whereas cell trajectory modelling of these cells indicates their likely role in giving rise to the different cell populations in patient PDACs (**Supplementary Fig. 1g-i**). Hence these CSC markers have in vivo co-expression and relevance in human PDACs, which support previous studies characterising these markers.

- CSCs separated from the non-CSC population as follows. The CSCs from the A13A PDAC cell line were firstly selected using CD133 magnetic beads and were anoikis-resistant and able to self-renew while being enriched for CSC marker expression when cultured in non-adherent tumour sphere-forming conditions (PMID: 36909530). In contrast, PDAC non-CSCs undergo anoikis as single cells in 3D non-adherent conditions but proliferate readily in adherent conditions.

- All cell lines used in this study express WNT ligand receptors confirmed by RNA-sequencing.

Lines 149-151. Mechanisms of chemoresistance of PDACs is likely to be multi-modal and not simply attributable to CSCs. Statement should be tempered accordingly.

Response 2.6. Thank you. We have now tempered this statement accordingly in line 151.

Figure 2a. Which cell line was used in the ATAC-seq analysis?

Response 2.7. We used A13A cells. We added this information in the manuscript in line 153.

Lines 204-206: No references are provided in support of this statement.

Response 2.8. Thank you. We have now added references PMID: 2985364 and PMID: 33816288 to this statement.

PMID: 2985364. Tiriác, H. et al. Organoid Profiling Identifies Common Responders to Chemotherapy in Pancreatic Cancer. *Cancer Discov* 8, 1112-1129 (2018).

PMID: 33816288. Verduin, M., Hoeben, A., De Ruyscher, D. & Vooijs, M. Patient-Derived Cancer Organoids as Predictors of Treatment Response. *Front Oncol* 11, 641980 (2021).

Line 209. Fig 2j does not show E2F4 expression levels as stated in the text.

Response 2.9. We have now corrected this by removing the reference to Fig. 2j.

Line 213/214. Should just be Figure 2l (not l-m) and Figure 2n (not n-o)?

Response 2.10. We have now corrected these.

Supp Fig 2, Fig 7a. Source of these Chip-seq data should be provided the first time the data are cited.

Response 2.11. We have now added the source of the ChIP-seq data on line 223 in the text where we first time cite these data.

Figure 3b, c. Authors do not state what the data are (mean mRNA expression?), how WNT expression was quantified (qPCR?) or how many time experiments were performed.

Response 2.12. Thank you for pointing this out. We have provided this data in the Figure 3b,c legend by adding: "The heatmaps in (b-c) show the normalised mean mRNA expression of three biological replicates analysed by qPCR."

Line 249. Supp Fig 3e-f?

Response 2.13. We have now corrected this.

Line 265. Authors state there is stronger correlation of RBL2 expression with specific WNT ligands but no data are shown.

Response 2.14. We have removed this sentence since this transcriptional data is not sufficiently strong to warrant its inclusion.

Line 268. What is the data supporting low RBL2 expression in PDACs compared to other cell types?
Response 2.15. This was based on comparing the expression of RBL2 in different cell population in PDAC patient samples based on scRNA-seq. While RBL2 is indeed more highly expressed in some cell types (e.g. T-cells, endothelial cells, monocytes) compared to cancer cells the expression is some other cell types (e.g. B-cells) is not as high. Hence, we have toned down this claim in the manuscript.

Figure 3I. What are the RBL2+ cells?

Response 2.16. The RBL2+ cells depict PDAC cancer cells that express RBL2 mRNA among the total cancer cell population. This gives an understanding of the WNT ligand expression levels in RBL2+ cancer cells compared to the total cancer cell population. The data indicates that among the RBL2+ cells there are proportionally more cells with no expression of WNT ligands than in the total cancer cell population.

Line 348: b-catenin misspelled.

Response 2.17. We have now corrected this.

Line 354: No reference provided to support statement.

Response 2.18. We have now added the reference to the sentence: "CSCs are able to metastasize due to their increased invasive capacity mediated by EMT (PMID: 22554795).

Lines 370-374. No references provided to support statements.

Response 2.19. Thank you. We have now added references that support the statements:

"Lastly, pancreatic cancers contain a range of different cell types, of which the most abundant are fibroblasts due to the highly desmoplastic and fibrosis-promoting conditions in the transforming pancreatic tissue (PMID: 30665410). Due to high fibrosis, the dense stroma provides a physical barrier for therapeutics to reach cancer cells (PMID: 34638513). Stromal cells also form an environment that can support cancer cell proliferation and CSC characteristics (PMID: 36010993). Hence, the formation of the dense desmoplastic stroma is important not only for understanding the cell signalling forming

the CSC niche but also for therapeutic accessibility (PMID: 30665410, PMID: 34638513, PMID: 36010993).

Figure 5k. Which cells is b-catenin activity being measured in?

Response 2.20. b-catenin activity was measured in stellate cells. We added this information in the text: “These treatment conditions also match β -catenin-dependent transcription as shown by increased luciferase activity in stellate cells (Fig. 5k).”

Fig 5k. Are WNT receptors expressed on stellate cells?

Response 2.21. There are 10 (FZD1-10) Frizzled receptors and Lrp5/6 co-receptors that function as WNT receptors. We have confirmed the expression of WNT receptors on A13A PDAC non-CSCs and CSCs by RNA-sequencing (Fig. 5c). These data show that A13A cells express FZD5, FZD6 and FZD1 most highly among FZD receptors, and they also express LRP5/6 co-receptors. We also investigated FZD and Lrp5/6 expression in stellate cells by comparing the expression of FZDs in PDAC non-CSCs and stellate cells by qPCR (Fig. 5d). Stellate cells express FZD1-8 at varying levels. Considering the levels in PDAC non-CSCs, the expression in stellate cells is highest for FZD1, FZD4 and FZD6. There is also moderate expression of FZD2 and FZD3 whereas the expression of FZD5 and FZD7 is low, and there is no expression of FZD8 and FZD9. The co-repressors LRP5 and LRP6 are highly expressed in stellate cells similarly to PDAC cells.

We also visualised the expression of FZD6 and FZD5 as dot plots by using single-cell RNA-seq data in patient PDAC samples (Fig. 1l). The data show different cell types in the patient sample, including cancer cells (dark yellow coloured subpopulation named “epithelial”) and stellate/fibroblast cells (blue coloured subpopulation). As in A13A cells, FZD5 has a lower expression in fibroblasts/stellate cells compared to cancer cells, but FZD6 is higher in fibroblasts/stellate cells in patient samples.

WNTs and FZD5/6 are expressed in all subpopulations of cancer cells in primary PDAC patient samples as shown by scRNA-seq analyses (Supplementary Fig. 2c).

Line 475/Supp Fig 6D: Text refers to “mutations” in Rb family but data shown is for gains and deletions.

Response 2.22. Thank you. We have changed the text from “mutations” to “gaining DNA sequences and deletions”.

Line 518. Should “transcriptional activity” be just “transcription” of WNT growth factors?

Response 2.22. Thank you, we have now corrected this.

Reviewer #3 - cell cycle, FUCCI (Remarks to the Author):

Stefania Miliati et. al.; pRb/RBL2-E2F1/4-GCN5 axis regulates cancer stem cell formation and G0 phase entry/exit by non-cell-autonomous mechanisms (MS No. NCOMMS-22-54010-T)

Summary of the key results

Stefania Miliati et al. demonstrated the molecular biology underlying the regulation of cancer stemness by RB-E2F family and WNT signaling using pancreatic and breast cancer cell lines. They also claimed that depletion of cancer stem cells and recovery of sensitivity to anticancer drugs can be achieved by preventing their privileged entry into the G0 phase using GCN5 inhibitor.

While the authors provide detailed evidence for these findings, the characterization of cancer stem cells in pancreatic and breast cancer in the context of RB-E2F family and WNT signaling has already been extensively reported. Moreover, the use of cell cycle synchronization therapy by inhibiting G0 phase entry was demonstrated by Ito et al. in 2009 in Nature to induce cell cycle entry of quiescent leukemia stem cells using ascorbic acid, and Yano et al. in 2013 in Clinical Cancer Research demonstrated the recovery of sensitivity to cytotoxic agents by introducing FUCCI from the quiescent phase to the proliferative phase in gastric cancer stem cells, in vitro and in vivo.

In this paper, the authors focus on the G0 phase, which cannot be visualized by original FUCCI, using advanced FUCCI developed by them, and suggest that more imaging data should be presented to establish the link between RB-E2F family and WNT signaling. One major concern in cancer stem cell research is that cancer stem cells are extremely rare compared to non-cancer stem cells. The authors proceed with their research without enriching cancer stem cells through cell sorting. Is there any meaningful comparison between 1% and 2%? In clinical practice, it falls within the margin of error. While the authors have performed TCGA analysis, there are concerns about the reproducibility of their results in their own cohort and the lack of evaluation as solid tumors such as in vivo or PDX.

Data & methodology

Suggested improvements: experiments, data for possible revision
The following are major and minor concerns that need to be addressed:

Major concern;

1. The description of the PDAC cells used in Figure 1a is missing. It is crucial to know whether cell lines, primary cultures from clinical samples, or both were used.

Response 3.1. Thank you for this question.

We used the CSCs from the PDAC A13A cell line in Figure 1a. We have now included this information in the manuscript. In later experiments we additionally use CSCs from additional PDAC lines as well as CSC population isolated from fresh surgically obtained primary PDACs from patients.

The PDAC CSCs are a subpopulation of PDAC cells that have a stem cell-like state, allowing them to self-renew and give rise to tumours (Hermann et al., 2007; Li et al., 2007; Lonardo et al., 2011). This phenotype allows CSCs to give rise to the whole tumour with its entire cellular heterogeneity and thereby supports metastases formation and development of resistance to current cancer therapeutics. The existence of developmentally plastic CSCs has been discovered in the brain, breast, colon, oesophagus, liver, lung, ovarian, prostate, stomach and thyroid cancers, among others. In the case of PDAC, the first reports of cancer stem cells date back to 2007 (Hermann et al., 2007; Li et al., 2007). Since then, pancreatic CSCs have been shown to be involved in PDAC resistance to chemotherapy, displaying increased prevalence within the tumour after treatment with gemcitabine (Hermann and Sainz, 2018; Mueller et al., 2009). We have performed extensive experimental validation of the cells used in our study.

First, we performed a characterisation of marker expression in our PDAC cell lines by analysing markers that have been associated with the CSC cellular phenotype: EpCAM, CD44, CD24, PROM1/CD133, SSEA4, CXCR4, ABCG2. Cells were grown in standard adherent conditions and from single cells in 3D suspension as spheres that enriches for anoikis-resistant cells that have CSC characteristics. PDAC non-CSCs undergo anoikis as single cells in 3D non-adherent conditions but proliferate readily in adherent conditions. A13A cells showed an increase in CD44, PROM1/CD133, SSEA4 and CXCR4 expression in CSC spheres, whereas FG cells showed an increase in CD44, PROM1, SSEA4 and ABCG2 expression in CSC spheres (**Supplementary Fig. 1A**), thus indicating that several CSC markers attributed to the CSC populations are enriched in 3D spheres in our cell lines. To investigate the self-renewal capacity we sorted cells into CD133+/SSEA4+ and CD133-/SSEA4- subpopulations, and performed tumour sphere assays on these by using unsorted cells as a control. Marker positive cells showed a significantly higher number of spheres compared to both marker negative and unsorted cells (**Supplementary Fig. 1B**), indicating increased self-renewal capacity for CSC marker positive cells.

To investigate the chemoresistance of the CSC population in our PDAC cells, we treated the cells with chemotherapy reagent gemcitabine (GEM) and FOLFIRINOX (FOL) that are currently in clinical use as PDAC patient therapeutics. Gemcitabine and FOLFIRINOX treatment of PDAC cells for 5 days enriched for cells expressing CD133+/SSEA4+, CD44+/CD133+, OCT4+/CD133+, OCT4+/SSEA4+ from ~1% double positive cells to 80%, 60-75%, 18-20% and 22-25%, respectively (**Supplementary Figure 1C**). Triple positive cells increased from 0.5% to 45-60% upon Gemcitabine and FOLFIRINOX treatment (**Supplementary Figure 1D**). The increase of the CSC marker expressing cells occurs by eliminating most of the PDAC cells that do not express these CSC markers because 98-99.5% of cells treated with Gemcitabine and FOLFIRINOX die. This results in selective survival and enrichment of the rare OCT4-GFP+/CD133+/SSEA4+/EPCAM+ CSCs. Next, we also investigated the chemoresistance of cells grown in 3D sphere conditions that enriches of CSCs, and standard adherent 2D culture conditions. Cells grown as spheres have higher chemoresistance as shown by the higher survival of cells upon Gemcitabine and FOLFIRINOX treatment, whereas cells grown in standard adherent 2D culture conditions have higher chemosensitivity as indicated by the more drastic loss of viable cells (**Supplementary Figure 1E**).

We confirmed the relevancy of our CSC markers in primary human PDACs at single cell level by analysing single-cell RNA-sequencing data. This showed a subset of PDAC cells co-expressing OCT4+/CD133+/SSEA4+ markers, whereas cell trajectory modelling of these cells indicates their likely role in giving rise to the different cell populations in patient PDACs (**Supplementary Figure 1F-H**). Hence these CSC markers have *in vivo* co-expression and relevance in human PDACs, which support previous studies characterising these markers.

2. While the discussion primarily focuses on Panc1 cells, what was the reason for choosing Panc1 specifically? Also, why was MiaPaCa2, which has a higher CSC composition, not used?

Response 3.1. Thank you for this question. In our secretome experiments and RNA-seq experiments described in Figure 1 we used A13A PDAC cell line that has been used also in other studies (PMID: 36909530). We also used L3.6pl, HPAFII and Panc1 for confirmation experiments of knockdown and overexpression effects, and ChIP-qPCR experiments. The A13A-FUCCI cells were used for cell cycle studies. We also used PDAC organoids to complement our experimental data on non-CSCs and CSCs from PDAC cell lines A13A, L3.6pl, HPAFII and Panc1. To avoid confusion, we have now carefully revised the manuscript text to include information on the cells that were used for each experiment.

We used A13A, L3.6pl, HPAFII and Panc1 cell lines for our study because we have characterised the CSC subpopulation more extensively in these cell lines and have performed RNA-seq experiments on A13A and L3.6pl cells. We also includes HPAFII and Panc1 cell lines which are widely used in the community. We included Panc1 as one of our cell lines since Panc1 shows a particularly high resistance to currently used chemotherapy reagent such as Gemcitabine. This was of interest to us since our experiments included aspects of finding possible alternative therapeutic regiments/compounds (e.g. GCN5 inhibitors) that would increase the susceptibility of Gemcitabine-resistant Panc1 cells to elimination. We could have used MiaPaCa2 as well, but in our case, the possibly higher CSC composition in MiaPaCa2 cells was not crucial because we are enriching for CSCs in our experiments by using the 3D suspension conditions. CSCs are anoikis-resistant and able to self-renew while being enriched for CSC marker expression when cultured in non-adherent tumour sphere-forming conditions (PMID: 36909530). In contrast, PDAC non-CSCs undergo anoikis as single cells in 3D non-adherent conditions but proliferate readily in adherent conditions. This method has been extensively used in the PDAC Cancer Stem Cell field to enrich for and temporarily expand the CSC population (e.g. PMID: 22056140, PMID: 18371365).

PMID: 22056140: Lonardo, E. et al. Nodal/Activin signaling drives self-renewal and tumorigenicity of pancreatic cancer stem cells and provides a target for combined drug therapy. *Cell stem cell* 9, 433-446 (2011).

PMID: 18371365: Hermann, P.C. et al. Distinct populations of cancer stem cells determine tumor growth and metastatic activity in human pancreatic cancer. *Cell stem cell* 1, 313-323 (2007).

3. Is the increase in WNT signaling in CSC sphere culture shown in Figure 2m-o really as low as 0.3%? Although the low value can be understood due to the mixture of CSC and non-CSC in the organoids, why is the value still extremely low in CSC sphere culture?

Response 3.1. Thank you for this question. We apologize, but we believe there has been a misunderstanding of the experiment and the interpretation of the Figure 2m-o results by the Reviewer 3. The WNT signal in CSC sphere culture is not 0.3%.

In the manuscript text (lines 211-219) we described these experiments. However, here we provide a further clarifying description of the experiment by using Figure 2m as an example. The data in Figure 2m shows ChIP-qPCR experiments. The y-axis is “% of input” which means the % of the initial chromatin material before the specific pulldowns with IgG control antibody, anti-pRb antibody, anti-RBL1 antibody or anti-RBL2 antibody. The “% of input” does not show percentage of cells but it is a common way of quantifying the antibody pulldown efficiency in ChIP experiments, and it shows specific pulldown compared to IgG control antibody pulldowns normalised to the total input

chromatin. The chromatin in Figure 2m experiments was isolated from cultured CSCs in the 3D sphere suspension condition, which allows for the enrichment and self-renewal of CSCs since non-CSCs will undergo anoikis in these conditions. The surviving CSCs that form the spheres are highly enriched as we have confirmed by flow cytometry analyses of CSC markers (~99% CD133/PROM1+ and 75% SSEA4+). We have now included further characterisation of the CSCs in the cells that were used for these experiments and others throughout the manuscript (**Supplementary Fig. 1**). In sum, these CSCs express some commonly used cancer stem cell markers (**Supplementary Fig. 1a**); they have strongly elevated self-renewal capacity following FACS sorting for the marker positive from a heterogeneous cancer cell population containing marker positive and negative cells (Supplementary Fig. 1B); these CSCs survive upon Gemcitabine and FOLFIRINOX treatments which kills non-CSCs in the heterogeneous cancer cell population (**Supplementary Fig. 1c-e**); cells expressing the same CSC markers (CD133+/SSEA4+/OCT4+) are detected also in freshly resected PDAC patient tumours analysed by RNA-sequencing (**Supplementary Fig. 1f**); and cell trajectory analyses in freshly resected PDAC patient tumours analysed by RNA-sequencing indicates their likely involvement in the formation of the different cancer cell populations in the tumour (Supplementary Fig. 1G-H). Collectively, the ChIP-qPCR experiment in Figure 2m showed that pRb, RBL1 and RBL2 bind to WNT7A, WNT4, WNT7B and WNT10A loci in CSCs.

We also did a similar ChIP-qPCR experiment in PDAC organoids which represents cells that mimic the cancer cell composition found in the tumour (**Fig. 2o**). Collectively, the ChIP-qPCR experiment in Figure 2o also showed that pRb, RBL1 and RBL2 bind to WNT7A, WNT4, WNT7B and WNT10A loci in organoids.

The difference between CSC cultures and PDAC organoids is that CSC culture contains a highly enriched population of cancer stem cells that can self-renew and they express CSC markers (~99% CD133/PROM1+ and 75% SSEA4+ in our experiments), whereas the proportion of CSCs in organoids is very low (~0.5-2% depending on the marker) as in the original tumour, and the bulk of the organoids are non-CSCs. These two systems allow us to compare CSCs vs non-CSCs in culture conditions that aim to mimic the tumour by using the state-of-the-art in the pancreatic cancer field.

4. The development of advanced FUCCI in Figure 4 is excellent, especially depicting G0 phase, which is difficult. However, since mKate is ultimately a bright red color, it is difficult to distinguish it from RFP. Please show evidence of mKate in FACS and its presence in G0 phase.

Response 3.1. Thank you for this question. RFP and mKate can be separated both on flow cytometry and confocal microscopy despite being relatively close.

We have added the following to the manuscript: “We verified the FUCCI system by flow cytometry analyses of the advanced FUCCI (RFP-hCdt1(30/120)_mAG-hGEM(1/110)_mKate2-p27(K-)) integrated into the AAVS1 locus in A13A PDAC CSCs show the signals of late G1 (hCdt+), G1/S transition (hCdt+/Geminin+), S/G2/M (Geminin+), and early G1 (hCdt-/Geminin-) (Supplementary Fig. 6c). The visualisation of hCdt and p27 shows the separation of hCdt+/p27- and hCdt+/p27+ cells (**Supplementary Fig. 6d**). Cells residing in the G0 phase are hCdt+/p27+ and they should be Ki67 negative, which is a marker for proliferating cells. The staining of FUCCI-CSCs for Ki67 shows a population of cells with very low Ki67, which marks G0 phase cells (**Supplementary Fig. 6e**). Gating of p27+ cells (highest signal for p27) shows that these cells are indeed negative for Ki67 signal (**Supplementary Fig. 6f**), thus confirming that the advanced FUCCI is able to distinguish G0 cells in PDAC.”

We have also included representative confocal microscopy images of PDAC Fucci-A13A cells to show the possibility of separating cells that are p27+/hCdt1+ or only hCdt1+.

5. The CSC expressing FUCI that formed spheres in Figure 4I should be in a quiescent state because CSC maintains cancer stemness during sphere formation. However, the picture shows more green colors for S/G2/M. Why is this?

Response 3.1. Thank you for this very interesting and intriguing question.

- The general understanding in the pancreatic cancer stem cell field is that cancer stem cells are not quiescent in the 3D sphere forming conditions but cancer stem cells with self-renew and proliferate to make up the spheres that consist of cancer stem cells (e.g. PMID: 22056140; PMID: 18371365). The FUCI-expressing cancer stem cells proliferate without losing their stem cell characteristics. Therefore, we would not expect CSCs to be quiescent, which is in agreement with the presence of green (S/G2/M phase) cells in the spheres. Indeed, when we study cancer stem cell marker expression in spheres we see that the large majority of cells in spheres express cancer stem cell markers up to 7 days after cell splitting to single CSCs (see our Response 3.1 for more experimental validation of CSCs in sphere culture conditions). Altogether, the results in Figure 4I are in agreement with our expectations that CSCs self-renew to make up the spheres, and this is in turn in agreement with other published PDAC CSC research papers that have used the 3D CSC sphere culture conditions (e.g. PMID: 22056140; PMID: 18371365).

There is currently still little insight to the cell cycle and G0 phase regulation in PDAC CSCs. Regarding the results on G0 phase in our CSCs, this is an exciting topic, and our data suggest that the CSCs are able to temporarily enter a shallow quiescent state but exit it after some time due to CSC culture conditions which contain growth factors that can support cell proliferation and CSC stemness. Since there are growth stimuli present for the cells in the media, and the cells also contain oncogenic KRAS mutations that support proliferation, CSCs temporarily enter and after a while exit this shallow G0 phase. We had included this aspect in our discussion section in line 647-660, where argued that “the E2F-RB-WNT circuitry could regulate the entry and exit from G0/cell dormancy or the depth of the „quiescent“ state. CSCs could temporarily reside in a “shallow quiescent” state, which has been termed “GAlert” or “primed”. This is different from a “deeply quiescent” state from which CSC can exit but this take longer than exiting the shallow or primed state. For instance, the primed state in Hematopoietic Stem Cells (HSCs) has been shown to be regulated by the level of CDK6 (refs 80-82). The strategy for leukaemia stem cells involves forcing the dormant cells to enter the cell cycle, which makes them more susceptible for chemotherapy-mediated elimination (ref 83).

80. van Velthoven, C.T.J. & Rando, T.A. Stem Cell Quiescence: Dynamism, Restraint, and Cellular Idling. *Cell stem cell* **24**, 213-225 (2019).
81. Rodgers, J.T. *et al.* mTORC1 controls the adaptive transition of quiescent stem cells from G0 to G(Alert). *Nature* **510**, 393-396 (2014).
82. Laurenti, E. *et al.* CDK6 levels regulate quiescence exit in human hematopoietic stem cells. *Cell stem cell* **16**, 302-313 (2015).
83. Vetrie, D., Helgason, G.V. & Copland, M. The leukaemia stem cell: similarities, differences and clinical prospects in CML and AML. *Nat Rev Cancer* **20**, 158-173 (2020).

6. It was mentioned that an increase in WNT signaling leads to entry into the proliferative phase.

However, it is believed that CSC becomes non-CSC during the proliferative phase. Why can WNT signaling maintain CSC?

Response 3.6. Thank you for this question.

- Since our results indicate that the cells in the sphere cultures express stem cell markers, and they can be expanded through self-renewal for 2-3 passages, it suggests that CSCs maintain their stemness characteristics for this time period. This notion is also supported by other research publications on PDAC CSCs. Importantly, the regulation of CSCs and the CSC characteristics upon proliferation in other cancer types (e.g. leukemic stem cells, and stem cells in gastrointestinal cancers in the gut) is likely to be different compared to PDACs, because these organs have a clear stem cell compartment that is naturally involved in continuously actively replenishing the differentiated cell types (e.g. dormant hematopoietic stem cells undergo asymmetric cell division, where one cell enters the cell cycle to gradually differentiate to myeloid/lymphoid progenitors and further to differentiated blood cells; or the stem cells in the intestinal crypts that also undergo asymmetric cell division and then one of these daughter cells differentiates to progenitor cells and further to differentiated cell types).
- We added the following to the manuscript: “Next we studied the mechanisms of WNT signalling maintaining CSCs. We investigated the expression of transcription factors that are involved in the maintenance of CSCs ⁵¹. We found a number of factors induced by WNT signalling and repressed by WNT inhibition by using qPCR. These include CSC maintenance factors *HOXA4*, *HOXA5*, *HOXA3*, *YAP1*, *MSI2*, *HIF1A*, *NOTCH2*, *MEIS1*, *OCT4*, *NES*, which are known as self-renewal factors in CSCs in various cancers (**Supplemental Fig. 4i**). Furthermore, the self-renewal factor loci are bound by β -catenin as shown by CHIP-sequencing analyses of published data, and CHIP-qPCR of β -catenin in A13A CSCs treated with WNT7A/7B ligands (**Fig. 4j-k**). Collectively, these data indicate that WNT/ β -catenin signalling directly regulates the expression of several well-known CSC self-renewal factors in pancreatic CSCs.”

7. The study of CSC involves concentrating them from model cells using specific markers. Can CSC characteristics be concluded by only 3% of CSC present in bulk without cell sorting?

Response 3.7. We apologize, but there has been a misunderstanding of our experiments, which we are glad to clarify. Importantly, we have not performed experiments on cells where only 3% of CSCs are in bulk cell population. Our CSCs that we have used for experiments are highly enriched. We have confirmed by flow cytometry analyses the cell population for the expression of CSC markers. For A13A

CSCs that were used in the main experiments that aim to study CSCs, we used ~99% CD133/PROM1+ and >75% SSEA4+ CSCs in all our experiments grown in sphere conditions. For other PDAC cell lines, we also confirmed their CSC marker expression and we used antibody-mediated bead capture for obtaining pure CSC populations for our experiments. Hence, our CSC cultures are highly pure and they are not contaminated by non-CSCs. We are not certain, where this misunderstanding has come from but we assure the reviewer that we have taken extensive efforts to use pure populations of CSCs in our secretome analyses, ChIP-qPCR experiments, western blots, ATAC-seq experiments and all the other experiments where we describe using CSCs.

It should be noted that besides CSCs, we have also used the PDAC organoid system in separate experiments. The CSCs and organoid experiments should not be confused with each other. Organoids represent the bulk PDAC population and we do not claim that these organoid experiments focus on CSCs. Instead, the organoids serve as a non-CSC comparison to our CSC experiments so that we would gain insight to both CSCs and non-CSCs. We hope this clarifies the general experimental setup and the cells used in our experiments.

8. Did CSC decrease faster when enriched and WNT signaling was forcibly weakened than when they naturally decreased?

Response 3.8. Thank you for this question. We have now addressed this interesting question and added the results to the manuscript as follows: “We also checked whether CSCs decrease faster when enriched and WNT signaling is forcibly weakened. For this, we sorted cells to CD133+/OCT4+/SSEA4+ CSCs by FACS and placed them in CSC sphere culture conditions. We measured the number of cells positive for CD133+/OCT4+/SSEA4+ CSC markers in A13A and L3.6pl lines upon WNT signalling inhibition with IWR and IWP2. Indeed, CSCs decreased when WNT signalling was inhibited by IWR and IWP2 for 3 days and further decreased at 7 days of treatment compared to enriched CSCs that maintained CSC marker expression better in both A13A and L3.6pl lines (**Supplementary Fig. 7c**).”

9. FOLFIRINOX and G nabP are representative chemotherapy for pancreatic cancer. Were treatments carried out using GEM, as well as 5-FU, OX, IRI, and PTX? When treated with various anticancer agents, does the CSC, which is resistant, increase, and the non-CSC, which is not resistant, decrease?

Response 3.9. Thank you for this question. We performed the experiments as follows: “To investigate the chemoresistance of the CSC population in our PDAC cells, we treated the cells with chemotherapy reagents Gemcitabine (GEM), FOLFIRINOX, and Nab-paclitaxel for 5 days. Gemcitabine, FOLFIRINOX and Nab-paclitaxel treatment of heterogeneous PDAC cells for 5 days enriched for cells expressing

CD133+/SSEA4+, CD44+/CD133+, OCT4+/CD133+, OCT4+/SSEA4+ from ~1% double positive cells to 80%, 60-75%, 18-20% and 16-25%, respectively (**Supplementary Fig. 1c**). Triple positive cells increased from 0.5% to 33%, 21% and 14% upon Gemcitabine, FOLFIRINOX and Nab-paclitaxel treatment (**Supplementary Figure 1d**). The increase of the CSC marker expressing cells occurs by inducing cell death of a large majority of PDAC non-CSCs that do not express these CSC markers because 95-99% of

PDAC cells treated with Gemcitabine, FOLFIRINOX, and Nab-paclitaxel die through apoptosis (**Supplementary Figure 1e**). This results in selective survival and enrichment of the rare CSCs expressing CD133, SSEA4, CD44 and OCT4. Collectively, CSCs survive and are increased upon treatment with anti-cancer agents whereas non-CSCs decrease due to apoptosis. Since the TGFβ/Activin-SMAD2/3 signalling pathway regulates stem cell-like characteristics of CSCs (Feng et., 2023; Lonardo et al., 2015), we also tested the impact of TGFβ/Activin signalling on CSC resistance to currently used chemotherapeutics by treating the cells with the TGFβ/Activin signalling inhibitor SB431542 in combination with Gemcitabine and FOLFIRINOX. Inhibition of TGFβ/Activin signalling strikingly reduced the chemoresistance of PDAC cells as indicated by reduced numbers of CSC marker expressing cells and therefore the overall number of surviving PDAC cells (Supplementary Fig. 1f). These results emphasize the crucial importance of TGFβ/Activin signalling on CSC maintenance and their elevated chemoresistant characteristics."

10. Does chemotherapy increase WNT signaling? Does chemotherapy

increase WNT signaling in CSC in particular? Also, if resistance to chemotherapy becomes constant over the long term, does WNT signaling remain at high levels?

Response 3.10. Thank you for this interesting question. We have now performed the requested experiments to answer these points.

We transfected A13A PDAC cells with M50 Super 8x TOPFlash and Renilla luciferase as an internal control. We treated the heterogeneous PDAC population with Gemcitabine, FOLFIRINOX and Nab-paclitaxel for 1, 3, 5 and 10 days and measured M50 Super 8x TOPFlash and Renilla luciferase signals at each time point. We observed an increase in the normalised WNT-dependent luciferase signalling at day 3 and day 5, whereas the prolonged treatment until day 10 resulted in a reduction of WNT-dependent luciferase signalling. These results allow us to make several conclusions by putting them in the context of our other experimental results described in the manuscript: firstly, since we did not observe an increase in WNT-dependent signalling after day 1 it suggests that WNT signalling increase is not a general stress response of the cells. Rather, the increase in the WNT signalling is caused by the selective survival of CSCs over non-CSCs, and CSCs have a higher WNT signalling as we had found in Figure 1. We have shown that chemotherapy treatment selectively kills non-CSCs while CSCs are more resistant to chemotherapy reagents than non-CSCs (See response 3.9). Secondly, when chemotherapy becomes constant over the longer term (day 10 in our case since all non-CSCs are killed by that time), the WNT signalling is reduced. The reason for the decrease of WNT signalling at day 10 is that CSCs were initially capable of self-renewal by progressing through the cell cycle but the constant chemotherapy treatment causes the accumulation of CSCs in the G0 phase (Supplementary Fig. 4d). We have shown that in G0 phase, WNT signalling activity is reduced via E2FpRb/RBL2 mediated repression of WNT ligands (Figure 4s), thus suggesting a mechanism for this observation.

11. Merge brightfield and fluorescence images in Figure 5h. Fluorescence alone lacks objectivity.

Response 3.11. We apologize but we did not capture the bright field images for overlapping the representative fluorescence images in Figure 5h. Fortunately, we have provided the quantification of the same cells as shown by the increase in total number of cells in Figure 5g. This provides an even more objective quantitative data for these experiments.

12. The phenomenon of inducing cells from G0/G1 to S/G2/M phase and killing them with anticancer drugs demonstrated using FUCCI and spheres was first reported in Yano S. et al. CCR 2013. This should be cited.

Response 3.12. We have now cited this interesting research in line 660-662.

It should be noted that Yano S. et al. CCR 2013 focused on human gastric cancer cells and they used genetically-engineered telomerase-specific oncolytic adenovirus for eliminating quiescent cancer

stem-like cells. On the other hand, our research focuses on pancreatic cancer and we identified small molecule inhibitors of epigenetic regulators.

13. Does WNT ligand increase when pancreatic cancer cells and fibroblasts are co-cultured?

Response 3.12. Thank you for this question. We have now performed the requested experiment: “We also checked whether WNT-dependent signalling increases when pancreatic cancer cells and fibroblasts are co-cultured. We transfected FG PDAC cells with M50 Super 8x TOPFlash and Renilla luciferase as an internal control, and placed the cells in the co-culture system together with fibroblasts. Co-culturing of cancer cells with fibroblasts for 5 days resulted in an increase in β -catenin dependent WNT signalling in cancer cells compared to culturing cancer cells without fibroblasts (**Supplementary Fig. 7a**).”

We also investigated the expression of WNT ligands in the different cell populations in tumours of PDAC patients by analysing single-cell RNA-sequencing data from primary PDACs (panel a-b below). These analyses indicate that both cancer cells and fibroblasts in PDACs express WNT ligands, but these are different for cancer cells and fibroblasts (panel c below). The fibroblasts express WNT5A and WNT2 most highly, whereas cancer cells express WNT7A, WNT7B and WNT10A most highly, but also WNT4, WNT11 and WNT5A (of note, WNT8A expression was below detection limit in PDAC tumour).

14. In the study of breast cancer, why were the hormone receptor-positive and HER2-negative breast cancer cell lines HCC1500 and CAL-51 selected? Would the relationship between RB-E2F, WNT signaling, and CSC be similar in triple-negative breast cancer cell lines that are rich in CSCs, such as MDA-MB-231?

Response 3.8. Thank you for this question. We selected HCC1500 (ref. 60) and Cal-51 (ref. 62), because they harbour RBL2 loss of heterozygosity (LOH) mutations induced by a frameshift in the gene product and have reduced expression of RBL2 protein (**Supplementary Fig. 6k**). This was particularly useful for our study that largely investigated the function of RBL2.

To investigate the relationship between RB-E2F and WNT signalling, we followed the advice of the reviewer to use MDA-MB-231: “Conditioned media from cells with WNT pathway inhibition by IWR and IWP2 led to reduced CSC sphere numbers compared to conditioned media alone while the addition of purified WNT ligands significantly increased CSC sphere formation, indicating that the WNT pathway promotes the self-renewal capacity of CSCs. Furthermore, conditioned media from RBL2 OE cells showed reduced supportive effects on CSC sphere formation compared to conditioned media from control cells (**Fig. 7f**), indicating that RBL2 reduces the secretion of CSC-supportive paracrine factors such as WNT ligands into the media.”

15. Were the typical chemotherapies for breast cancer, 5-FU, PTX, and DOX, used for treatment? Did treating with various chemotherapeutic agents result in an increase in CSCs with resistance and a decrease in non-CSCs without resistance?

Response 3.15. Thank you for this question. We have now performed this experiment and added the following new information to the manuscript: “We treated HCC1500, CAL-51 and MDA-MB-231 with typical chemotherapies for breast cancer, 5-FU, PTX, and DOX, for 5 days and measured CSC marker expressing cells CD133+/CD44+ by flow cytometry and also by measuring total cell survival upon these chemotherapy reagent treatments. The results indicated that the cells surviving 5-FU, PTX and DOX are strongly enriched for CSC markers CD133+/CD44+ whereas non-CSCs not expressing these markers are eliminated by apoptosis due to their lower resistance to 5-FU, PTX and DOX (**Supplementary Fig. 9f-g**).”

16. The percentage of CSCs in Figure 6c is too low. It is not possible to discuss an increase in CSCs of less than 2%. Furthermore, it is difficult to assert that the positive cells are truly positive, given the clustering of the FACS population.

Response 3.16. Thank you for this question. We apologize, but there seems to be a misunderstanding of this experiment, which we are glad to clarify. The experiment shown in Figure 6c represents the treatment of the heterogeneous PDAC cell line grown in standard culture conditions without the enrichment of CSCs. Hence, the population contains mainly non-CSCs since the cells are negative for the CSC markers. This % of CSC marker expressing cells that we see are expected, and the percentages are in agreement with previously published data in standard culture conditions (e.g. PMID: 22056140).

The aim of the experiment was to see if the small molecule GCN5 inhibitors (GSK4027 and L-Moses) impact non-CSCs or CSCs by looking at the percentage of CSC marker positive cells in this heterogeneous PDAC cell population. In the control treatment sample (DMSO) the steady-state % of double CSC-marker expressing cells is around 1-1.25%. We have used negative controls for the flow cytometry to determine where to place the gates for detecting positive cells. Upon small molecule inhibitor treatment for 5 days, the % of double CSC-marker expressing cells (OCT4+/SSEA4+) is reduced from 1.25% to 0.036% with GSK4027 and 0.021% with L-Moses. These results were repeated and the decrease was consistent in replicate experiments. This change indicates an approximately 30-fold decrease in CSC marker expressing cells for GSK4027 treatment and 50-fold decrease in CSC marker expressing cells for L-Moses. We further confirmed these findings by different experimental methods such as by using analysing the self-renewal of CSCs via tumour spheres formation (Figure 6f). In this experiment, we are investigating the self-renewal capacity and observe the consistent reduction of CSC self-renewal in CSCs from three different PDAC lines (Figure 6f). Given the use of multiple experimental systems, which provide results that support similar conclusions, we are confident in these results.

17. Only a reduction in the number of spheres is shown in Figure 6f, but what about their size? Photographs should also be included.

Response 3.17. Thank you for this question. We have now included the photographs and the sphere size measurements for Figure 6g-h. We observed a reduction in sphere size upon GCN5 inhibition with GSK4027 and L-Moses compared to the control compounds GSK4028 and D-Moses.

18. The results are based solely on in vitro cell line experiments for both pancreatic and breast cancer. Are the same results seen in vivo or in clinical samples? It is necessary to show similar results in a cohort other than TCGA.

Response 3.18. Thank you for this question.

We have now also used CSCs from primary tumours isolated from surgically removed PDAC tumours. We have performed extensive additional experiments by using surgically isolated PDAC patient tumours, and added new experimental results to the manuscript: “Next, we studied the effects of GCN5 inhibition in PDAC patient-derived freshly isolated primary tumours (**Fig. 6m**). We performed multiplexed flow cytometry analyses on PDAC tumours derived from three PDAC patients by obtaining biopsies from fresh surgically removed tumour tissue. Surgically removed PDAC patient tumour samples from consenting patients were confirmed by histologists before further processing for 1) confirming the enrichment of CSC marker CD133, CD44 and SSEA4 expression upon Gemcitabine chemotherapy reagent treatment; 2) effects of WNT/ β -catenin inhibition with IWR and its effect on CSCs; 3) the effects of GCN5 inhibitors GSK4028 and L-Moses on CSCs as single treatment and in combination with currently used chemotherapy reagent Gemcitabine. The tumour samples were dissociated into single cells with a commercial verified kit, and CD45 capture MicroBeads were used for removing hematopoietic cells, which allowed us to focus our analyses on cancer cells that are CD45 negative. This also reduces false positive signals that could come from non-cancer cells expressing the markers. Firstly, we investigated the enrichment of CSC markers CD133, CD44 and SSEA4 upon Gemcitabine chemotherapy reagent treatment in tumours from three PDAC patients. The treatment of patient-derived PDAC cells with 0.5 μ M Gemcitabine for 3 days resulted in extensive cell death, whereas cells that survived the treatment were enriched for the expression of single CD133, CD44 and SSEA4 markers (**Supplementary Fig. 7k-m**), double positive CD133+/CD44+, CD133/SSEA4+ and CD44+/SSEA4+ cells (**Fig. 6n,p**), and CD133+/CD44+/SSEA4+ cells (**Fig. 6o,p**). This indicates the clinically used chemotherapy reagent Gemcitabine is able to eliminate non-CSCs but does not kill CSCs which are selectively surviving this treatment. Since WNT signalling inhibition reduced CSC self-renewal and chemoresistance of PDAC cell line-derived CSCs, we used co-treated patient cells with Gemcitabine and WNT/ β -catenin signalling inhibitor IWR. The co-treatment with IWR reduced the total live cells and the number of cells expressing CSC markers, suggesting that IWR decreases the chemoresistance of CSCs to Gemcitabine (**Fig. 6n,p**). Lastly, we isolated the CSC subpopulation from freshly isolated patient primary PDAC tumours by using FACS-sorting of triple positive CD133+/CD44+/SSEA4+ cells and treated these with GCN5 inhibitors for 5 days followed by measuring live cell numbers. GCN5 inhibition decreased the number of live patient-derived CSCs while WNT inhibition with IWR also had a moderate effects on live CSCs compared to DMSO treated cells (**Fig. 6r**).

We also analysed the expression of WNT ligands from surgically removed PDACs from three patients by isolating CD133+/SSEA4+/EPCAM+ CSCs and comparing them to CD133-/SSEA4-/EPCAM+ non-CSCs (Fig. 3f-g). WNT7A, WNT7B, WNT10A and WNT4 indicated higher expression in CSCs compared to non-CSCs. Furthermore, CDK4/6 inhibition with Palbociclib led to a reduction in these WNT ligands in CSCs (Fig. 3g). Collectively, these data suggest that PDACs limit the activity of pRb/RBL2 mostly through inactivating phosphorylation events via CDKs.

Lastly, CSC sphere assays with freshly isolated patient cells indicated reduced CSC sphere formation upon WNT inhibitor IWR treatment compared to DMSO controls (**Supplementary Fig. 7n-o**). These results provide validation of our findings on WNT signalling between CSC subpopulation in PDAC cell lines and CSCs in freshly isolated patient tumours.

Appropriate use of statistics, handling of uncertainty

No problems with statistical methods

Conclusions: robustness, validity, reliability.

References: appropriate acknowledgements to previous studies?

The following references should be addressed;

Yano S, Tazawa H, Hashimoto Y, Shirakawa Y, Kuroda S, Nishizaki M, Kishimoto H, Uno F, Nagasaka T, Urata Y, Kagawa S, Hoffman RM, Fujiwara T. A genetically engineered oncolytic adenovirus decoys and lethally traps quiescent cancer stem-like cells in S/G2/M phases. *Clin Cancer Res*. 2013 Dec 1;19(23):6495-505. doi: 10.1158/1078-0432.

Chittajallu DR, Florian S, Kohler RH, Iwamoto Y, Orth JD, Weissleder R, Danuser G, Mitchison TJ. In vivo cell-cycle profiling in xenograft tumors by quantitative intravital microscopy. *Nat Methods*. 2015 Jun;12(6):577-85. doi: 10.1038/nmeth.3363.

Response 3.19. Thank you. We have added these citations to the manuscript. Yano et al (ref 95 in line 845) and Chittajallu et al (ref. 96 in line 846).

Reviewers' Comments:

Reviewer #2:

Remarks to the Author:

The Authors have provided a comprehensive response and included an extensive body of new data. Limitations of the study remain the lack of in vivo validation of findings, but most other issues raised have been satisfactorily addressed. Please provide the WNT pathway mutation of the cell lines used as requested in the original review.

Reviewer #3:

Remarks to the Author:

Stefania Mili and colleagues have molecularly elucidated that the properties of cancer stemness are controlled by the RB-E2F family and WNT signaling using pancreatic and breast cancer cell lines. Furthermore, they advocate for the depletion of cancer stem cells and the restoration of sensitivity to anticancer agents by preventing their privileged entry into the G0 phase using the GCN5 inhibitor.

Reviewer #3's assigned review focused on aspects related to FUCCI. While the authors have addressed all of Reviewer #3's comments, there is still a deficiency in terms of photographic evidence in the context of FUCCI imaging.

Indeed, representative photographs of FUCCI fluorescence are essential in the sections related to FUCCI.

1. Reiterating the point, the FUCCI fluorescence in Figure 4l is too faint within the spheroids. It is difficult to discern with the current presentation.
2. In Supplementary Figure 6g, is it accurate to state that exposure to chemotherapy leads to late G1 phase arrest? Exposure to many chemotherapeutic agents typically results in G2/M checkpoint arrest, leading to an increase in the S/G2 phase. Please clarify this discrepancy.
3. The photographs of spheroids in Figure 6h are still too small. The cell clusters appear too diminutive, making it difficult to perceive them as a collective spheroid.
4. It is important to note that the study did not explicitly demonstrate the WNT signal in the context of carcinogenesis, so it might be an overinterpretation to extrapolate extensively on this point.

Reviewer #4:

Remarks to the Author:

This is a quite thorough and well executed study demonstrating a role for E2F transcription factors in driving expression of Wnt ligands in pancreatic and breast cancer cells with the potential for autocrine and paracrine modulation of tumor growth and progression. The extensive bioinformatic analysis is rationally and appropriately applied using mostly established methods. However, in the interest of transparency, there are some critical missing details and labeling errors that need to be addressed prior to publication as well as some tempering of interpretation in these and other aspects of the work as enumerated below.

Concerns related to data analysis:

1. There is insufficient methodological detail presented regarding analysis of ligand-receptor pairs. What package (e.g. within Seurat, or through manual annotation of gene ontologies and cell types, etc.) was used for this analysis and the generation of corresponding circus plots? Similarly, the source annotations for gene ontology analysis should be specified and the specific approach for calculating differentially expressed genes and false discovery rates within computational suites should be specified or at least denoted as 'default'. Critically, without functional analysis of source and target of these potential signaling axes in vivo they should be qualified as "putative or implied"

paracrine mechanisms, or something similar. That is, the analytical models do not show extensive cellular crosstalk. They suggest or imply it.

2. Genes used for manual cell type classification should be indicated.

3. Supplementary Figure 1 G-I contains panels that are difficult to understand. What does the table indicate; how was it calculated/tabulated? What method was employed for Cell Trajectory analysis? Was the branchpoint designated as start? If so, why does pseudotime predict 0 at a distal position? The key in panel I does not match the plotted colors and these are not referenced coherently to the plot in G.

Minor:

4. Figure 1m is mislabeled in the figure legend.

5. Although indicated in the methods, line 224 should also reference the study from which single cell gene expression data is derived to clarify the analysis is carried out on previously published data.

6. Fig. 2h does not appear to be referenced in the text.

Concerns related to other aspects of the work:

7. Extensive characterization of a subpopulation of cells with distinct marker and transcriptional profiles, and functional outputs, is described in Supplemental Fig. 1 but as there is still some controversy surrounding the potential stem like activity in other sub-populations in multiple cancer types, it would be helpful to delineate/clarify the CSC definition used here in the figure legend by adding the marker designation to the figure title or details. Also, at least one representative primary data figure of sphere growth should accompany the figure to inform what was counted in the bar graphs.

8. Given that only Wnt8A and Wnt10B show subpopulation specific binding of E2Fs at their promoters, what is the evidence that those two Wnts specifically are required for activation of functional differences in the opposing compartments? The differential expression of those Wnts is notably absent from Fig. 1 L-M. Does the conclusion that effects are "non-cell-autonomous" as indicated in the title depend on this? This should be discussed. For instance, is a distinction between "cell autonomous" and "autocrine" intended? Is a paracrine target of CSC signaling clearly delineated? Related to this, various subtypes of fibroblasts exist within the PDAC microenvironment. From the scRNA-seq data in Fig S1, it would be informative to determine what subtype of CAF is most responsible for WNT crosstalk between CSCs and fibroblasts.

9. Fig. 2I: Typically, phosphorylated pRb would be compared to total pRb levels.

10. Line 154-157: The authors argue that the increase in CSC markers after treatment with various therapies is a result of cell-death of non-CSC cells. Is the possibility that exposure to treatments causes non-CSC cells to adopt a CSC-like state excluded?

11. Line 229-233: The authors claim PDAC_C1 cluster visualized in Fig S2 contains higher levels of CSC markers and elevated WNT family member expression however this difference appears to be quite modest compared to other clusters (i.e. PDAC_C3). Statistical analysis of enrichment of these CSC/WNT genes in the C1 cluster should be performed to demonstrate this difference.

12. Fig 2M: The mechanism by which Palbociclib causes decreased WNT expression is unclear. Namely, are the effects due to decreased WNT expression by CSCs or a depletion of the CSC population?

13. Fig 8S d.: KM plots should be stratified by clinical or PAM50 subtype as CDk4/6 inhibitors + endocrine therapy is standard of care for HR+ tumors in the context of metastatic disease and may confound the analysis.

14. Page 22: MCF-7 are more luminal than a basal subtype (such as HCC1500 and Cal-51), therefore there are multiple additional biological differences besides RBL2 expression that might explain the differences in Wnt expression.

Point by point response

General Response

We would like to thank the Reviewers and the Editor for their helpful feedback and insightful suggestions. We provide a point-by-point response that addresses all the comments from the reviewers with additional experimental data and textual changes to the revised manuscript. We have highlighted the changes within the revised manuscript in blue font colour, and added the additional research data into the manuscript figures and text as described in the point-by-point response.

Collectively, the feedback from the reviewers has further improved the manuscript and emphasises its timeliness for publication.

REVIEWER COMMENTS

Reviewer #2 - (also asked to comment on R1)

(Remarks to the Author):

The Authors have provided a comprehensive response and included an extensive body of new data. Limitations of the study remain the lack of in vivo validation of findings, but most other issues raised have been satisfactorily addressed. Please provide the WNT pathway mutation of the cell lines used as requested in the original review.

Response 2.1. Thank you for the positive feedback. We have now provided the mutation information on the cell lines used in Materials and Methods section. However, based on the available information WNT pathway mutations have not been listed for the used cell lines. We added the following information on the known mutations in the cell lines: "The L3.6pl and FG cells were provided by Isaiah J. Fidler through the cell line bank at the University of Texas MD Anderson Cancer Center. L3.6pl and FG cell lines have a KRAS p.Gly12Asp (c.35G>A) genetic background. The A13A cells were provided by Christine Iacobuzio-Donahue at The Memorial Sloan Kettering Cancer Center. The genetic background of A13A cell line is copy number gain of GATA-6 and cTAGE1; KRAS G12V; TP53 WT; SMAD4 WT. PANC1 and HPAFII cells were obtained from ATCC. PANC1 has a homozygous deletion of CDKN2A; Mutations: KRAS p.Gly12Asp (c.35G>A) heterozygous; TP53 p.Arg273His (c.818G>A) homozygous. HPAFII has mutations: CDKN2A p.Arg29_Ala34del (c.85_102del18) heterozygous; KRAS p.Gly12Asp (c.35G>A) heterozygous; TP53 p.Pro151Ser (c.451C>T) homozygous. MDA-MB-231 has a CDKN2A and CDKN2B Homozygous deletion; Mutations: KRAS p.Gly13Asp (c.38G>A) Heterozygous; BRAF p.Gly464Val (c.1391G>T) heterozygous; TERT c.1-124C>T (c.228C>T); TP53 p.Arg280Lys (c.839G>A) homozygous. CAL-51 has a PIK3CA p.E542K mutation. HCC1500 has PIK3CA c.3075C>T and c.*155C>T heterozygous mutations. MCF7 cell line has a homozygous deletion of CDKN2A and mutations: GATA3 p.Asp336Glyfs*17 (c.1006dupG) heterozygous; PIK3CA p.Glu545Lys (c.1633G>A) heterozygous; TP53 wt. T47D has mutations: PIK3CA p.His1047Arg (c.3140A>G) heterozygous; TP53 p.Leu194Phe (c.580C>T) homozygous. The HPDE6c7 cell line, with normal KRAS, *TP53*, *c-myc*, and *p16^{INK4A}* genotypes, was purchased from Kerfast (USA) and maintained in keratinocyte serum-free medium supplemented with human recombinant epidermal growth factor and bovine pituitary extract (Thermo Fisher Scientific, 17005042). HPDE6c7+KRAS (H6C7) has a KRAS (G12V) that is expressed together with eGFP."

Reviewer #3 (Remarks to the Author):

Stefania Miliuti and colleagues have molecularly elucidated that the properties of cancer stemness are controlled

by the RB-E2F family and WNT signaling using pancreatic and breast cancer cell lines. Furthermore, they advocate for the depletion of cancer stem cells and the restoration of sensitivity to anticancer agents by preventing their privileged entry into the G0 phase using the GCN5 inhibitor.

Reviewer #3's assigned review focused on aspects related to FUCCI. While the authors have addressed all of Reviewer #3's comments, there is still a deficiency in terms of photographic evidence in the context of FUCCI imaging.

Indeed, representative photographs of FUCCI fluorescence are essential in the sections related to FUCCI.

1. Reiterating the point, the FUCCI fluorescence in Figure 4l is too faint within the spheroids. It is difficult to discern with the current presentation.

Response 3.1. Thank you for this comment. We have now provided the improved image below and substituted it in the revised manuscript.

I

Advanced FUCCI CSCs

We would also like to emphasize a few aspects: 1) the spheres in Figure 4l are live FUCCI cells imaged without cell fixation and they are grown in 3D suspension. Hence, the cells within the sphere do not reside strictly on the same plane and the fluorescence signal appears more dim for some cells than for cells on the same imaging plane. 2) The FUCCI-CSCs have many non-fluorescent cells in a heterogeneously proliferating population because a large fraction of the cells reside in early G1 phase (see also Early G1 gate in Figure 4m) which has neither red (late G1 phase) nor green (S/G2/M phase) colour. These cells will gradually become red during entering the late G1 phase. 3) Since the FUCCI-CSCs depicted are live cells, we took the image with conventional fluorescence microscope because confocal microscope causes cell damage to live proliferating cells due to laser intensity, leading to increased cell death after imaging. We aimed to preserve the cells as healthy as possible for these experiments. The FUCCI image in spheres aims to provide just a visual confirmation that the signals are present and can be detected, which is the case.

2. In Supplementary Figure 6g, is it accurate to state that exposure to chemotherapy leads to late G1 phase arrest? Exposure to many chemotherapeutic agents typically results in G2/M checkpoint arrest, leading to an increase in the S/G2 phase. Please clarify this discrepancy.

Response 3.2. Thank you for this useful comment. We have now clarified the text to address this aspect in the manuscript in lines 499-505: "We treated the CSC-FUCCI spheres for 5 days with WNT7A/B or conditioned media collected from IWP2, pRb KD and RBL2 KD, and performed flow cytometry analysis with the FUCCI (Fig. 4l and Supplementary Fig. 6g-i). Gemcitabine treatment led to an increased fraction positive for both Cdt1-mRFP and p27k(-)-mKATE2 signals which marks G0-phase cells (Fig. 4l and Supplementary Fig. 6h). Since exposure to many chemotherapeutic agents typically results in G2/M checkpoint arrest or the induction of apoptosis upon extensive cytotoxic effects, these results indicate that Gemcitabine at the used concentrations leads mostly to cell death. Interestingly, media from IWP2-treated cells also increased G0-phase cells while WNT7A/B ligands and media from pRB KD or RBL2 KD cells led to a reduction in G0-phase cells (Fig. 4l and Supplementary Fig. 6h). The stimulation of proliferation by the WNT7A/B ligand also increased the susceptibility of cells to Gemcitabine-induced apoptosis (Supplementary Fig. 6i)."

3. The photographs of spheroids in Figure 6h are still too small. The cell clusters appear too diminutive, making it difficult to perceive them as a collective spheroid.

Response 3.3. Thank you. Regarding this aspect, we would like to mention that for this particular experiment, we took the shown representative bright field images 48 h before the standard end point due to practical reasons. This is the reason the spheres appear somewhat smaller on the representative images than usually, although the quantification of sphere numbers and sizes was still done by Celigo plate reader at the usual time point. We took the representative images of the treatments 48 earlier, because on that occasion, our aim was

to keep the cells as healthy as possible at the end point by avoiding keeping the plates out of the incubator for imaging (thereby avoiding prolonged lower temperatures and changes in CO₂ that could affect gene expression). At the expected endpoint, we still performed the Celigo analysis, and additionally, a total RNA extraction from the treatments for checking some gene expression changes. If the reviewer still insists that the representative images are not suitable then we can replace them.

4. It is important to note that the study did not explicitly demonstrate the WNT signal in the context of carcinogenesis, so it might be an overinterpretation to extrapolate extensively on this point.

Response 3.4. Thank you for this comment. We have added this aspect to the discussion section in line 991-995 as a limitation of the current study: “Deciphering the function of WNT signal in PDAC formation and its role particularly in PDAC CSCs, will benefit from *in vivo* tumorigenesis experiments to assess the relative importance of the WNT ligands in carcinogenesis, and the relative importance of RB1, RBL1 and RBL2 family members as regulators of WNT ligands in pancreatic tumorigenesis. In addition, the function of GCN5 as a possible therapeutic target in PDACs should be addressed in future *in vivo* studies.”

Reviewer #4 - scRNA-seq, ATAC-seq, stem cells (Remarks to the Author):

This is a quite thorough and well executed study demonstrating a role for E2F transcription factors in driving expression of Wnt ligands in pancreatic and breast cancer cells with the potential for autocrine and paracrine modulation of tumor growth and progression. The extensive bioinformatic analysis is rationally and appropriately applied using mostly established methods. However, in the interest of transparency, there are some critical missing details and labeling errors that need to be addressed prior to publication as well as some tempering of interpretation in these and other aspects of the work as enumerated below.

Concerns related to data analysis:

1. There is insufficient methodological detail presented regarding analysis of ligand-receptor pairs. What package (e.g. within Seurat, or through manual annotation of gene ontologies and cell types, etc.) was used for this analysis and the generation of corresponding circus plots? Similarly, the source annotations for gene ontology analysis should be specified and the specific approach for calculating differentially expressed genes and false discovery rates within computational suites should be specified or at least denoted as ‘default’. Critically, without functional analysis of source and target of these potential signaling axes *in vivo* they should be qualified as “putative or implied” paracrine mechanisms, or something similar. That is, the analytical models do not show extensive cellular crosstalk. They suggest or imply it.

Response 4.1. Thank you for this helpful comment.

- We have now provided further methodological details as requested and added the following text to our manuscript in the Materials and Methods section:

Ligand-Receptor pairs determination

PDAC patient single-cell RNA-seq data were obtained from the study of Steele et al.,⁴¹ where data from 20 treatment-naïve donors out of 27 total patients were included in this study. The paired reads were mapped to the hg38 reference genome to generate gene expression matrices using Cell Ranger v7.0.0 (<https://www.10xgenomics.com/cn/support/software/cell-ranger>). The raw matrices were then analyzed using the Seurat R package (v4)¹³⁰. Cells with low-quality profiles were excluded based on the number of detected genes (>400) and the total number of UMIs (>600). Mitochondria gene-enriched cells were further removed afterwards during clustering. The raw read counts were normalized using the NormalizeData function, and variable genes were identified for each sample. Data from different samples were then integrated using the FastMNN algorithm. To identify cell types, Uniform Manifold Approximation and Projection (UMAP) dimensionality reduction was performed on integrated data with the RunUMAP function. The FindNeighbors and FindCluster functions were used to cluster cells based on the global transcriptional profile.

For cell-cell interaction inference, normalized expression matrix and cluster annotations were exported from the Seurat object as input for cellphonedb. Cellphonedb v4.1.0¹³² was used as a ligand-receptor interaction

database. “statistical analysis” method from Cellphonedb tools was used to assess the strength and significance of interactions. For the circus plots we used Circlize package ¹³³, version 0.4.15 in R to draw the plots (<https://www.R-project.org/>).

References

41. Steele, N.G. *et al.* Multimodal Mapping of the Tumor and Peripheral Blood Immune Landscape in Human Pancreatic Cancer. *Nat Cancer* **1**, 1097-1112 (2020).
130. Hao, Y. *et al.* Integrated analysis of multimodal single-cell data. *Cell* **184**, 3573-3587 e3529 (2021).
131. Trapnell, C. *et al.* The dynamics and regulators of cell fate decisions are revealed by pseudotemporal ordering of single cells. *Nat Biotechnol* **32**, 381-386 (2014).
132. Garcia-Alonso, L. *et al.* Single-cell roadmap of human gonadal development. *Nature* **607**, 540-547 (2022).
133. Gu, Z., Gu, L., Eils, R., Schlesner, M. & Brors, B. circlize Implements and enhances circular visualization in R. *Bioinformatics* **30**, 2811-2812 (2014).

- We have also included “putative” and “implied” paracrine mechanisms in the text as suggested by the reviewer: line 239, line 240, line 546, line 538
- We have now added the the information on source annotations for gene ontology analysis and the approach for calculating differentially expressed genes and false discovery rates within computational suites in Materials and Methods:

RNA-sequencing analysis

Raw reads were cleaned using fastp v0.23.2 with default parameters ¹³⁵. Cleaned reads were confirmed high-quality using FastQC v0.11.9 ¹³⁶ and then mapped to the human genome hg38 using STAR v2.7.3a ¹³⁷. Mapped reads were quantified using featureCounts v2.0.0 ¹³⁸ and analyzed using DESeq2 v1.34.0 ¹³⁹. For differential gene expression analysis, the effect size was shrunk using the ashR method ¹⁴⁰, and p-values were adjusted using the independent hypothesis weighting (IHW) method ¹⁴¹. Significant differential gene expression was defined as $|\log_2 \text{Fold change}| > 1$ for upregulated genes and $|\log_2 \text{Fold change}| < -1$ for downregulated genes at an adjusted p-value < 0.05 .

Functional enrichment analysis

Source annotations for gene ontology analysis and the approach for calculating differentially expressed genes and false discovery rates within computational suites were kept at ‘default’ unless specified. For RNA-seq data, GSEA was performed using fgsea v1.22.0 ¹⁴². The gene sets used in GSEA were obtained from the Molecular Signatures Database v7.5.1 ¹⁴³. RNA-seq over-representation analyses was performed using g:GOST ¹⁴⁴.

References

135. Chen, S., Zhou, Y., Chen, Y. & Gu, J. fastp: an ultra-fast all-in-one FASTQ preprocessor. *Bioinformatics* **34**, i884-i890 (2018).
136. de Sena Brandine, G. & Smith, A.D. Falco: high-speed FastQC emulation for quality control of sequencing data. *F1000Res* **8**, 1874 (2019).
137. Dobin, A. *et al.* STAR: ultrafast universal RNA-seq aligner. *Bioinformatics* **29**, 15-21 (2013).
138. Liao, Y., Smyth, G.K. & Shi, W. featureCounts: an efficient general purpose program for assigning sequence reads to genomic features. *Bioinformatics* **30**, 923-930 (2014).
139. Love, M.I., Huber, W. & Anders, S. Moderated estimation of fold change and dispersion for RNA-seq data with DESeq2. *Genome Biol* **15**, 550 (2014).
140. Stephens, M. False discovery rates: a new deal. *Biostatistics* **18**, 275-294 (2017).
141. Ignatiadis, N., Klaus, B., Zaugg, J.B. & Huber, W. Data-driven hypothesis weighting increases detection power in genome-scale multiple testing. *Nature methods* **13**, 577-580 (2016).

142. Korotkevich, G., Sukhov, V. & Sergushichev, A. Fast gene set enrichment analysis. . *Preprint at bioRxiv* <https://doi.org/10.1101/060012> (2016).
143. Subramanian, A. *et al.* Gene set enrichment analysis: a knowledge-based approach for interpreting genome-wide expression profiles. *Proceedings of the National Academy of Sciences of the United States of America* **102**, 15545-15550 (2005).
144. Raudvere, U. *et al.* g:Profiler: a web server for functional enrichment analysis and conversions of gene lists (2019 update). *Nucleic Acids Res* **47**, W191-W198 (2019).

2. Genes used for manual cell type classification should be indicated.

Response 4.2. Thank you for this feedback. We used the following genes for cell type classification and we have now added this information in the Materials and Methods: “We used the following genes for cell type classification: epithelial/cancer cells (EPCAM, LCN2, AGR2, KRT8, KRT18), T cells (CD3D, CD3E), Myeloid cells (LYZ, CD14), NK cells (KLRF1m NCAM1), B cells (CD19, MS4A1, IGKC), fibroblasts (COL1A1, COL1A2, COL5A2, DCN, LUM), mast cells (TPSAB1, TPSB2, CPA3), neural cells (GPM6B, PLP1), endothelial cells (negative for listed markers).” The markers used for cell type classification are also shown in Figure 1h.

3. Supplementary Figure 1 G-I contains panels that are difficult to understand. What does the table indicate; how was it calculated/tabulated? What method was employed for Cell Trajectory analysis? Was the branchpoint designated as start? If so, why does pseudotime predict 0 at a distal position? The key in panel I does not match the plotted colors and these are not referenced coherently to the plot in G.

Response 4.3. Thank you for this helpful comment. We have now clarified the figure panels and provide more information.

- We added the following description to the figure legend:

“(g-i) Developmental trajectory analysis of CSCs in PDAC patient tumours based on single-cell RNA-sequencing analysis of patient tumour samples. (g) UMAP of cell populations in patient tumours. Ductal 1 and ductal 2 cell are cancer cell populations (circled populations), which contain Oct4+/CD133+/EPCAM+ cancer cells marked as CSCs in Ductal 1 (red) and CSCs in Ductal 2 (blue) population. The rest of the cells contain various cell types found in patient tumours (grey). Cluster information is shown. The table below UMAP plot indicates the number of cells for each cell population, and CSC cell numbers mark Oct4+/CD133+/EPCAM+ cancer cells. (h) Developmental trajectory of PDAC ductal cells based on single-cell RNA-sequencing analysis of patient tumour samples. Pseudo-time 0 is the starting point. Pseudo-time of ductal cells was inferred by Monocle2 and each point corresponds to a single cell. (i) Developmental trajectory of Oct4+/CD133+/EPCAM+ CSCs. Oct4+/CD133+/EPCAM+ CSCs progress on the developmental trajectory indicating that CSCs (red) in Ductal 1

(blue) is the earlier stage that gives rise to CSCs (purple) in Ductal 2 cells (green). These populations depict the same cells described in (g).”

- We also added the following description in the Materials and Methods:

Constructing single cell trajectories in PDAC

The Monocle2 package (v2.8.0)¹³¹ was used to analyze single cell trajectories in order to discover the cell-state transitions.

131. Trapnell, C. et al. The dynamics and regulators of cell fate decisions are revealed by pseudotemporal ordering of single cells. Nat Biotechnol 32, 381-386 (2014).

Minor:

4. Figure 1m is mislabeled in the figure legend.

Response 4.4. We have now corrected it.

5. Although indicated in the methods, line 224 should also reference the study from which single cell gene expression data is derived to clarify the analysis is carried out on previously published data.

Response 4.5. We have now added the citation to the text in line 224.

6. Fig. 2h does not appear to be referenced in the text.

Response 4.6. We have added the reference in the text in lines 283-284: “These WNT ligands are expressed in PDAC cells as shown by analysis of scRNA-seq data (Fig. 2h).”

Concerns related to other aspects of the work:

7. Extensive characterization of a subpopulation of cells with distinct marker and transcriptional profiles, and functional outputs, is described in Supplemental Fig. 1 but as there is still some controversy surrounding the potential stem like activity in other sub-populations in multiple cancer types, it would be helpful to delineate/clarify the CSC definition used here in the figure legend by adding the marker designation to the figure title or details. Also, at least one representative primary data figure of sphere growth should accompany the figure to inform what was counted in the bar graphs.

Response 4.7. Thank you for this helpful feedback. We have added the marker designation to the figure subpanels for clarification:

Subpanel a) A13A CSCs indicate the enrichment of CD44, CD133, SSEA4 and CXCR4 markers, whereas L3.6pl CSCs indicate the enrichment of CD44, CD133, SSEA4 and ALCG2 markers.

Subpanel b) Self-renewal of Oct4/CD133/SSEA4 CSC marker positive and negative cell populations in A13A and L3.6pl cells sorted by FACS followed by CSC sphere assays.

We have also added representative figures of adherently grown cells and spheres to Supplemental Fig. 1a.

8. Given that only Wnt8A and Wnt10B show subpopulation specific binding of E2Fs at their promoters, what is the evidence that those two Wnts specifically are required for activation of functional differences in the opposing compartments? The differential expression of those Wnts is notably absent from Fig. 1 L-M. Does the conclusion that effects are “non-cell-autonomous” as indicated in the title depend on this? This should be discussed. For instance, is a distinction between “cell autonomous” and “autocrine” intended? Is a paracrine target of CSC signaling clearly delineated? Related to this, various subtypes of fibroblasts exist within the PDAC microenvironment. From the scRNA-seq data in Fig S1, it would be informative to determine what subtype of CAF is most responsible for WNT crosstalk between CSCs and fibroblasts.

Response 4.8. Thank you for these questions.

- We would like to clarify that by comparing the ATAC-seq peaks between non-CSCs and CSCs, we noticed ATAC-seq peaks near WNT8A in non-CSCs whereas it was absent in CSCs, and ATAC-seq peak near WNT10B loci in CSCs whereas this peak was absent in non-CSCs. Importantly, as the reviewer indicates, WNT8A is not detectable in PDAC patient cells based on scRNA-seq, and WNT10B has a very low expression compared to several other WNT ligands (WNT7A, WNT7B, WNT10A). This leads us to conclude about WNT8A and WNT10B that the observed differences in ATAC-seq peaks near WNT8A/WNT10B loci between non-CSCs and CSCs do not lead to the activation of their expression. This was one of the reasons we focused our detailed experiments on WNT7A, WNT7B, WNT10A and WNT4. These four WNT ligands are clearly expressed in PDAC cells/CSCs whereas WNT8A and WNT10B are not expressed and seem to have negligible impact of PDAC development. Our conclusion of “non-cell-autonomous” effects in the title depend on our results describing the function of WNT7A, WNT7B,

WNT10A and WNT4. We have added this in the discussion section of the manuscript in a shortened form.

- In the text, we have tried to make a distinction between the terms “cell autonomous”, “autocrine” and “paracrine”. By “cell autonomous” we have described the intracellular gene expression mechanisms inside the nucleus that do not involve secretion of factors (e.g. E2F function inside the cells). By “autocrine” mechanisms we describe WNT ligand mediated mechanisms where these ligands are produced by the cell type and receives the cells (e.g. autocrine signalling between CSCs). By “paracrine” mechanisms we mean signalling between different cell types (e.g. CSCs and fibroblasts; CSCs and non-CSCs). We hope this is acceptable for the reviewer.
- We have shown that CSCs provide paracrine signalling to fibroblast subtypes and added the following results to our manuscript: “We further aimed to determine what subtype of CAF is most responsible for WNT crosstalk between CSCs and fibroblasts. Detailed scRNA-seq analysis of PDAC patients indicated the presence of different types of CAFs such as myCAFs, apCAFs, iCAFs, pericytes, epithelial-like, chondrocyte-like, peri-islet Schwann cells, and proliferating CAFs (**Supplementary Fig. 6j-k**) which have been identified previously in PDACs⁶⁶. The ligand-receptor pairs mediating the putative crosstalk between CSCs and the different types of CAFs revealed for instance, the possible crosstalk between CSCs and proliferating CAFs mediated by WNT7B-FZD5/6-LRP5/6, and WNT7B-FZD2/8-LRP5/6 between CSCs and iCAFs as well as myCAFs (**Supplementary Fig. 6l**). Therefore, CSCs are likely to be in functional crosstalk with several cell types in PDACs, including several fibroblast subtypes, which contribute to the characteristics and the CSC niche in the tumour.”

- We added the figure legend as follows: (j) UMAP of the different subtypes of fibroblasts in PDAC patient tumours. (k) Marker genes used to detect the different subtypes of fibroblasts in PDAC patient tumours. (l) Paracrine signalling between CSCs (PDAC Cluster 1) and subtypes of fibroblasts in PDAC patient primary tumours.

9. Fig. 2l: Typically, phosphorylated pRb would be compared to total pRb levels.

Response 4.9. Thank you for this comment. We have now added the total pRb levels to Fig. 2l.

10. Line 154-157: The authors argue that the increase in CSC markers after treatment with various therapies is a result of cell-death of non-CSC cells. Is the possibility that exposure to treatments causes non-CSC cells to adopt a CSC-like state excluded?

Response 4.10. Thank you for this very interesting question. In our experiments, we have treated cancer cells for 5 days and we observe extensive cell death already after 48 hours and the dying cells lift off the plate until the 5 day time point. In these experiments, we used relatively high concentrations of chemotherapy reagents which we had determined on survival curve experiments, which showed efficient induction of cell death for the majority of cells at the selected reagent concentration. Of note, the percentage of cells that survives the chemotherapy reagent treatment at our tested concentrations is comparable or even somewhat lower compared to the percentage of cells that co-express the CSC markers in cell populations that have not been treated with chemotherapeutic reagents. Therefore, we believe that the large majority of cells that survive the chemotherapy treatment were already expressing the CSC markers before the treatment. However, it is true we cannot exclude the possibility that for a rather small fraction of cells the exposure to treatment has caused non-CSC cells to adopt a CSC-like state. At our selected reagent concentrations, this cell fraction is likely to be small. On the other hand, the scenario proposed by the reviewer could occur more frequently if we would use a lower dose of the chemotherapy reagents, which allow more cells to survive the initial treatment and adopt a CSC-like state over prolonged incubation and thereby acquiring a resistant phenotype.

11. Line 229-233: The authors claim PDAC_C1 cluster visualized in Fig S2 contains higher levels of CSC markers and elevated WNT family member expression however this difference appears to be quite modest compared to other clusters (i.e. PDAC_C3). Statistical analysis of enrichment of these CSC/WNT genes in the C1 cluster should be performed to demonstrate this difference.

Response 4.11. Thank you for this question. We have now provided the statistical analysis of enrichment of these genes in the C1 cluster and added these results to Supplementary Fig. 2d). We revised the text in the manuscript as follows: "This cancer cell subpopulation also showed statistically significant elevated expression of WNT7B, WNT7A, WNT10A and WNT4 besides the CSC markers, compared to the rest of the PDAC cells (**Supplementary Fig. 2c-d**)." We also added the Figure legend: "(d) Table showing the statistical significance and fold changes of each genes in PDAC_C1 cluster compared to all other PDAC cancer cell clusters C2-C8 combined."

d

Gene Symbol	P values	log2(average fold change)	Adjusted P value
WNT7B	1.7E-12	0.500911692	6.13E-08
WNT7A	1.1E-06	0.103666514	4.05E-02
WNT10A	2.6E-14	0.491873821	9.53E-10
WNT4	3.7E-36	0.606578146	1.34E-31
CCND1	5.1E-90	1.055247741	1.86E-85
CCNE1	5.4E-01	0.049239927	1.00E+00
FZD6	2.0E-24	0.347205477	7.42E-20
FZD5	1.6E-54	1.378857695	5.93E-50
E2F1	5.2E-02	0.258231879	1.00E+00
E2F4	2.0E-19	0.093530481	7.17E-15
CDK4	1.6E-03	0.076765712	1.00E+00
CDK6	1.8E-21	0.110738306	6.69E-17
CDK2	4.5E-05	0.007054789	1.00E+00
ALDH1A1	1.7E-72	1.574436323	6.28E-68
PROM1	1.5E-43	1.404276581	5.56E-39
KLF4	1.5E-36	0.501374677	5.33E-32

12. Fig 2M: The mechanism by which Palbociclib causes decreased WNT expression is unclear. Namely, are the effects due to decreased WNT expression by CSCs or a depletion of the CSC population?

Response 4.12. Thank you, this is an intriguing question. To try to answer this, we analysed the relative expression of CSC markers in H6C7+KRASmut cells and H6C7+KRASmut+CDK4/6 inhibitor treated cells by flow cytometry as we had done for the WNT ligands. We reasoned that if the CSC population is depleted by Palbociclib during the 48 hour treatment, we might see a reduction in the cells expressing CSC markers that we used in our CSCs. Our results indicated that the relative expression for CD133, OCT4, and SSEA4 is not significantly different. Hence, our results suggest that the short (48 hour) treatment with Palbociclib is likely to be mediated by CDK4/6-mediated effects on E2F-pRb. However, a long treatment of cells with Palbociclib could have additional effects that could also impact cell differentiation/dedifferentiation balance.

13. Fig 8S d.: KM plots should be stratified by clinical or PAM50 subtype as CDk4/6 inhibitors + endocrine therapy is standard of care for HR+ tumors in the context of metastatic disease and may confound the analysis.

Response 4.13. Thank you for this useful suggestion. We have now provided the BRCA subtypes in addition to the compiled total BRCA Kaplan-Myer plots. Interestingly, high WNT7A expression suggests a correlation trend with higher mortality in HER2+ non-luminal subtype BRCA (p=0.11), whereas high WNT3A expression suggests a correlation trend with higher mortality in Luminal B subtype BRCA (p=0.059), and WNT9A in Luminal A subtype BRCA (p=0.072). Therefore, different WNT ligands may have more pronounced roles in specific BRCA subtypes.

We have added this information in the manuscript text and included the KM plots based on BRCA subtypes in Supplementary Figure 8: "Among the WNT ligands, elevated expression of WNT7B, WNT9A, WNT3A in top 25% expressing versus bottom 25% expressing all BRCA patients combined correlate significantly with lower patient survival (Supplementary Fig. 9d-e). BRCA subtype stratification in addition to the compiled total BRCA Kaplan-Myer plots provide further insight. Interestingly, top 50% expressing versus bottom 50% expressing WNT7A patients suggests a correlation trend with higher mortality in HER2+ non-luminal subtype BRCA (p=0.11), whereas high WNT3A expression suggests a correlation trend with higher mortality in Luminal B subtype BRCA

($p=0.059$), and WNT9A in Luminal A subtype BRCA (s) ($p=0.072$) (Supplementary Fig. 9d). Therefore, different WNT ligands may have more pronounced roles in specific BRCA subtypes, altogether suggesting that WNT ligand expression regulation by the RB-E2F axis has an important function on disease progression of BRCA patients.”

14. Page 22: MCF-7 are more luminal than a basal subtype (such as HCC1500 and Cal-51), therefore there are multiple additional biological differences besides RBL2 expression that might explain the differences in Wnt expression.

Response 4.14. Thank you for highlighting this aspect. We mentioned it in the manuscript text in line 804-806: “MCF-7 are more luminal than a basal subtype (such as HCC1500 and Cal-51), therefore there are multiple additional biological differences besides RBL2 expression that might explain the differences in WNT ligand expression.”

Reviewers' Comments:

Reviewer #2:

Remarks to the Author:

No further issues

Reviewer #3:

Remarks to the Author:

The authors claim that confocal laser microscopy leads to cell death in FUCCI imaging, but in Reviewer #3's in vitro sphere experiments and real-time imaging in tumors in vivo, little cell death occurred after laser irradiation. Also, since the G0 phase is colorless, I am skeptical that experiments using FUCCI are meaningful if FUCCI imaging of spheres cannot be substituted for analysis by flow cytometry.

The authors did not consistently exchange photographs, despite our request for strongly magnified photographs.

Based on the above, the authors have not responded to Reviewer #3's points. I leave further decisions to the editorial board.

Reviewer #4:

Remarks to the Author:

The authors have satisfactorily addressed all the concerns and questions raised.

Point by point response

General Response

We would like to thank the Reviewers and the Editor for their helpful feedback and insightful suggestions. We provide a point-by-point response that addresses all the comments from the reviewers with extensive additional experimental data and textual changes to the revised manuscript. We have highlighted the changes within the revised manuscript in blue font colour, and added the additional research data into the manuscript figures and text as described in the point-by-point response.

Collectively, the feedback from the reviewers has further improved the manuscript and emphasises its timeliness for publication.

REVIEWERS' COMMENTS

Reviewer #2 (Remarks to the Author):

No further issues

Response

We are glad we have satisfactorily answered all Reviewer #2 comments.

Reviewer #3 (Remarks to the Author):

The authors claim that confocal laser microscopy leads to cell death in FUCCI imaging, but in Reviewer #3's in vitro sphere experiments and real-time imaging in tumors in vivo, little cell death occurred after laser irradiation.

Response

Thank you for this comment. Prolonged exposure of cells to laser irradiation can cause cellular stress in live cells and at high laser intensities for long time, this stress can increase susceptibility to apoptosis. This is a broadly known fact. In our experience, some cell types are more sensitive than others and laser intensities can be different. Hence, the general goal is to keep the exposure of cells to confocal lasers to a minimum (by avoiding prolonged exposure) if the objective is to culture or analyse the FUCCI cells further. This was our reasoning.

Also, since the G0 phase is colorless, I am skeptical that experiments using FUCCI are meaningful if FUCCI imaging of spheres cannot be substituted for analysis by flow cytometry.

Response

We are sorry but there are inaccuracies in this statement. The G0 phase is not colorless in our system but expresses high mKATE2 and RFP, we clearly have described the FUCCI system in Figure 4J and provided further experimental proof. Early G1 phase cells are colourless but not G0 cells.

I am skeptical that experiments using FUCCI are meaningful if FUCCI imaging of spheres cannot be substituted for analysis by flow cytometry.

Response

This comment by the reviewer is unfortunately factually flawed as shown by the experimental data we already provided in our previous point by point response. We clearly can use both FUCCI imaging of sphere as well as flow cytometry with this system, as shown by our data so the concerns from the reviewer are unwarranted. We have used the FUCCI cells extensively for cell sorting experiments.

Previously, we added the following to the manuscript: “We verified the FUCCI system by flow cytometry analyses of the advanced FUCCI (RFP-hCdt1(30/120)_mAG-hGEM(1/110)_mKate2-p27(K-))

integrated into the AAVS1 locus in A13A PDAC CSCs show the signals of late G1 (hCdt+), G1/S transition (hCdt+/Geminin+), S/G2/M (Geminin+), and early G1 (hCdt-/Geminin-) (Supplementary Fig. 6c). The visualisation of hCdt and p27 shows the separation of hCdt+/p27- and hCdt+/p27+ cells (Supplementary Fig. 6d). Cells residing in the G0 phase are hCdt+/p27+ and they should be Ki67 negative, which is a

marker for proliferating cells. The staining of FUCCI-CSCs for Ki67 shows a population of cells with very low Ki67, which marks G0 phase cells (**Supplementary Fig. 6e**). Gating of p27+ cells (highest signal for p27) shows that these cells are indeed negative for Ki67 signal (**Supplementary Fig. 6f**), thus confirming that the advanced FUCCI is able to distinguish G0 cells in PDAC.”

We also included representative confocal microscopy images of 2D PDAC FUCCI-A13A cells to show the possibility of separating cells that are p27+/hCdt1+ or only hCdt1+.

The authors did not consistently exchange photographs, despite our request for strongly magnified photographs.

Response

We apologize but there seems to be another confusion. In the previous point-by-point request, there is no indication that that the reviewer asks for “consistent exchange” of previous photographs to “strongly magnified” photographs. The previous letter does not contain any wording that requests “strongly magnified” photographs. In Figure 4I we provided the requested brighter FUCCI 3D CSC spheres in suspension as shown below which even allows to distinguish between individual cells due to the overlay with bright field.

Advanced FUCCI CSCs

Based on the above, the authors have not responded to Reviewer #3's points. I leave further decisions to the editorial board.

Response

We hope our clarification has been sufficient at this stage but we are glad to provide further information if the editorial board needs.

Reviewer #4 (Remarks to the Author):

The authors have satisfactorily addressed all the concerns and questions raised.

Response

We are glad we have satisfactorily answered all Reviewer #4 comments.